Registered Report

# Social media use in adolescents with and without mental health conditions

**Luisa Fassi** [1,2] ✉**, Amanda M. Ferguson**[1]**, Andrew K. Przybylski** [3]**, Tamsin J. Ford**[2] **& Amy Orben** [1]

Concerns about the relationship between social media use and adolescent mental health are growing, yet few studies focus on adolescents with clinical-level mental health symptoms. This limits our understanding of how social media use varies across mental health profiles. In this Registered Report, we analyse nationally representative UK data (*N* = 3,340, aged 11–19 years) including diagnostic assessments by clinical raters alongside quantitative and qualitative social media measures. As hypothesized, adolescents with mental health conditions reported spending more time on social media and were less happy about the number of online friends than adolescents without conditions. We also found hypothesized differences in social media use by condition type: adolescents with internalizing conditions reported spending more time on social media, engaging in more social comparison and experiencing greater impact of feedback on mood, alongside lower happiness about the number of online friends and lower honest self-disclosure. In contrast, those with externalizing conditions only reported higher time spent. These findings emphasize the need to consider diverse adolescent mental health profiles in policy and clinical practice.

Adolescents around the world have experienced a decline in their mental health over the past decade[1]. Recent UK data suggests that one in six 7–16-year olds and one in four 17–19-year olds have a probable mental health condition, a clear rise from the one in nine and one in ten recorded in 2017, respectively[2]. As 48% of those with a mental health condition first experience relevant symptoms before the age of 18 years[3], this increased mental health burden will negatively impact society and the economy, as well as adolescent and adult life[4]. Many have raised concerns that this trend has been caused, at least in part, by increased adolescent social media use, which has revolutionized how adolescents live, learn and interact: 93% of 12–17-year olds now report having a social media profile[5].

To address these concerns, academic investigation of social media use and adolescent mental health has increased substantially in recent years[6]. Research teams have recruited adolescent populations in schools, universities or as part of broader community-based samples to identify cross-sectional and longitudinal links between increased smartphone or social media use and scores on questionnaires

of depression[7,8], anxiety[9], disordered eating[10] and other mental health symptoms[10–12]. These studies have primarily found small positive associations. Some researchers have used these to argue that there exists a causal link between social media use and mental health declines (that is, "screen time, perhaps especially social media, may have larger effects on adolescent girls' mental health than on boys' and that is indeed what we found, with social media significantly correlated with depressive symptoms [...]"[13] p. 13). Such arguments, in turn, have been used to call for restrictive policy regulations to limit smartphone and social media use in adolescent age groups[14].

However, many researchers have also questioned the strength of the current evidence base and highlighted that existing studies do not support the idea that there is a causal relationship linking social media use to mental health. Indeed, the literature provides many conflicting results[15]. Researchers have not only debated about a lack of longitudinal or causal evidence[16], but have also disagreed about what effect sizes matter[17–19] and how to deal with the substantial individual differences present[20,21], which have been linked to factors such as age[22], gender[23,24] and ethnicity[25].

[1]MRC Cognition and Brain Sciences Unit, University of Cambridge, Cambridge, UK. [2]Department of Psychiatry, University of Cambridge, Cambridge, UK. [3]Oxford Internet Institute, University of Oxford, Oxford, UK. ✉e-mail: Luisa.Fassi@mrc-cbu.cam.ac.uk

Across these topics of debate, however, researchers have largely overlooked how the kind of instruments used to measure mental health, as well as the populations being studied, limits their ability to draw meaningful inferences about the relationship of social media with adolescent mental health in the first place. So far, most studies have examined school- or community-based adolescent samples[15,16,26], relating scores on mental health questionnaires (for example, the Hospital Anxiety and Depression Scale[9,27]) to time spent on social media. The rationale for doing so is that questionnaires capturing continuous clinical symptoms are informative when reasoning about social media use in relation to the whole spectrum of mental health, across types of severity and clinical presentation. However, this approach is not a suitable surrogate for studying links between social media and mental health in adolescents with versus without mental health conditions, for two main reasons. First, it reduces the complexity of clinical presentations to the tail end of variation in selected mental health symptoms among mostly healthy individuals. Second, it ignores the potentially important differences between those who endorse symptoms on a questionnaire and those who reach diagnostic criteria in standard clinical classifications. For example, an adolescent can score very highly on a questionnaire measuring depressive symptoms, but not meet the criteria for a diagnosis if the queried symptoms have only been present for a short time or if they are better explained by a different condition or situation.

To address this issue, select studies have moved beyond such an approach, dichotomizing symptom severity by applying cut-off scores to mental health questionnaires to reflect the presence or absence of a mental health diagnosis[23]. However, dichotomization does not solve many of the problems highlighted above and is known to have low sensitivity when predicting clinical diagnoses. Indeed, those with a mental health condition can score below the threshold on some scales[28,29]. In other words, although researchers would like to demonstrate the presence or absence of specific links between mental health conditions and social media use, the measures of psychopathology they employ might not be appropriate for these goals.

Importantly, the assumption that patterns of social media use found in non-clinical or community samples will generalize to those with mental health conditions has not yet been systematically tested. To our knowledge, only a few studies—most qualitative—have documented different social media use experiences in clinical adolescent populations, including those fulfilling stringent diagnostic criteria for a clinical condition, attending mental health services or being hospitalized for suicidal ideation and suicide attempts[30–33]. Adolescents in these studies reported both positive and negative social media experiences, such as enhanced social connection and trouble downregulating their use. Broadly speaking, these experiences aligned with established risk and protective factors previously linked to mental health in offline spaces and suggest there is no clear-cut positive or negative association between mental health and social media use. The studies also raise the idea that vulnerable youth might experience heightened emotional responses to social media use. However, this has not been directly assessed due to the lack of non-clinical comparison groups. Such comparisons are therefore necessary to identify differences in social media use between adolescents with and without mental health conditions.

Owing to the lack of research among young people with mental health conditions[26], the important question of whether social media use varies across different types of conditions also remains unaddressed. For example, an adolescent with an internalizing condition (for example, generalized anxiety disorder or depressive disorder) might use and feel impacted by social media differently than an adolescent with an externalizing condition (for example, attention deficit hyperactivity disorder or conduct disorder). This is because, despite both groups presenting mental health symptoms at clinical levels, their experiences of psychopathology can be qualitatively different. Internalizing conditions involve negative emotionality towards the

**Table 1 | Summary of our categorization of mental health conditions into internalizing and externalizing**

| Grouping of mental health conditions | List of mental health conditions diagnosed in the DAWBA |
|---|---|
| Any | Separation anxiety disorder, generalized anxiety disorder, obsessive–compulsive disorder, specific phobia, social phobia, agoraphobia, panic disorder, post-traumatic stress disorder, other anxiety disorder, major depressive episode, other depressive episode, hyperkinetic disorder, other hyperactivity disorder, oppositional defiant disorder, conduct disorder (general, confined to family, unsocialized, socialized and other), attention deficit hyperactivity disorder, other disruptive behavioural disorders, any behavioural disorder, autism spectrum disorder, eating disorder, tic disorder, selective mutism, psychosis, body dysmorphic disorder, bipolar affective disorder, mania, attachment disorder, feeding disorder, sleep disorder and eliminating disorder. |
| Internalizing | Separation anxiety disorder, generalized anxiety disorder, obsessive–compulsive disorder, specific phobia, social phobia, agoraphobia, panic disorder, post-traumatic stress disorder, other anxiety disorder, body dysmorphic disorder, major depressive episode, other depressive episode and eating disorders. |
| Externalizing | Hyperkinetic disorder, other hyperactivity disorder, oppositional defiant disorder, conduct disorder (general, confined to the family, unsocialized, socialized and other), attention deficit hyperactivity disorder, other disruptive behavioural disorders and any behavioural disorder. |
| Excluded[a] from questions 2 and 3 | Autism spectrum disorder, tic disorder, psychotic disorders, mania and bipolar affective disorder. |

[a]We exclude adolescents diagnosed with these conditions as they do not clearly map onto the symptomatology of either internalizing or externalizing diagnoses[34,65]. Further, because the diagnostic variance of mania and bipolar disorder accounted for by internalizing pathology is lower than most other internalizing disorders, such as anxiety and depression[33], we decided to exclude these diagnoses from the internalizing category, together with autism spectrum disorder, tic disorder and psychotic disorders. DAWBA, Development and Wellbeing Assessment.

self, expressed through ruminative thought patterns, worries and social withdrawal[34]. On the contrary, externalizing conditions involve negative emotionality towards others, expressed through impulsivity, risk taking and disinhibition[35]. Studies that assess mental health with select questionnaires cannot comprehensively account for and investigate such clinical diversity. This is a substantial shortcoming, given the need for research to understand how social media use relates to the growing number of adolescents experiencing mental health symptoms at clinical levels.

This Registered Report provides critical data and evidence-based insights into how social media and mental health are related across adolescent populations who meet and do not meet diagnostic criteria for a wide range of mental health conditions. Given the cross-sectional nature of the data and planned analyses, the study results do not provide causal evidence. Hence, all reported coefficients indicate associations, with the possibility of bidirectional relationships and third variables affecting social media use, mental health or the relationship between the two. We analysed the nationally representative Mental Health of Children and Young People (MHCYP) study[36], a cross-sectional survey carried out by National Health Service (NHS) Digital in 2017 that collected data from over 3,000 adolescents (11–19 years old) in England. In place of completing self-report measures of mental health, the participants in this study underwent multi-informant standardized diagnostic assessments evaluated by professional clinical raters for different mental health conditions (Table 1). We note that, in the stage 1 report, the terms 'clinical and non-clinical population' were used, but in stage 2 this was changed to 'adolescents with and without mental health conditions' to clarify that participants in this study were not

recruited or diagnosed by a clinic, but instead underwent the mental health assessment as part of the MHCYP study.

Further, to gain a comprehensive understanding of how social media use differs across adolescents with and without a mental health condition, we examined both quantitative and qualitative dimensions of social media use. Measuring only time spent provides a crude and simplistic estimate of social media use, conflating distinct analytical levels and missing a rich range of psychological factors such as appraisal and motivations that might vary as a function of mental health[37]. Researchers have therefore called for quantitative time-based measures of social media use to be complemented by more qualitative engagement-based measures capturing adolescents' social media activities and their appraisal of them[16,38–42]. Such practice is, however, still relatively rare. In this Registered Report, we included both types of measures, namely, time spent on social media and dimensions of social media engagement that could incur mental health risks (that is, online social comparison, monitoring and impact of online feedback and lack of control over time spent online) or benefits (that is, online friendship, as well as opportunities for honest self-disclosure and authentic self-presentation). By complementing quantitative and qualitative dimensions of social media use, this work provides a more solid foundation for mechanistic research aimed at informing future targeted interventions, clinical practice and policy actions benefitting adolescent mental health.

We used existing literature on adolescents' mental health in relation to both online and offline contexts (Table 2) to guide our hypotheses and analyses of the data along three lines of enquiry. Specifically, we evaluated whether social media use differs in adolescents with versus without a mental health condition (Question 1), with an internalizing or externalizing condition versus without a condition (Question 2) and with an internalizing versus externalizing condition (Question 3).

First, we expected adolescents with any mental health condition to report engaging with social media differently than those without a condition. For instance, gathering information about peers is particularly important during adolescence when young people develop a sense of personal and social identity[43]. However, high levels of upward social comparison (that is, comparisons with those believed to be of higher status than the self) have been associated with poorer mental health[44–48]. Previous work suggests that most social comparisons made on social media sites are upward rather than downward[49], possibly because individuals tend to portray themselves in an ideal manner online[50,51]. Social media could further amplify these processes as platforms offer continuous and more concrete opportunities for comparing oneself with others, such as browsing profiles without initiating social interaction[52]. Indeed, engagement in self-denigrating online social comparisons was a common theme raised by adolescent psychiatric inpatients during qualitative interviews[31]. We therefore expected adolescents with a mental health condition to engage in more online social comparison than those without a condition (hypothesis H1.1b). Similarly (see Table 2 for a detailed overview of the literature in support of each hypothesis), we expected them to spend more time on social media[8,15] (H1.1a), report more lack of control over time spent online[31,53] (H1.1c), while also monitoring[54,55] (H1.1 d) and feeling more impacted by online feedback[56,57] (H1.1e).

In contrast, we expected adolescents with any condition to engage less with social media in ways that might be protective for their mental health. For instance, we hypothesized that adolescents with a mental health condition are less happy with the number of friends they have online than adolescents without a condition (H1.2f). This is because social connections protect against long-term adverse physical and emotional outcomes, particularly during adolescence[48], and young people with mental health conditions often report difficulties with peers, having few friends and wanting to be alone[58–60]. Similarly, we expected adolescents with a mental health condition to engage in less honest online self-disclosure[61,62] (H1.2g) and authentic online

self-presentation[63,64] (H1.2h) compared with adolescents without a condition (Table 2).

Second, we predicted social media use to vary between adolescents with different symptomatology, focusing specifically on how the social media use of adolescents with an internalizing or externalizing condition (Table 1) differed from those without a condition.

We know from the mental health literature that internalizing conditions involve negative self-views, rumination, worries and social withdrawal[34,65]. Notably, these symptoms could be relevant to how young people present themselves and engage with others on social media[31,32,66]. For instance, adolescents with internalizing conditions are more likely to notice discrepancies between their ideal and actual selves[67] and may compensate for these discrepancies via impression management offline. We expect this process to also occur online, given the multiple affordances social media platforms offer to curate one's image (for example, deleting old posts and editing new posts). Hence, we hypothesized that adolescents with internalizing conditions would be less likely to engage in authentic self-presentation on social media than adolescents without a mental health condition[68–70] (H2.2h). Further (Table 2), we hypothesized that adolescents with internalizing conditions would spend more time on social media[8,15,71] (H2.1a), engage in more online social comparison[30] (H2.1b) and online feedback monitoring[54] (H2.1d), feel more impacted by online feedback[56] (H2.1e), be less happy about online friends (H2.2f) and engage in less honest self-disclosure[62] (H2.2g) than those without a condition. In contrast, we did not expect a difference in whether they perceive a lack of control over the time they spend online (H2.0c)[53].

Compared with internalizing conditions, externalizing conditions involve impulsivity, low self-monitoring and risk taking[34,65]. These symptoms could be reflected in how social media is used by these groups. Hence, we hypothesized that adolescents with externalizing conditions would be more likely than adolescents without a condition to spend more time on social media[72] (H2.3a) and perceive they lack control over the time they spend online[53] (H2.3c). Further, we expected them to also be more dissatisfied with their number of online friends[73] (H2.4f). In contrast, we did not expect differences in the other dimensions of social media use (H2.0b,d,e,g,h; Table 2), since they primarily relate to symptoms of internalizing rather than externalizing conditions.

Our third question examined whether adolescents with internalizing conditions use social media differently than those with externalizing conditions. Specifically, we expected adolescents with internalizing conditions to report engaging in more online social comparison[30] (H3.1b) and online feedback monitoring[55] (H3.1d), feeling more impacted by online feedback[56] (H3.1e), engaging in less honest self-disclosure[61] (H3.2g) and less authentic self-presentation on social media[69] (H3.2h) than those with externalizing conditions. In contrast, we expected adolescents with externalizing conditions to report having less control over the time they spend online[53,73] compared to adolescents with internalizing conditions (H3.2c). Further, since adolescents with internalizing conditions tend to be unhappy with their social status[74] and adolescents with externalizing conditions tend to have trouble making and keeping friends in both online and offline contexts[73,75], we expect both groups to be similarly dissatisfied with their number of online friends (H3.0f). We also hypothesized that both groups would not differ in the amount of time they spend on social media (H3.0a), given that both adolescents with internalizing and externalizing symptoms have been reported to spend more time on social media than adolescents without these symptoms[33,76].

Altogether, this study comprehensively maps and compares different dimensions of social media use in adolescents with and without mental health conditions. Hence, it will lay the foundation for future mechanistic and translational research studying which specific social media dimensions relate to mental health in different adolescent groups. This will be a crucial first step to inform translational research

**Table 2 | Review of key social media and mental health literature used to formulate the hypotheses of our study**

| Social media use | Scientific literature | Relevant hypotheses | Key references |
|---|---|---|---|
| **Time spent on social media**<br>Item: "When you use social media sites or apps how much time in total do you spend using them on a typical school day/ weekend?" | Research on the relationship between social media use and mental health in different samples has yielded limited and conflicting results. On the one hand, studies have found that young people diagnosed with depression report spending more time on social media compared with non-clinical controls[33]. Further, studies that focus on excessive rather than average time spent on social media show comorbidity between anxiety, depression, attention deficit hyperactivity disorder and excessive time spent on social media[72]. However, in a sample of hospitalized adolescents with psychiatric conditions, the frequency of social media use and perception of overuse was not associated with clinical severity[32]. Additionally, in an independent clinical sample, using social media less, both overall and for messaging, was linked to higher levels of suicidal ideation over the next 30 days[106]. There is more work on community samples available (for example refs. [26,71]) examining time spent on social media (mostly self-reported) and its relation to depression, anxiety and other indicators of mental health. Recent reviews and meta-analyses have reached a general agreement that associations are weak and positive (higher social media use is linked with higher levels of anxiety and depression[8,15,107]). Overall, despite mixed evidence, research therefore seems to suggest a small relationship between more time spent on social media and lower mental health, both when considering internalizing and externalizing symptoms. | H1.1a: adolescents with any mental health condition will spend more time on social media than adolescents without a condition.<br>H2.1a: adolescents with internalizing mental health conditions will spend more time on social media than adolescents without a condition.<br>H2.3a: adolescents with externalizing mental health conditions will spend more time on social media than adolescents without a condition.<br>H3.0a: adolescents with externalizing mental health conditions will not differ from adolescents with internalizing conditions in time spent on social media. | Valkenburg et al.[15]<br>Cunningham et al.[8]<br>George et al.[71]<br>Gürbüz et al.[33]<br>Fassi et al.[26]<br>Hussain & Griffiths[72]<br>Hamilton et al.[106]<br>Riehm et al.[107]<br>Nesi et al.[32] |
| **Online social comparison**<br>Item: "I compare myself to others on social media sites and apps" | In offline contexts, social comparison provides means of gathering information about the social world. This is particularly important during adolescence when young people need to develop a sense of personal and social identity and adjust to bodily changes[43]. Despite social comparison being instrumental for maturation into adulthood, its exacerbation has been associated with poor mental health, particularly in relation to internalizing symptoms. For instance, adolescents with anxiety and eating disorders engage in less favourable and more frequent social comparisons than adolescents without these conditions[44–48]. Social media platforms offer continuous and more concrete opportunities for social comparison than offline contexts, for instance, by allowing people to browse others' profiles without initiating social interaction[52]. Indeed, adolescents with a depression diagnosis reported unfavourably comparing themselves with others when using social media[30]. Similarly, among adolescents hospitalized for suicidal behaviour, 30% reported engaging in self-denigrating social comparisons, particularly body-related ones[31]. | H1.1b: adolescents with any mental health condition will engage in more online social comparison than adolescents without a condition.<br>H2.0b: adolescents with externalizing mental health conditions will not differ in online social comparison from adolescents without a condition.<br>H2.1b: adolescents with internalizing mental health conditions engage in more online social comparison than adolescents without a condition.<br>H3.1b: adolescents with internalizing mental health conditions will engage in more online social comparison than adolescents with externalizing conditions. | Krayer et al.[43]<br>Corning et al.[44]<br>Goodman et al.[45]<br>Rao et al.[48]<br>Thwaites & Dagnan[46]<br>Troop et al.[47]<br>Pempek et al.[52]<br>Radovic et al.[30]<br>Weinstein et al.[31] |
| **Perceived lack of control over time spent online**<br>Item: "In general, I spend more time on social media than I mean to" | Difficulties in managing personal goal pursuit in the face of internal, interpersonal and environmental forces that could impede it have been closely linked to the symptomatology of different mental health conditions, particularly externalizing ones. For example, attention deficit hyperactivity disorder and conduct problems are characterized by lower levels of self-monitoring, self-instruction and goal setting[53]. Technological developments now allow individuals to access social media at any location or time of the day. Consequently, online platforms are not limited to a particular environment and repetitions of certain behaviours, such as opening an app or scrolling through one's feed[108], might result in adolescents feeling unable to reduce the time they spend online despite being motivated to do so. We expect difficulties in self-regulation and goal pursuit to also reflect in how adolescents with mental health conditions engage with social media. Indeed, in a sample of adolescent psychiatric inpatients, self-regulation of social media behaviour was associated with mental health symptoms[109]. Further, in qualitative interviews, 40% of inpatients hospitalized for suicidal behaviour reported trouble regulating social media use and feeling 'addicted'[31]. | H1.1c: adolescents with any mental health condition will be more likely to lack control over time spent online than adolescents without a condition.<br>H2.0c: adolescents with internalizing mental health conditions will not differ in lack control over time spent online from adolescents without a condition.<br>H2.3c: adolescents with externalizing mental health conditions will be more likely to lack control over time spent online than adolescents without a condition.<br>H3.2c: adolescents with externalizing mental health conditions will be more likely to lack control over time spent online than adolescents with internalizing conditions. | Strauman et al.[53]<br>Winds et al.[109]<br>Weinstein et al.[31]<br>Bayer et al.[108] |
| **Monitoring of online feedback**<br>Item: "I monitor the amount of likes, comments and shares I get on social media" | Adolescents and adults with depression, anxiety and eating disorders, among other internalizing conditions, have a higher tendency to seek feedback from others[110,11], particularly when negative, and engage in more reassurance seeking[35]. Further, compulsivity in social media checking behaviours has been associated with anxiety, depression and problematic smartphone use[112], therefore becoming a cause of clinical and developmental concern[76]. Consistent with these results, online feedback seeking has been associated with depressive symptoms in adolescent community samples[54]. This association holds even after accounting for the effects of the overall frequency of technology use, offline excessive reassurance seeking and prior depressive symptoms. | H1.1d: adolescents with any mental health condition will be more likely to monitor online feedback than adolescents without a condition.<br>H2.0d: adolescents with externalizing mental health conditions will not differ in monitoring online feedback from adolescents without a condition.<br>H2.1d: adolescents with internalizing mental health conditions will be more likely to monitor online feedback than adolescents without a condition.<br>H3.1d: adolescents with internalizing mental health conditions will be more likely to monitor online feedback than adolescents with externalizing conditions. | Hames et al.[111]<br>Gillet et al.[55]<br>Elhai et al.[112]<br>Barry et al.[76]<br>Nesi & Prinstein[54]<br>Clerkin et al.[110] |

**Table 2 (continued) | Review of key social media and mental health literature used to formulate the hypotheses of our study**

| Social media use | Scientific literature | Relevant hypotheses | Key references |
|---|---|---|---|
| **Perceived impact of online feedback** Item: "The amount of likes, comments and shares I get on social media has an impact on my mood" | Compared with people without a mental health condition, those with internalizing conditions (for example, depression[113]) tend to have biased perceptions and recall of interpersonal feedback. This bias is apparent in both their symptoms, such as negative beliefs about the self and worry, as well as neural markers, such as feedback- and error-related negativity[56,57]. Social media platforms offer continuous opportunities for exposure to feedback from peers and strangers. Further, feedback is often more quantifiable and permanent than when received offline (for example, the number of likes received and comments to old posts can be revisited). We therefore expect the biased perception of social feedback demonstrated by some clinical groups to occur and possibly be heightened in online contexts[54]. | H1.1e: adolescents with any mental health condition will be more likely to feel impacted by online feedback than adolescents without a condition. H2.0e: adolescents with externalizing mental health conditions will not differ in feeling impacted by online feedback from adolescents without a condition. H2.1e: adolescents with internalizing mental health conditions will be more likely to feel impacted by online feedback than adolescents without a condition. H3.1e: adolescents with internalizing mental health conditions will be more likely to feel impacted by online feedback than adolescents with externalizing conditions. | Gotlib[113] Tobias & Ito[56] Tucker et al.[57] Nesi & Prinstein[54] |
| **Online friendship** Item: "I am happy with the number of friends I have on social media" | In offline contexts, social connections are a protective factor against long-term adverse physical and emotional outcomes, particularly during adolescence[114]. In line with this evidence, young people with both internalizing and externalizing mental health conditions report difficulties with peers, having few friends and wanting to be alone[58–60]. Further, adolescents with internalizing conditions tend to be unhappy with their social status and adolescents with externalizing conditions tend to have trouble making and keeping friends, both online and offline[73,75]. | H1.2f: adolescents with any mental health condition will be less happy about the number of online friends than adolescents without a condition. H2.2f: adolescents with internalizing mental health condition will be less happy about the number of online friends they have than adolescents without a condition. H2.4f: adolescents with externalizing mental health conditions will be less happy about the number of online friends they have than adolescents without a condition. H3.0f: adolescents with internalizing mental health conditions will be as happy about the number of online friends they have as adolescents with externalizing conditions. | Viner et al.[114] Asselmann et al.[58] McBride & Preyde[59] Sibley et al.[60] Bagwell et al.[75] Dawson et al.[73] |
| **Honest online self-disclosure** Item: "I can be honest with people on social media sites and apps about how I am feeling" | Self-disclosure is a communication process by which one person reveals information about themselves to another[115]. The extent of self-disclosure has been associated with higher relationship quality, intimacy and well-being in offline contexts. In adolescent psychiatric inpatients, low levels of self-disclosure have been linked to suicidality, with anxiety and depression mediating this association[61]. The quality of self-disclosure also differs in people with low compared with high psychological distress. For instance, people that are less distressed tend to disclose more positive information whereas those high in distress tend to disclose more negative and less honest information[62]. This effect appears to also occur on social media platforms. For instance, depressed individuals are more likely to post darker, bluer and greyer images than people without depressive symptoms[116]. Communicating personal and emotional information increases the risk of embarrassment and rejection[117], which is compounded by adolescents' increased sensitivity to peer feedback and anxiety regarding negative social evaluations[118]. We expect this to be especially difficult for adolescents with internalizing conditions, as they must balance the rewards associated with self-disclosure[119] with considerations of how that disclosure might be received by their peers. | H1.2g: adolescents with any mental health condition will engage in less online honest self-disclosure than adolescents without a condition. H2.0g: adolescents with externalizing mental health conditions will not differ in online honest self-disclosure from adolescents without a condition. H2.2g: adolescents with internalizing mental health conditions will engage in less online honest self-disclosure than adolescents without a condition. H3.2g: adolescents with internalizing mental health conditions will engage in less online honest self-disclosure than adolescents with externalizing conditions. | Sprecher & Hendrick[115] Horesh & Apter[61] Chen[62] Reece & Danforth[116] Omarzu[117] Van den bos[118] Vijayakumar et al.[119] |
| **Authentic online self-presentation** Item: "My social media profile is a true reflection of myself" | Different internalizing conditions such as anxiety, depression and eating disorders are characterized by the internalization of an ideal self that, once compared with perceptions of the actual self, results in negative self-evaluations. Adolescents with these conditions therefore create perfectionistic self-presentations to combat negative self-narratives and project desirable images of themselves in the mind of others[63,64,68–70]. We expect this process to also occur online, given the multiple affordances offered by social media platforms to curate one's image, such as deleting old posts, editing new posts and optimizing messages before sending. Hence, adolescents with a mental health condition, particularly if an internalizing condition, might be more likely to engage in impression management online to compensate for negative self-evaluations[62]. | H1.2h: adolescents with any mental health condition will engage in less authentic online self-presentation than adolescents without a condition. H2.0h: adolescents with externalizing mental health conditions will not differ in authentic online self-presentation from adolescents without a condition. H2.2h: adolescents with internalizing mental health condition will engage in less authentic online self-presentation than adolescents without a condition. H3.2h: adolescents with internalizing mental health conditions will engage in less authentic online self-presentation than adolescents with externalizing conditions. | Flett & Hewitt[63] Jain & Sudhir[64] O'Connor et al.[70] Sassaroli et al.[68] Calvo et al.[69] Chen[62] |

The social media items are reported in bold for clarity.

**Table 3 | Descriptive information by mental health condition**

| Mental health category | N | Age mean | Age s.d. | Sex proportion (male) | SES* |
|---|---|---|---|---|---|
| No condition | 2,821 | 14.71 | 2.48 | 0.50 | 0.42 |
| Any condition | 519 | 15.10 | 2.45 | 0.47 | 0.41 |
| Externalizing | 104 | 14.27 | 2.04 | 0.72 | 0.45 |
| Internalizing | 282 | 15.94 | 2.35 | 0.28 | 0.40 |
| Other | 76 | 14.00 | 2.59 | 0.80 | 0.36 |
| Between comorbidity | 57 | 13.93 | 1.80 | 0.51 | 0.46 |

*The proportion of participants in the fourth quartile (the most deprived category based on the Index of Multiple Deprivation). SES, socioeconomic status.

and clinical practice, as well as the design of targeted interventions and policies to improve children's and adolescents' mental health.

## Results

### Sample description
Our final sample included 3,340 young people (Table 3) aged 11–19 (mean 14.77, s.d. 2.48) years. The sample was 50% male and 50% female, 16% of participants had at least one mental health condition ($N = 519$), 8% had an internalizing condition ($N = 282$) and 3% had an externalizing condition ($N = 104$). Descriptive statistics split by social media items, sex and diagnostic groups are reported in Supplementary Tables 5–8 and Supplementary Figs. 3–5. Further, an overview of our hypotheses and results is provided in Supplementary Table 9.

### Any mental health condition versus no condition
We first compared adolescents with versus without mental health conditions, irrespective of condition type (Fig. 1, H1). In line with our hypothesis, we found that adolescents with any mental health condition reported spending more time on social media than adolescents without a condition (H1.1a; $g = 0.46$ (90% confidence interval (CI) 0.38 to 0.54); NHST (null hypothesis significance testing): $\beta = 0.87$, s.e.m. of 0.08, $t = 10.48$, $P = 2.745 \times 10^{-25}$; EQV (equivalence testing): $t(633.8) = 1.21$, $P = 0.916$). Specifically, differences were positive, statistically significant and non-equivalent. That is, we could not reject differences as large or larger than our preregistered equivalence bounds (smallest effect size of interest (SESOI), $g = 0.4$), indicating potentially meaningful differences between adolescents with and without a condition. On the contrary, the results did not support our hypothesis for online social comparison (H1.1b; $g = 0.30$ (90% CI 0.22 to 0.39); NHST: $\beta = 0.42$, s.e.m. of 0.07, $t = 6.52$, $P = 7.973 \times 10^{-11}$; EQV: $t(641.08) = -1.89$, $P = 0.026$), lack of control over time spent online (H1.1c; $g = 0.27$ (90% CI 0.19 to 0.35); NHST: $\beta = 0.39$, s.e.m. of 0.07, $t = 5.52$, $P = 3.614 \times 10^{-8}$; EQV: $t(674.42) = -2.68$, $P = 0.002$) and impact of online feedback on mood (H1.1e; $g = 0.29$ (90% CI 0.21 to 0.38); NHST: $\beta = 0.38$, s.e.m. of 0.06, $t = 6.36$, $P = 2.269 \times 10^{-10}$; EQV: $t(608.98) = -2.06$, $P = 0.016$). For these dimensions of social media engagement, effect sizes were positive, statistically significant but equivalent (that is, they fell within the preregistered SESOIs). This indicates a difference that is too small to be theoretically meaningful between adolescents with and without a mental health condition. Last, for monitoring of online feedback, we found differences that were not statistically significant and also equivalent, indicating no statistically nor theoretically meaningful differences between adolescents with versus without a condition (H1.1d; $g = 0.08$ (90% CI −0.01 to 0.15) NHST: $\beta = 0.11$, s.e.m. of 0.07, $t = 1.55$, $P = 0.121$; EQV: $t(653.24) = -6.47$, $P = 0.000$).

We further hypothesized that adolescents with any mental health condition would score lower than adolescents without a condition on dimensions of social media use that could incur mental health benefits, namely, happiness about the number of online friendships (H1.2f), honest self-disclosure (H1.2g) and authentic self-presentation

(H1.2h). In line with our hypothesis, we found lower happiness about the number of online friendships (H1.2f; $g = -0.37$ (90% CI −0.45 to −0.29); NHST: $\beta = -0.33$, s.e.m. of 0.04, $t = -8.23$, $P = 2.660 \times 10^{-16}$; EQV: $t(590.4) = 0.56$, $P = 0.277$), for which the effect size was negative, statistically significant and non-equivalent (that is, large enough to be potentially meaningful). In contrast, we did not find differences in honest self-disclosure (H1.2g; $g = -0.30$ (90% CI −0.38 to −0.22); NHST: $\beta = -0.39$, s.e.m. of 0.06, $t = -6.37$, $P = 2.213 \times 10^{-10}$; EQV: $t(629.39) = 1.931$, $P = 0.028$) and authentic self-presentation (H1.2h; $g = -0.19$ (90% CI −0.28 to −0.11); NHST: $\beta = -0.24$, s.e.m. of 0.06, $t = -3.98$, $P = 7.085 \times 10^{-5}$; EQV: $t(624.53) = 4.04$, $P = 0.000$). In these cases, effect sizes were negative and statistically significant but equivalent, suggesting that differences between those with and without a mental health condition were too small to be theoretically meaningful.

### Internalizing/externalizing conditions versus no condition
Our second question concerned the extent to which adolescents with internalizing or externalizing conditions differed in their social media use from adolescents without a condition (Fig. 1, H2).

**Internalizing versus no condition.** Our hypotheses were grounded in the mental health literature, which suggests that anxiety and depressive disorders are characterized by negative self-views, rumination, worries and social withdrawal. We expected these symptoms to be mirrored in adolescents' online experiences. The results supported our hypotheses for time spent on social media (H2.1a; $g = 0.62$ (90% CI 0.51 to 0.73); NHST: $\beta = 1.12$, s.e.m. of 0.11, $t = 10.32$, $P = 1.609 \times 10^{-25}$; EQV: $t(317.39) = 3.21$, $P = 0.999$), online social comparison (H2.1b; $g = 0.54$ (90% CI 0.43 to 0.65); NHST: $\beta = 0.76$, s.e.m. of 0.08, $t = 9.12$, $P = 1.304 \times 10^{-19}$; EQV: $t(318.57) = 2.11$, $P = 0.994$) and the impact of online feedback (H2.1e; $g = 0.38$ (90% CI 0.27 to 0.49); NHST: $\beta = 0.51$, s.e.m. of 0.08, $t = 6.61$, $P = 4.494 \times 10^{-11}$; EQV: $t(306.67) = -0.27$, $P = 0.385$), where we found positive, statistically significant, and non-equivalent effect sizes, suggesting potentially meaningful differences between adolescents with internalizing versus no condition. In contrast, the results did not support our hypothesis for monitoring of online feedback, where differences were not statistically significant and were also too small to be considered meaningful (H2.1d; $g = 0.13$ (90% CI 0.03 to 0.25); NHST: $\beta = 0.20$, s.e.m. of 0.09, $t = 2.13$, $P = 0.033$; EQV: $t(324.47) = -4.13$, $P = 0.000$).

For those with internalizing conditions, we also hypothesized decreased levels of happiness about the number of online friendships (H2.2f), honest self-disclosure (H2.2g) and authentic self-presentation (H2.2h). Our hypotheses were confirmed for happiness about the number of online friendships (H2.2f; $g = -0.45$ (90% CI −0.55 to −0.35); NHST: $\beta = -0.40$, s.e.m. of 0.05, $t = -7.91$, $P = 3.49 \times 10^{-15}$; EQV: $t(304.45) = -0.69$, $P = 0.776$) and honest self-disclosure (H2.2g; $g = -0.31$ (90% CI −0.42 to −0.20); NHST: $\beta = -0.41$, s.e.m. of 0.08, $t = -5.16$, $P = 2.670 \times 10^{-7}$; EQV: $t(314.2) = 1.32$, $P = 0.088$), where we found negative, statistically significant and potentially meaningful differences. In other words, those with internalizing conditions scored lower than adolescents with no condition. In contrast, we did not find support for meaningful differences in authentic self-presentation (H2.2h; $g = -0.19$ (90% CI −0.30 to −0.08); NHST: $\beta = -0.25$, s.e.m. of 0.08, $t = -3.16$, $P = 0.002$; EQV: $t(310.64) = 3.08$, $P = 0.500 \times 10^{-4}$), where the effect size was statistically significant but equivalent, and therefore too small to be considered meaningful. Last, we expected no differences in lack of control over time spent online for adolescents with internalizing versus no condition (H2.0c). The results did not support our hypothesis, showing positive, statistically significant and potentially meaningful differences ($g = 0.43$ (90% CI 0.33 to 0.55); NHST: $\beta = 0.60$, s.e.m. of 0.09, $t = 6.74$, $P = 1.91 \times 10^{-11}$; EQV: $t(336.39) = 0.489$, $P = 0.689$).

**Externalizing versus no condition.** Externalizing conditions are characterized by impulsivity, low self-monitoring, and risk taking.

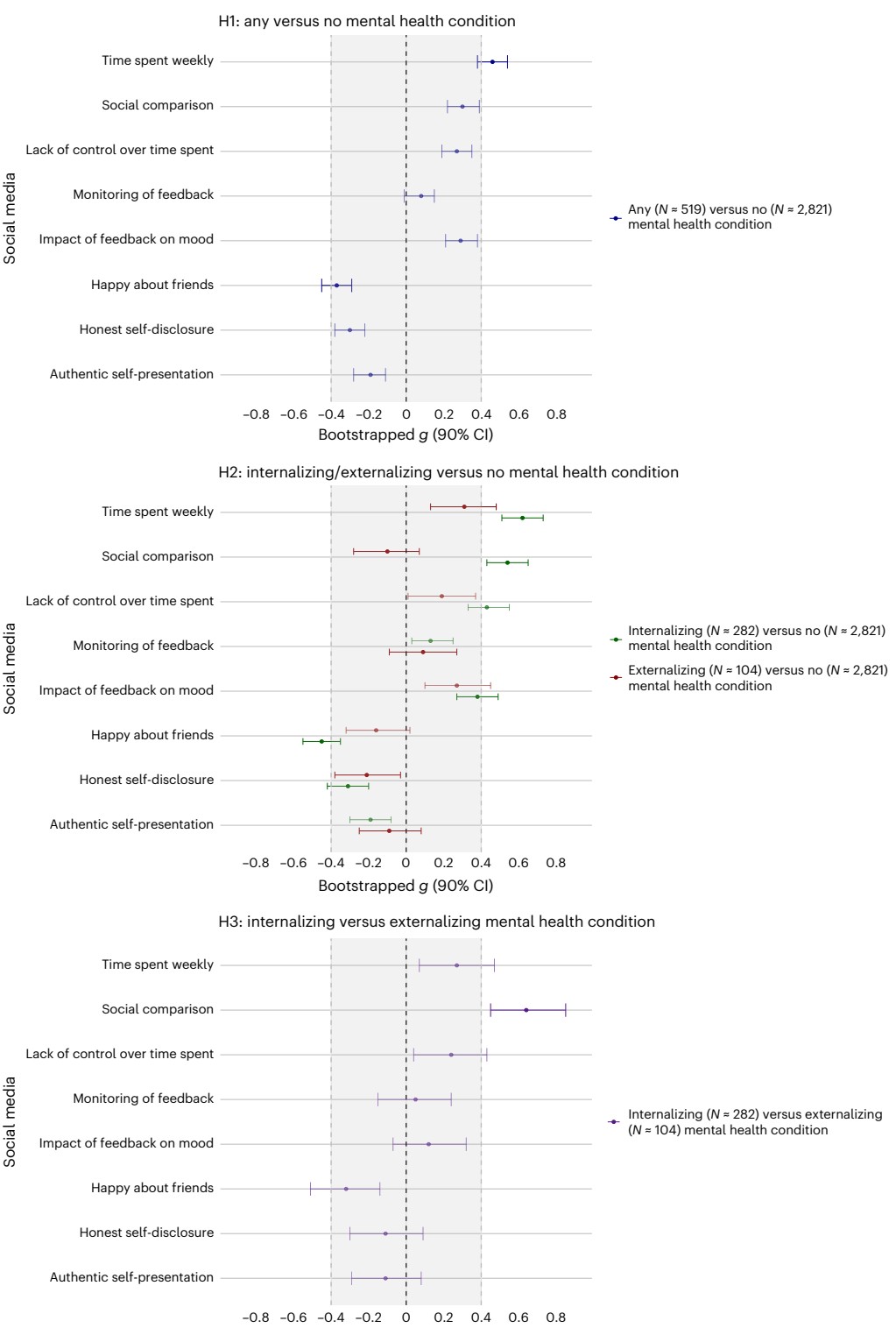

**Fig. 1 | Differences in social media use for the three group comparisons.**
Top: hypothesis 1 (H1, any mental health condition versus no condition).
Middle: hypothesis 2 (H2, internalizing/externalizing versus no condition).
Bottom: hypothesis 3 (H3, internalizing versus externalizing condition).
Data are presented as mean differences based on Hedges's $g$ effect size ($g$)
and its corresponding 90% CI. The shaded area indicates the SESOI ($g = 0.4$,
corresponding to $d = 0.4$). If the 90% CI lies completely within the SESOI, we
concluded equivalence, and therefore no meaningful differences. Bolded effect
sizes reflect comparisons that supported our hypotheses, while faded effect
sizes reflect comparisons that did not support our hypotheses. The starting
sample size includes social media users ($N = 519$ for adolescents with any mental
health condition, $N = 282$ for internalizing conditions, $N = 104$ for externalizing
conditions and $N = 2,821$ for no mental health condition). However, the exact
sample sizes for each comparison are reported in Supplementary Tables 5 and
6 and differ for each dimension of social media use, given that we planned to
analyse those separately.

We expected such symptoms to be reflected in how social media is used by this group. The results supported our hypotheses for time spent on social media (H2.3a; $g$ = 0.31 (90% CI 0.13 to 0.48); NHST: $\beta$ = 0.58, s.e.m. of 0.17, $t$ = 3.40, $P$ = 0.001; EQV: $t$(105.73) = −0.83, $P$ = 0.232), where we found positive differences that were statistically significant and large enough to be theoretically meaningful. The results did not support our hypotheses for the lack of control over time spent online (H2.3c; $g$ = 0.19 (90% CI 0.01 to 0.37); NHST: $\beta$ = 0.27, s.e.m. of 0.15, $t$ = 1.83, $P$ = 0.068; EQV: $t$(100.39) = −2.01, $P$ = 0.017) and happiness about the number of online friendships (H2.4f; $g$ = −0.16 (90% CI −0.32 to 0.02); NHST: $\beta$ = −0.12, s.e.m. of 0.09, $t$ = −1.39, $P$ = 0.165; EQV: $t$(96.65) = 2.34, $P$ = 0.023), where differences were not statistically significant and were too small to be meaningful.

Last, we expected no differences between adolescents with externalizing and no conditions in online social comparison (H2.0b), monitoring of online feedback (H2.0d), feeling impacted by online feedback (H2.0e), honest online self-disclosure (H2.0g) and authentic self-presentation (H2.0h). The results supported our hypotheses for online social comparison (H2.0b; $g$ = −0.10 (90% CI −0.28 to 0.07); NHST: $\beta$ = −0.13, s.e.m. of 0.14, $t$ = −0.97, $P$ = 0.331; EQV: $t$(104.81) = 2.94, $P$ = 0.000), monitoring of feedback (H2.0d; $g$ = 0.09 (90% CI −0.08 to 0.27), NHST: $\beta$ = 0.13, s.e.m. of 0.16, $t$ = 0.82, $P$ = 0.410; EQV: $t$(97.07) = −2.92, $P$ = 0.004), honest self-disclosure (H2.0g; $g$ = −0.21 (90% CI −0.38 to −0.03), NHST: $\beta$ = −0.26, s.e.m. of 0.13, $t$ = −2.01, $P$ = 0.045; EQV: $t$(98.82) = 1.79, $P$ = 0.040), and authentic self-presentation (H2.0h; $g$ = −0.09 (90% CI −0.25 to 0.08), NHST: $\beta$ = −0.11, s.e.m. of 0.13, $t$ = −0.85, $P$ = 0.395; EQV: $t$(98.29) = 2.96, $P$ = 0.003), as these effect sizes were not statistically significant and were also too small to be considered meaningful. In contrast, the results did not support our hypothesis for the impact of feedback on mood (H2.0e; $g$ = 0.27 (90% CI 0.10 to 0.45), NHST: $\beta$ = 0.34, s.e.m. of 0.13, $t$ = 2.71, $P$ = 0.007; EQV: $t$(96.78) = −1.14, $P$ = 0.120), where we found positive, significant and potentially meaningful differences.

### Internalizing versus externalizing conditions

Our third question focused only on adolescents with a mental health condition and specifically examined whether those with an internalizing condition use social media differently than those with an externalizing condition (Fig. 1, H3).

The results supported our hypotheses for online social comparison (H3.1b; $g$ = 0.64 (90% CI 0.45 to 0.85); NHST: $\beta$ = 0.89, s.e.m. of 0.16, $t$ = 5.75, $P$ = 0.000, EQV: $t$(203.72) = 2.187, $P$ = 0.993), where we found positive differences that were statistically significant and large enough to be theoretically meaningful in adolescents with internalizing compared with externalizing conditions. In contrast, the results did not support our hypotheses for the monitoring of online feedback (H3.1d; $g$ = 0.05 (90% CI −0.15 to 0.24); NHST: $\beta$ = 0.07, s.e.m. of 0.18, $t$ = 0.40, $P$ = 0.689, EQV: $t$(157.17) = −2.924, $P$ = 0.002) and impact of online feedback on mood (H3.1e; $g$ = 0.12 (90% CI −0.07 to 0.32); NHST: $\beta$ = 0.16, s.e.m. of 0.14, $t$ = 1.12, $P$ = 0.261, EQV: $t$(171.21) = −2.421, $P$ = 0.008), where differences were not statistically significant and were too small to be meaningful.

We further hypothesized that adolescents with internalizing conditions would score lower than adolescents with externalizing conditions in lack of control over time spent online (H3.2c), honest self-disclosure (H3.2g) and authentic self-presentation (H3.2h). However, we found neither significant nor meaningful differences across these dimensions (H3.2g; $g$ = −0.11 (90% CI −0.30 to 0.09); NHST: $\beta$ = −0.15, s.e.m. of 0.15, $t$ = −0.99, $P$ = 0.323, EQV: $t$(170.73) = 2.47, $P$ = 0.004; H3.2h; $g$ = −0.11 (90% CI −0.29 to 0.08); NHST: $\beta$ = −0.14, s.e.m. of 0.15, $t$ = −0.93, $P$ = 0.351, EQV: $t$(185.78) = 2.51, $P$ = 0.005). Further, the results were inconclusive for lack of control over time spent online (H3.2c; $g$ = 0.24 (90% CI 0.04 to 0.43); NHST: $\beta$ = 0.33, s.e.m. of 0.17, $t$ = 1.98, $P$ = 0.048, EQV: $t$(156.91) = −1.37, $P$ = 0.080), where we did not find statistically significant differences nor we could reject meaningfully large effect sizes.

Last, we hypothesized that adolescents with internalizing conditions would not differ from adolescents with externalizing conditions in time spent on social media (H3.0a) and happiness in the number of online friendships (H3.0f). The results did not confirm our hypotheses. For time spent on social media, we found positive differences (internalizing higher than externalizing) that were statistically significant and potentially meaningful (H3.0a; $g$ = 0.27 (90% CI 0.07 to 0.47); NHST: $\beta$ = 0.54, s.e.m. of 0.191, $t$ = 2.76, $P$ = 0.006, EQV: $t$(171.33) = −1.15, $P$ = 0.1125). Further, we found that adolescents with internalizing conditions reported lower happiness about the number of online friends than adolescents with externalizing conditions. In this case, differences were negative, statistically significant and potentially meaningful (H3.0f; $g$ = −0.32 (90% CI −0.51 to −0.14); NHST: $\beta$ = −0.29, s.e.m. of 0.10, $t$ = −2.93, $P$ = 0.003, EQV: $t$(206.5) = 0.707, $P$ = 0.223).

### Exploratory and sensitivity analyses

To extend our findings, we conducted four sets of sensitivity analyses. First, we included adolescents with between-group comorbidities in question 2, such as any internalizing condition with a comorbid externalizing condition, or vice versa, and compared them with those without a condition (Supplementary Tables 15 and 16). Second, we examined the association between mental health severity, conceptualized as the number of conditions (irrespective of diagnostic type), and social media use (Supplementary Table 17). Third, we focused on adolescents with specific conditions and compared their social media use with that of adolescents without a condition. In this case, we only tested conditions for which we were sufficiently powered, namely, major depressive disorder ($N \approx 86$; Supplementary Table 18), generalized anxiety disorder ($N \approx 75$; Supplementary Table 19) and social anxiety disorder ($N \approx 70$; Supplementary Table 20). Last, we tested our hypotheses for time spent on social media separately for school days and weekends, rather than using a composite score (Supplementary Table 21).

Overall, the results were largely in line with our primary findings with a few exceptions. Namely, we found that adolescents with externalizing and between-group comorbidity (Supplementary Table 16) reported less honest self-disclosure than those without a condition. For the sensitivity analysis on mental health severity, we found that the number of conditions, irrespective of type, was associated with time spent on social media, social comparison, monitoring of feedback and impact of feedback on mood (Supplementary Table 17). Last, we did not find differences for time spent on weekdays versus weekends/holidays (Supplementary Table 21). We note that these sensitivity analyses were exploratory and conducted on relatively small sample sizes, which limits the robustness of these findings.

## Discussion

In this Registered Report, we analysed differences in social media use between adolescents with and without mental health conditions in a UK sample of over 3,000 participants. Overall, we found significant and meaningful differences across both quantitative (time spent) and qualitative (for example, online social comparison and happiness about the number of online friends) dimensions of social media use.

Interestingly, the largest difference in social media use between those with and without mental health conditions was in the time spent on social media, with the former reporting higher usage. It is important to note that this measure was self-reported, which is known to have only a moderate correlation with objective measures such as sensing data[77]. This raises the question of whether those with mental health conditions perceive that they spend more time on social media or whether they actually do so. Further, we observed that adolescents with a mental health condition reported lower satisfaction with the number of their online friends. In offline contexts, social connections serve as a protective factor against long-term adverse physical and emotional outcomes, especially during adolescence. Our findings therefore suggest that the difficulties with peer relationships experienced by youth

clinical groups offline may also be reflected in their online interactions. For the other dimensions of social media use examined, our results followed the hypothesized direction and were statistically significant (with the exception of monitoring of online feedback). However, the differences were not large enough to be considered meaningful. This might be explained by the relatively high threshold set as our SESOI, which was a moderate effect size grounded in literature on sleep and physical exercise, both established markers of psychopathology.

We next compared social media use between adolescents with a specific mental health condition (internalizing or externalizing) versus no condition. In this case, the results largely supported our hypotheses, whereby adolescents with internalizing conditions demonstrated higher time spent on social media, increased social comparison, greater impact of social media feedback on mood, lower satisfaction with the number of online friends and lower honest self-disclosure compared to those without a mental health condition. Unexpectedly, we also found that adolescents with internalizing conditions reported a higher lack of control over their time spent online, an engagement dimension that we had instead hypothesized would be more pronounced in those with externalizing conditions. For adolescents with externalizing conditions, the only meaningful difference when compared with those with no condition was increased time spent online, with no notable differences across other dimensions of social media use. These results might be explained by the fact that the dimensions of social media engagement used in this study were largely framed around internal experiences (that is, they enquired about one's emotions and thoughts) that could be more effective indicators of internalizing rather than externalizing conditions.

Finally, by limiting our focus to only adolescents with mental health conditions, we compared those with internalizing to those with externalizing conditions. We found that adolescents with internalizing conditions engaged in higher online social comparison. This finding aligns with existing research indicating that adolescents with depressive and anxious symptomatology tend to unfavourably compare themselves with others on social media. Given that social media platforms provide continuous and concrete opportunities for social comparison—such as browsing others' profiles without initiating social interaction[52]—social comparison may represent a critical mechanism associated with internalizing symptoms on social media. Further, adolescents with internalizing conditions were less happy about the number of their online friends. This may be because their tendency towards negative self-evaluation and social comparison leads them to make negative evaluations of their social status[78]. In contrast, adolescents with externalizing conditions might be less focused on social comparison and more on immediate social interactions, resulting in greater satisfaction with their online friendships.

Altogether, this study has three key strengths. First, the complex survey design, which utilized random probability sampling, produced a nationally representative UK sample. Second, all participants in the sample underwent a standardized multi-informant assessment by professional clinical raters. This method provided comprehensive information about participants' mental health without only relying on self-reported questionnaires, self-diagnoses or mental health service use, measures that only capture a subset of individuals with mental health conditions[79,80]. Last, we examined both qualitative and quantitative aspects of social media use, offering insights beyond time spent into relevant engagement dimensions such as online social comparison and the impact of feedback on mood.

We also acknowledge some limitations of our study. First, regarding study design, we analysed raw associations from cross-sectional data. Therefore, no causal or directional inference can be drawn from these findings, including whether the onset of mental health conditions affects the examined dimensions of social media use or vice versa. Further, given the relatively small sample size in our externalizing group ($N = 104$) and the relatively large number of individuals with

between-group comorbidities ($N = 57$), our findings concerning externalizing conditions should be interpreted with caution. Specifically, our question 2 and 3 results exclude adolescents with a relatively common clinical profile of comorbid externalizing and anxiety disorders. Additional work on how young people with this clinical profile use and experience social media is needed.

Further, although the sample was nationally representative, we cannot determine the extent to which our findings apply outside the UK. Given the socio-cultural factors influencing social media use and mental health conditions, replicating these results in other regions across both the Global North and Global South is crucial if generalizations are to be made[25]. We also acknowledge that the data were collected in 2017. While the examined dimensions of social media use remain relevant, the rapid evolution of platforms and user behaviours presents a potential limitation when applying our findings to current trends. Last, we acknowledge limitations regarding our measures. Specifically, we relied on self-reported social media data. Self-reports capture participants' perceptions, such as online social comparison, that cannot be objectively or reliably measured otherwise. However, they often fail to accurately reflect actual patterns in usage, particularly when estimating time spent on social media[77].

Future research should aim to replicate and expand on these findings in three key areas. First, studies using experimental and longitudinal designs are essential to clarify the temporal and causal dynamics linking various social media patterns to mental health conditions. Doing so would allow us to disentangle the within-person variation and directional relationship between social media use and mental health symptom presentation, onset or recovery. Second, regarding social media use, future studies could investigate other self-reported and clinically relevant aspects of engagement, such as time displacement from offline activities[81], the similarity between perceived self on social media and offline[82], as well as goal-directed social media use. Additionally, studies could assess differences in objective social media use, such as time spent on various apps, posting behaviours, active messaging and content exposure. Third, research involving adolescents with intellectual and learning disabilities is necessary to identify differences in this specific clinical group, which was not included in this study.

The results have implications for clinical practice. Specifically, we find key aspects of social media engagement that could inform the creation of guidelines for patient consultations and early intervention strategies. For example, this could include psychoeducation and cognitive-behavioural reappraisal techniques specifically aimed at online social comparison or the impact of social media feedback (for example, 'likes') on mood for adolescents with internalizing conditions.

Over the past years, there has been increasing concern that social media is negatively impacting young people's mental health, but very little research has compared social media use in those with and without mental health conditions[26]. In one of the first studies of its kind, we find that young people with mental health conditions report engaging with social media in different ways from those without a condition. This highlights aspects of social media use that might present an increased risk to this already vulnerable group and provides a window for future research to ensure that the digital world is safe for all children regardless of mental health status.

## Methods

### Ethics information

The MHCYP 2017 survey was reviewed and approved by the West London and GTAC Research Ethics Committee (reference: 16/LO/0155) and the Health Research Authority Confidentiality Advisory Group (reference: 16/CAG/0016) in 2016. Both parents and children provided consent to take part in data collection and were compensated with a £10 voucher for their time. Parents of children under 16 years were interviewed first and permission was sought to interview their child afterwards; the child then provided assent. Conversely, 17–19-year olds were

directly asked for their consent, with permission subsequently sought for their parents to be interviewed. Access to the data was granted to the research team by NHS Digital (DARS-NIC-424336-T7K7T-v0.6 r).

## Design

The MHCYP study is one of a series of national surveys on the mental health of children and young people in England administered in 1999, 2004, 2017, 2021 and 2022. In this Registered Report, we analysed the 2017 wave collected between January and October 2017: the most recent wave to be made available to researchers as well as the first wave to collect comprehensive data on adolescents' social media use and to include 17–19-year olds. We only analysed data from adolescents who reported being social media users aged 11–19 years, a total of 3,340 participants (50% male and 50% female) out of the full sample of 9,117. The survey was collected using a stratified probability sample of children and young people living in England who were registered with a general practitioner[36]. Data were collected via face-to-face interviews with adolescents and their parents. At the same time, if the family agreed, questionnaires were mailed to teachers (for the available data and key demographics see MHCYP 2017[36,83]).

**Measures.** *Time spent on social media.* The study measured time spent on social media using two questionnaire items: "When you use social media sites or apps, how much time do you spend using them on a typical school day?" (SMTimeSpentS) and "When you use social media sites or apps how much time in total do you spend using them on a typical weekend or holiday day?" (SMTimeSpentW). Participants answered both questions on a nine-point Likert scale: 1, less than 30 min; 2, more than 30 min but less than an hour; 3, 1–2 h; 4, 2–3 h; 5, 3–4 h; 6, 4–5 h; 7, 5–6 h; 8, 6–7 h; 9, more than 7 h. A single variable reflecting average social media hours was created from these two variables (SMTimeSpent; see Supplementary Table 1 for more details). To do so, we first calculated the mean time in hours for each response. For example, if a participant responded "one to two hours", we recoded this as 1.5 h. Participants that responded "more than seven hours" were recoded to 7.5 h, while participants that responded "less than 30 min" were recorded to 15 min (that is, 0.25 h). Weekday hours were then multiplied by 5, and weekend hours were multiplied by 2, and the products were summed and divided by 7 to establish a daily mean social media use variable, measured in hours. The SMTimeSpent variable was coded as continuous[84].

In the survey, the questions regarding time spent on social media for both weekdays (SMTimeSpentS) and weekends (SMTimeSpentW) were only asked of adolescents who responded that they use social media sites daily or on most days on a previous questionnaire item (SMFreqofUse; 1, daily or most days). Hence, participants that reported a lower frequency of social media use (that is, SMFreqofUse; 2, a few times a week; 3, once a week; 4, a few times a month; 5, once a month; 6, less often than once a month) were not asked these questions. To handle the resulting missing data in the SMTimeSpent variable, we coded any adolescents who responded to the SMFreqofUse question that they use social media between "a few times a week" and "once a week" to 45 min (0.75 h) and adolescents that responded "a few times a month" and "less often than once a month" to 15 min (0.25 h) on the SMTimeSpent question (for more information about our approach to missing data, see Supplementary Table 2).

*Social media engagement.* We analysed seven qualitative dimensions of social media engagement measured with questionnaire items tapping into experiences associated with both risks and benefits to adolescent mental health. All measures (summarized with related literature in Table 2) were developed in consultation with a young person advisory group (see the Supplementary Methods for more information), where children and adolescents defined the dimensions of social media use most relevant to them. The measures related to

mental health risks encompassed online social comparison ("I compare myself to others on social media"), lack of control over time spent online ("I spend more time on social media than I mean to"), monitoring of online feedback ("I monitor the amount of likes, comments and shares I get on social media") and the impact of online feedback ("The amount of likes, comments and shares I get on social media has an impact on my mood"). On the contrary, the measures indicative of mental health benefits included online friendship ("I am happy with the number of friends I have on social media"), honest self-disclosure ("I can be honest with people on social media sites and apps about how I am feeling") and authentic self-presentation ("My social media profile is a true reflection of myself").

Participants responded to these measures using a five-point Likert scale (1, disagree a lot; 2, disagree a little; 3, neither agree nor disagree; 4, agree a little; 5, agree a lot). We omitted the "Don't know" responses and coded 1–5 responses as continuous, given research suggesting that five-point continuous classifications perform as well as or occasionally better than categorical classifications[84]. We performed a sensitivity analysis to test whether examining social media use on weekdays versus weekends/holidays separately, rather than as a weighted average, changed the main results.

*Mental health conditions.* Face-to-face interviewers completed the Development and Wellbeing Assessment (DAWBA)[85] with parents and adolescents aged 11 years or over to establish mental health conditions. During the interview, participants were first led through the 25-item Strengths and Difficulties Questionnaire[86] (Supplementary Table 3). Second, the interviewer administered the DAWBA, a diagnostic tool shown to have good validity[87] and reliability[88]. The DAWBA uses structured and semi-structured questions to assess the presence and severity of symptoms for a wide range of DSM-5 or ICD-10 mental health disorders (Table 1). Each module starts with a few screening items, which, if answered negatively (indicative of the lack of symptoms), allow the interview to proceed to the next module with no loss of accuracy[87] (for example, Supplementary Table 4). There is one exception: if a participant scores highly on the Strengths and Difficulties Questionnaire, the interviewer is directed to ask in-depth internalizing disorder DAWBA modules, even if participants screened negative on the initial questions.

After the initial screening items, the subsequent structured items in the DAWBA modules relate directly to diagnostic criteria in DSM-5 and ICD-10. They are close-ended questions about specific mental health symptoms (for example, "In the last 4 weeks, have there been times when you have been very sad, miserable, unhappy or tearful?"). If a participant responds positively to these structured items, they are subsequently asked open-ended questions about these problems (for example, "Please describe your mood—sadness or irritability—and your level of interest in things"). During the assessment, interviewers transcribe open-ended responses verbatim and are also able to add personal comments beneath each response.

A team of clinical raters assessed the DAWBA's structured and qualitative information from all informants to decide whether an adolescent showed evidence of a DSM-5 or ICD-10 mental health condition that would warrant clinical treatment (Supplementary Figs. 1 and 2). To determine the presence of a condition, clinical raters (1) checked that the answers to structured comments were understood by the participants accurately; (2) interpreted any conflicts between child, parent and teacher responses and decided which assessment to prioritize; and (3) identified clinically impairing disorders that did not perfectly fit current operationalized diagnostic criteria or "not otherwise specified diagnosis" such as "other anxiety disorder".

Overall, we coded information on mental health conditions into two separate variables: (1) a binary variable indicating the presence of any mental health condition (diagnosis or no diagnosis) and (2) a categorical variable for the type of condition (internalizing diagnosis,

externalizing diagnosis or no diagnosis). We subdivided conditions identified via the diagnostic assessment into internalizing and externalizing using existing classifications of psychiatric diagnoses[34,65] (Table 1). This distinction draws from transdiagnostic research showing that different conditions (for example, anxiety, depression and eating disorders) are often comorbid, share underlying core symptoms[89] and can therefore be grouped to reflect clinical presentations with more validity[34]. Further, when we had enough power, we ran additional exploratory analyses examining responses to all social media questions (a–g) separately for individual conditions (specifically, major depressive disorder, generalized anxiety disorder and social anxiety disorder).

Comorbidity data were coded into two separate variables: (1) a binary variable indicating within-group comorbidity (any two diagnoses of either internalizing or externalizing: yes or no, for descriptive purposes only) and (2) a binary variable indicating internalizing–externalizing between-group comorbidity (any comorbid internalizing and externalizing diagnoses: yes or no). Individuals who showed between-group comorbidity were removed before the analysis of questions 2 and 3, given our goal to compare social media use between these groups. To increase the clinical utility of this work, we ran a sensitivity analysis for question 2, including people with between-group comorbidity.

### Analysis plan

We conducted all statistical analyses in R version 4.3.1 (R Core Team, 2021), testing the association between time- and engagement-based measures of social media use and mental health conditions using equivalence tests and linear regression models[90,91]. To control for the type I error rate across multiple tests, we set a corrected alpha level of 0.0125, accounting for the false discovery rate across our four tested hypotheses for any given social media item[92]. Our analytical approach was based on regressions rather than analysis of variance, as the former allow for more diverse predictors, unbalanced groups and inclusion of covariates in potential exploratory analyses[93]. Below, we describe the statistical analyses we used to test our questions and hypotheses (see the Supplementary Information for more details). Further, the analysis code is available on OSF[94].

### Questions and hypotheses

**Question 1: investigate whether adolescents with any mental health condition use social media differently than those without a condition.** To address question 1, we tested the association between social media use and mental health conditions. Specifically, we estimated linear regression models with mental health condition as a binary predictor (two levels: diagnosis versus no diagnosis) and social media use as a continuous outcome. The no diagnosis group was set as the reference level for these analyses. Below, we report our null and directional hypotheses, first for dimensions of social media use expected to reflect mental health risks and second for dimensions expected to reflect mental health benefits in adolescents with versus without a condition.

H0(1.0): adolescents with any mental health condition will not differ from adolescents without a condition in (a) time spent on social media, (b) online social comparison, (c) lacking control over time spent online, (d) monitoring of online feedback, (e) feeling impacted by online feedback, (f) happiness about the number of online friendships, (g) honest online self-disclosure and (h) authentic self-presentation online.

H1(1.1): adolescents with any mental health condition will score higher than adolescents without a condition in (a) time on social media, (b) online social comparison, (c) lacking control over time spent online, (d) monitoring of online feedback and (e) feeling impacted by online feedback.

H1(1.2): adolescents with any mental health condition will score lower than adolescents without a condition in (f) happiness about the number of online friendships, (g) honest online self-disclosure and (h) authentic self-presentation online.

To examine whether social media use varies with mental health severity, we conducted sensitivity analyses to test for a linear effect of the number of diagnoses on the social media responses using linear regression models.

**Question 2: investigate whether adolescents with an internalizing or externalizing condition use social media differently than those without a condition.** After assessing differences in social media use in adolescents with versus without any mental health condition, we examined whether adolescents with internalizing or externalizing conditions use social media differently than adolescents without a condition. Hence, we conducted linear regression models with diagnostic category as a categorical predictor (three levels: internalizing diagnosis, externalizing diagnosis and no diagnosis) and social media use as a continuous outcome. To test our hypotheses, we examined comparisons between two levels of the diagnostic category variable, with no diagnosis set as the reference level. For hypotheses 2.0[i], 2.1 and 2.2, we reported regression coefficients for internalizing versus no diagnosis, while for hypotheses 2.0[e], 2.3 and 2.4 we reported coefficients for externalizing versus no diagnosis. The null hypotheses marked by 'e' or 'i' indicate that, for the considered comparison, the null hypothesis was our primary hypothesis. Hence, for those dimensions of social media use, we expected no difference between adolescents with internalizing or externalizing conditions and those without a condition. We used 'e' to indicate primary null hypotheses related to externalizing versus no condition and 'i' to indicate our primary null hypotheses related to internalising condition.

H0(2.0): adolescents with internalizing or externalizing conditions will not differ from adolescents without a condition in (a) time on social media, (b) online social comparison[e], (c) lacking control over time spent online[i], (d) monitoring of online feedback[e], (e) feeling impacted by online feedback[e], (f) happiness about the number of online friendships, (g) honest online self-disclosure[e] and (h) authentic self-presentation online[e].

H1(2.1): adolescents with internalizing condition will score higher than adolescents without a condition in (a) time on social media, (b) online social comparison, (d) monitoring of online feedback and (e) feeling impacted by online feedback.

H1(2.2): adolescents with internalizing conditions will score lower than adolescents without a condition in (f) happiness about the number of online friendships, (g) honest online self-disclosure and (h) authentic self-presentation online.

H1(2.3): adolescents with externalizing conditions will score higher than adolescents without a condition in (a) time spent on social media and (c) lack of control over time spent online.

H1(2.4): adolescents with externalizing conditions will score lower than adolescents without a condition in (f) happiness about the number of online friendships.

**Question 3: investigate whether adolescents with an internalizing mental health condition use social media differently than those with an externalizing condition.** To address question 3, we examined how adolescents with internalizing conditions differed in social media engagement compared to adolescents with externalizing conditions. To this aim, we compared the internalizing and externalizing

levels of the diagnostic category variable described in question 2, with internalizing as the reference level. Also, in this case, the null hypotheses marked by 'c' indicate our primary hypotheses. Hence, for those dimensions of social media use, we expected no difference between adolescents with internalizing and externalizing conditions.

H0 (3.0): adolescents with internalizing conditions will not differ from adolescents with externalizing conditions in (a) time on social media[c], (b) online social comparison, (c) lacking control over time spent online, (d) monitoring of online feedback, (e) feeling impacted by online feedback, (f) happiness about the number of online friendships[c], (g) online self-disclosure and (h) authentic self-presentation online.

H1(3.1): adolescents with internalizing conditions will score higher than adolescents with externalizing conditions in (b) online social comparison, (d) monitoring of online feedback and (e) feeling impacted by online feedback.

H1(3.2): adolescents with internalizing conditions will score lower than adolescents with externalizing conditions in (c) lack of control over time spent online, (g) online self-disclosure and (h) authentic self-presentation online.

For all regression analyses, we treated the eight dimensions of social media use, including both time- and engagement-based measures, as separate outcomes predicted by information on mental health conditions as the only regressor. While it is common in research to use statistical control to remove confounding effects from a regression coefficient, appropriate control variables should be identified only after justifying a causal structure that includes the outcome, predictors and all theorized confounders. When the selected control variables are inappropriate or remain unjustified, controlling can result in biased regression estimates[95]. Further, recent literature warns against controlling for demographic factors such as sex without thought and instead prompts researchers to interrogate how this variable intersects with the predictors and outcomes under investigation[96]. In the present work, treating sex or age as a confounding variable would mean ignoring the possibility that there are meaningful sex or age differences in the examined relationships. As our goal is to investigate the overall association between social media use and mental health, we provided a descriptive account of the age and sex of adolescents included in each tested model rather than control for these demographics. For example, in question 1, age and sex of adolescents with any condition versus without a condition are reported for descriptive purposes (Table 3).

## Equivalence testing

Given that null hypothesis significance testing does not allow a comprehensive interpretation of statistically non-significant results[97], each model was complemented by equivalence tests[97] using the bootstrapped two one-sided tests (TOST) with the 'boot_T_TOST' function from the TOSTER package[91] to quantify support for the null hypothesis (H0(1–3)). This involves assessing whether the 90% CIs for the effect size lie inside prespecified equivalence bounds that indicate the SESOI. If the CIs lie inside the equivalence bounds, the effect size is interpreted as negligible. On the contrary, if the CIs spread outside the equivalence bounds, the effect size is considered meaningful in size. In this case, the 90% CIs are used rather than 95% because the effect size is tested against two equivalence bounds separately (that is, the upper and lower bound), reflecting $(1 - 2\alpha) \times 100\%$ (see the Supplementary Information for more details).

We established the equivalence bounds by identifying a theoretically meaningful SESOI (Cohen's $d$ = 0.4; refer to the Supplementary Methods for a detailed explanation). After an extensive scoping exercise to identify a suitable theoretical foundation, we determined that

the most relevant benchmark for our research questions are everyday behaviours linked to mental health, such as sleep and physical activity. These behaviours, much like social media use, are a regular part of daily routines but, unlike social media use, they are well-established markers of psychopathology, based on both theory[98–100] and matched empirical evidence[101,102]. Consequently, if the actual effect size of social media use is comparable to that observed for behaviours such as sleep and physical activity, we can confidently conclude that social media use also represents an everyday behaviour that exhibits meaningful group-level differences between clinical and non-clinical populations. The interpretation and analysis plan are presented separately for each hypothesis in the Supplementary Information.

Overall, we inferred support for the null hypotheses (that is, no meaningful difference in social media use between groups) if the 90% CI for this association lies within the equivalence bounds. Of note, while a theoretical SESOI based on everyday behaviours and their link to mental health served as our primary effect size of interest to allow for a clear confirmatory approach, we also identified secondary SESOIs based on the effect sizes that are practically and clinically meaningful. These secondary SESOIs play an important role in supporting the interpretation of our findings (as detailed in Supplementary Methods), ensuring the applicability of our study to both academic and practical domains.

Overall, we inferred support for the alternative hypotheses if (1) the coefficient for the association of social media use and our grouping variable for mental health diagnosis was significant, (2) the association followed the hypothesised direction and (3) the CIs for the association did not fall within the equivalence bounds. The interpretation and analysis plan are presented separately for each hypothesis in the Supplementary Information. Further, we detail the planned exploratory analyses in the Supplementary Methods.

Given that not all assumptions of linear models were met (Supplementary Table 9), we complemented our preregistered parametric analyses with exploratory nonparametric tests for all hypotheses (Supplementary Tables 10 and 11). We implemented the Brunner Munzel test based on the 'brunner_munzel' function for NHST and the 'simple_htest' function for equivalence testing from the TOSTER package in R[91]. This test, also known as the generalized Wilcoxon test, is a nonparametric test of stochastic equality between two samples that tests against the null hypothesis that for randomly selected values $X$ (that is, social media score in the adolescents without a condition) and $Y$ (that is, social media score in the adolescents with a condition), the probability of $X$ being greater than $Y$ is equal to the probability of $Y$ being greater than $X$. We documented all discrepancies (20% overall) between the parametric and nonparametric test results in Supplementary Tables 12–14.

## Sampling plan

**Inclusion criteria and sample size.** We included all participants in the MHCYP 2017 survey aged between 11 and 19 years at the time of assessment who have answered any of the social media use questions with anything other than "don't know". Only individuals with diagnostic information, including the absence of a diagnosis, were included in the MHCYP dataset. Since we examined each social media use question separately, we included participants who answered at least one of the seven social media engagement questions or the question about time spent on social media. No specific documentation was available regarding the response rates for the social media questions. Given that the questionnaires were administered through interviews with professionals, we expected minimal missing data for the social media responses[103]. Hence, using a conservative estimate, we assumed a 5% missingness for our power calculations.

To address question 1, all adolescents with and without a condition were included in our analysis. Hence, in question 1, we included (1) participants with conditions other than internalizing or externalizing disorders (for example, autism spectrum disorder, tic disorder

and psychotic disorders; Table 2); (2) participants with within-group comorbidity (multiple internalizing or externalizing diagnoses; for example, comorbid depressive and social anxiety disorder); and (c) participants with between-group comorbidity (both internalizing and externalizing conditions; for example, comorbid depressive and attention deficit hyperactivity disorder).

In contrast, for questions 2 and 3, we included (1) participants with internalizing or externalizing conditions only and (2) participants with within-group comorbidity (multiple internalizing or externalizing diagnoses). Hence, participants with other conditions or between-group comorbidity were excluded from the main analysis, given our goal to compare social media use in adolescents with externalizing and internalizing conditions. However, we ran sensitivity analyses to test the impact of including participants with between-group comorbidities in question 2.

On the basis of the summary demographics[83], after accounting for potential missing data, we estimated the sample of complete cases to be around $N = 3,854$ (accounting for 5% missingness in the sample of $N = 4,057$, 11–19-year olds), approximately 15% of whom received at least one mental health diagnosis ($N = 577$). Available documentation[83] suggests that approximately 19.2% of individuals with internalizing conditions have at least one comorbid externalizing condition, and approximately 28% of individuals with externalizing diagnoses have at least one comorbid internalizing diagnosis. This suggests approximately $N = 370$ with internalizing-only diagnoses and $N = 199$ with externalizing-only diagnoses.

**Power calculations.** We calculated the power by setting the SESOI ($d = 0.4$, refer to Supplementary Methods for a detailed explanation), the alpha level to 0.05 and the estimated sample size (estimated $N = 3,854$ based on existing MHCYP documentation). For the equivalence tests, power was calculated using the TOSTER package in R[91,104]. For the regression models, power was determined using the pwr package[105] in R. The code for these calculations is available on the OSF[94].

H0(1). We calculated the power for equivalence testing to detect a SESOI of $d = 0.4$ given at least $N \geq 577$ individuals with a condition and $N \geq 3,277$ individuals without a condition. The results indicate 100% power to reject the presence of effects that are larger than $d = 0.4$.

H1(1). We calculated the power to detect a statistical effect of condition on social media responses using linear regression. The results indicate 100% power to detect the SESOI ($d = 0.4$) with at least $N \geq 577$ individuals with a condition and $N \geq 3,277$ individuals without a condition.

H0(2). We calculated the power for equivalence testing to detect a SESOI of $d = 0.4$ given at least $N \geq 370$ individuals with an internalizing only condition and $N \geq 3,277$ with no condition. The results indicate 100% power to reject the presence of effects that are larger than $d = 0.4$. We calculated the power for equivalence testing to detect a SESOI of $d = 0.4$ given at least $N > 199$ individuals with an externalizing only condition and $N > 3,277$ with no condition. The results indicate 100% power to reject the presence of effects that are larger than $d = 0.4$.

H1(2). We calculated the power to detect a statistical effect of internalizing condition type (internalizing versus no condition) on social media responses using linear regression. The results indicate 100% power to detect the SESOI ($d = 0.4$) with at least $N \geq 370$ individuals with an internalizing-only condition and $N \geq 3,277$ with no condition. We calculated power to detect an effect of externalizing diagnosis type (that is, externalizing versus no condition) on social media responses using linear regression. The results indicate 100% power to detect the SESOI ($d = 0.4$) with at least $N \geq 199$ individuals with an externalizing-only condition and $N \geq 3,277$ with no condition.

H0(3). We calculated the power for equivalence testing to detect a SESOI of $d = 0.4$ given at least $N \geq 370$ individuals with internalizing only and $N \geq 199$ with externalizing only conditions. The results indicate 96% power to reject the presence of effects that are larger than $d = 0.4$.

H1(3). We calculated the power to detect a statistical effect of internalizing-only versus externalizing-only condition type on social media responses. The results indicate 99.8% power to the SESOI ($d = 0.4$) with at least $N \geq 370$ individuals with internalizing only and $N \geq 199$ with externalizing only conditions.

In addition to our a priori power calculations, we conducted power sensitivity analyses with the final sample size[105]. That is, we determined the smallest effect size that is observable with 95% power, given the corrected alpha of 0.0125 and our final sample size (Supplementary Table 22).

### Reporting summary

Further information on research design is available in the Nature Portfolio Reporting Summary linked to this article.

## Data availability

The MHCYP dataset is held on behalf of NHS Digital by the UK Data Service. Restrictions apply to the availability of this data for privacy and ethical reasons, which were used under license for this study. Data access can be requested by applying to the Data Access Request Service (DARS; number: DARS-NIC-424336-T7K7T-v0.6). Researchers interested in accessing the data can find further information via the DARS website at https://digital.nhs.uk/services/data-access-request-service-dars/dars-guidance. Source data are provided with this paper.

## Code availability

The analysis code can be found on the OSF[94].

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

## Acknowledgements

The UK Medical Research Council DTP PhD programme (RG86932) funded L.F. The Jacobs Foundation (CERES SUAI/084 G114119), the UK Medical Research Council (MC_UU_00030/13) and a UKRI Future Leaders Fellowship (MR/X034925/1) funded A.M.F., L.F. and A.O. The Huo Family Foundation, and the ESRC (ES/Y010736/1 and ES/T008709/1) funded A.K.P. Both A.K.P. and A.O. were supported by the Economic and Social Research Council (ES/T008709/1). T.J.F. is supported by the National Institute for Health and Care Research (NIHR204413, NIHR153625 and NIHR202025), the Swedish Research Council for Health Working Life and Welfare (2022-01002_Forte), and the Medical Research Council (MC_PC_20052). All research at the Department of Psychiatry in the University of Cambridge is supported by the NIHR Cambridge Biomedical Research Centre (NIHR203312) and the NIHR Applied Research Collaboration East of England. The views expressed are those of the author(s) and not necessarily those of the NIHR or the Department of Health and Social Care. T.J.F.'s research group receives funding from Place2Be, a third sector organisation that provides mental health training and intervention to UK schools. The funders had no role in study design, data collection and analysis, decision to publish or preparation of the manuscript. The MHCYP 2017 survey was funded by the Department of Health and Social Care, commissioned by NHS Digital, and carried out by the National Centre for Social Research, the Office for National Statistics and Youthinmind. We are very grateful to all the adolescents and families who took part in the study, the personnel for their help in recruiting them and the whole NHS Digital team that includes interviewers, technicians, research scientists, volunteers, managers, receptionists, nurses and the clinical raters. A special thank you to D. Lakens, M. Vuorre and A.R. Caldwell for their valuable advice on the statistical analyses.

## Author contributions

We present author contributions according to the CRediT (Contributor Roles Taxonomy). L.F.: conceptualization, methodology, formal analysis, writing— original draft, writing—review and editing. A.M.F.: conceptualization, methodology, formal analysis, writing—review and editing. A.K.P.: methodology, writing—review and editing, supervision. T.J.F.: writing—review and editing, supervision, resources, project administration. A.O.: conceptualization, methodology, writing— original draft, writing—review and editing, supervision. Of note, none of the authors that conceptualized the analysis previously accessed the MHCYP dataset. For the purpose of open access, the author has applied a Creative Commons Attribution (CC BY) licence to any Author Accepted Manuscript version arising from this submission.

## Competing interests

The authors declare no competing interests.

## Additional information

**Correspondence and requests for materials** should be addressed to Luisa Fassi.

**Protocol registration** The Stage 1 protocol for this Registered Report was accepted in principle on 7 December 2023. The protocol, as accepted by the journal, is available via Figshare at https://figshare.com/s/730e3b0d4da82e9b6a46.

# Reporting Summary

## Statistics

For all statistical analyses, confirm that the following items are present in the figure legend, table legend, main text, or Methods section.

| n/a | Confirmed | |
|---|---|---|
| ☐ | ☒ | The exact sample size (*n*) for each experimental group/condition, given as a discrete number and unit of measurement |
| ☐ | ☒ | A statement on whether measurements were taken from distinct samples or whether the same sample was measured repeatedly |
| ☐ | ☒ | The statistical test(s) used AND whether they are one- or two-sided<br>*Only common tests should be described solely by name; describe more complex techniques in the Methods section.* |
| ☐ | ☒ | A description of all covariates tested |
| ☐ | ☒ | A description of any assumptions or corrections, such as tests of normality and adjustment for multiple comparisons |
| ☐ | ☒ | A full description of the statistical parameters including central tendency (e.g. means) or other basic estimates (e.g. regression coefficient) AND variation (e.g. standard deviation) or associated estimates of uncertainty (e.g. confidence intervals) |
| ☐ | ☒ | For null hypothesis testing, the test statistic (e.g. *F*, *t*, *r*) with confidence intervals, effect sizes, degrees of freedom and *P* value noted<br>*Give P values as exact values whenever suitable.* |
| ☒ | ☐ | For Bayesian analysis, information on the choice of priors and Markov chain Monte Carlo settings |
| ☒ | ☐ | For hierarchical and complex designs, identification of the appropriate level for tests and full reporting of outcomes |
| ☐ | ☒ | Estimates of effect sizes (e.g. Cohen's *d*, Pearson's *r*), indicating how they were calculated |

*Our web collection on statistics for biologists contains articles on many of the points above.*

## Software and code

Policy information about availability of computer code

| Data collection | This study uses secondary data, details on how to access the data are reported in the Data Availability Statement. Specifically, the MHCYP dataset is held on behalf of NHS Digital by the UK Data Service. Restrictions apply to the availability of this data for privacy and ethical reasons, which were used under license for this study. Data access can be requested by applying to the Data Access Request Service (DARS; number for this study: DARS-NIC-424336-T7K7T-v0.6). Researchers interested in accessing the data can find further information on the DARS website (see https://digital.nhs.uk/services/data-access-request-service-dars/dars-guidance). |
|---|---|
| Data analysis | The analysis code used to run the power analysis (stage 1) and all other analyses (stage 2) can be found at: https://osf.io/s2dwu/?view_only=5acced2ddb884f9481d439a7746f4dd1. Details on each code script are reported in the READ.ME file.<br><br>We used R version 4.4.0 (2024-04-24), and the following packages: Cairo (version: 1.6-2), readstata13 (version: 0.10.1), dplyr (version 1.1.2), TOSTER (version: 0.8.0), ggplot2 (version 3.5.1), patchwork (version: 1.2.0.9000), tidyverse (version: 2.0.0). |

For manuscripts utilizing custom algorithms or software that are central to the research but not yet described in published literature, software must be made available to editors and reviewers. We strongly encourage code deposition in a community repository (e.g. GitHub). See the Nature Portfolio guidelines for submitting code & software for further information.

# Data

Policy information about availability of data
All manuscripts must include a data availability statement. This statement should provide the following information, where applicable:
- Accession codes, unique identifiers, or web links for publicly available datasets
- A description of any restrictions on data availability
- For clinical datasets or third party data, please ensure that the statement adheres to our policy

The MHCYP dataset is held on behalf of NHS Digital by the UK Data Service. Restrictions apply to the availability of this data for privacy and ethical reasons, which were used under license for this study. Data access can be requested by applying to the Data Access Request Service (DARS; number: DARS-NIC-424336-T7K7T-v0.6). Researchers interested in accessing the data can find further information on the DARS website (i.e., https://digital.nhs.uk/services/data-access-request-service-dars/dars-guidance).

# Research involving human participants, their data, or biological material

Policy information about studies with human participants or human data. See also policy information about sex, gender (identity/presentation), and sexual orientation and race, ethnicity and racism.

| | |
|---|---|
| Reporting on sex and gender | Our final sample included 3,340 young people aged 11 to 19 (age: mean = 14.77, sd = 2.48). The sample was 50% male and 50% female. Both sex and age were measured using self-report. We provide descriptives for age and sex separately for each group in Table 3 and the section 'population characteristics' below. We also provide separate descriptive statistics separately for males and females (sex variable) for each social media item in the Supplementary Results (Supplementary Table 5-8). We do not provide separate descriptive statistics for age as this question falls outside the scope of this study. |
| Reporting on race, ethnicity, or other socially relevant groupings | While it is common in research to use statistical control to remove confounding effects from a regression coefficient, appropriate control variables should be identified only after justifying a causal structure that includes the outcome, exposure and all theorised confounders. When the selected control variables are inappropriate or remain unjustified, controlling can result in biased regression estimates. Further, recent literature warns against controlling for demographic factors such as sex without thought and instead prompts researchers to interrogate how this variable intersects with the exposure and outcomes under investigation. In the present work, treating sex or age as a confounding variable would mean ignoring the possibility that there are meaningful sex or age differences in the examined relationships. As our goal is to investigate the association between social media use and mental health diagnosis, we provided a descriptive account of the age and sex of adolescents in each group. As shown below, age and sex of adolescents with vs without a mental health condition are reported for descriptive purposes. |
| Population characteristics | The age and sex of each examined group are reported in Table 3 in the main manuscript and below:<br><br>- No mental health (N = 2821, age mean = 14.71, male proportion: 0.50)<br>- Any mental health condition (N = 519, age mean = 15.10, male proportion = 0.47)<br>- Externalising condition(N = 104, age mean = 14.27, male proportion = 0.72)<br>- Internalising condition (N = 282, age mean = 15.94, male proportion = 0.80)<br>- Other conditions (N = 76, age mean = 14, male proportion = 0.80)<br>- Comorbidity between internalising and externalising condition (N = 57, age mean = 13.93, male proportion = 0.51). |
| Recruitment | The MHCYP study is one of a series of national surveys on the mental health of children and young people in England administered in 1999, 2004, 2017, 2021 and 2022. In this Registered Report, we analysed the 2017 wave collected between January and October 2017: the most recent wave to be made available to researchers as well as the first wave to collect comprehensive data on adolescents' social media use and to include 17 to 19-year-olds. We only analysed data from adolescents who reported being social media users aged 11-19 years, a total of 3,340 participants (50 % male, 50% female) out of the full sample of 9,117. The survey was collected using a stratified probability sample of children and young people living in England who were registered with a general practitioner. Data was collected via face-to-face interviews with adolescents and their parents. At the same time, if the family agreed, questionnaires were mailed to teachers (for the available data and key demographics see MHCYP 2017: https://digital.nhs.uk/data-and-information/publications/statistical/mental-health-of-children-and-young-people-in-england/2017/2017). |
| Ethics oversight | The MHCYP 2017 survey was reviewed and approved by the West London & GTAC Research Ethics Committee (REC reference: 16/LO/0155) and the Health Research Authority Confidentiality Advisory Group (CAG reference: 16/CAG/0016) in 2016. Both parents and children provided consent to take part in data collection and were compensated with a 10£ voucher for their time. Parents of children under 16 years were interviewed first, and permission was sought to interview their child afterwards; the child then provided assent. Conversely, 17–19-year-olds were directly asked for their consent, with permission subsequently sought for their parents to be interviewed. |

Note that full information on the approval of the study protocol must also be provided in the manuscript.

# Field-specific reporting

Please select the one below that is the best fit for your research. If you are not sure, read the appropriate sections before making your selection.

☐ Life sciences   ☒ Behavioural & social sciences   ☐ Ecological, evolutionary & environmental sciences

# Behavioural & social sciences study design

All studies must disclose on these points even when the disclosure is negative.

| | |
|---|---|
| Study description | This study analyses data from a quantitative cross-sectional survey collected in 2017 (Mental Health of Children and Young People). |
| Research sample | For the 2017 survey, a stratified, multistage random probability sample of 18,029 children was drawn from the NHS Patient Register in October 2016. Children and young people were eligible to participate if they were aged 2 to 19, lived in England, and were registered with a GP. The sample was designed to be representative of the population of children and young people aged 2 to 19 living in England. The final sample consisted of 9,117 children and young people. For this study, we analyzed data from 3,340 participants. This subset was selected based on age (focusing on adolescents aged 9–11 years) and social media use (including only those who reported being social media users). |
| Sampling strategy | A stratified multistage random probability sample was used for the survey, involving a two-stage process. Full information on the sampling can be found at: https://files.digital.nhs.uk/60/1CB03A/MHCYP%202017%20Survey%20Design%20and%20Methods.pdf<br><br>To determine the sample size needed to answer our research questions, we run power analyses and power sensitivity analyses. A priori power calculations were conducted and reported in the Stage 1 registered report. For these calculations, we set were the smallest effect size of interest (SESOI, d = 0.4; see Supplementary Methods for details), an alpha level of 0.05, and an estimated sample size of N = 3,854. The code for these calculations is available on the Open Science Framework (OSF) at https://osf.io/s2dwu/?view_only=5acced2ddb884f9481d439a7746f4dd1. Results showed that we were sufficiently powered (> 95%) to detect our smallest effect size of interest for all research questions.<br><br>In addition to the a priori power calculations, we conducted power sensitivity analyses using the final sample size, which was known only after starting data analysis. In this case we determined the smallest effect size that could be detected with 95% power, given an alpha level of 0.0125 (corrected for multiple comparisons) and the final sample size (N = 3340). As reported in Supplementary Table 18, these analyses demonstrated power to detect our smallest effect size of interest (d = 0.4) across our research questions. |
| Data collection | Data were collected as part of a national survey. Researchers were therefore blind to the study conditions and hypotheses. All interviews were conducted individually, involving only the clinical rater and the interviewee (child, parent, or teacher) and were carried out either in person or in a computerized format.<br><br>For participants aged 11 to 16 years, the process began with an initial interview with the parent or legal guardian, followed by a separate interview with the child. Young people aged 17 to 19 were interviewed directly, with their parent also interviewed if both parties consented. Teachers of 5 to 16-year-olds were invited to complete an online or paper questionnaire if consent was provided.<br><br>Mental health assessments were conducted using the detailed and comprehensive Development and Well-Being Assessment (DAWBA; Goodman et al., 2000), which evaluates a range of mental health conditions, including emotional, hyperactivity, and behavioral disorders, as well as less common conditions like autism. After completing the interviews, trained clinical raters reviewed the data to assess the presence of mental health conditions for each participant. |
| Timing | Data collection occurred over nine months, from January to September 2017. For participants aged 11–16 years, data collection was conducted between January and June 2017 to maintain consistency with previous surveys in the series and to ensure that teacher questionnaires could be completed and returned before the end of the school summer term. For children aged 16 or younger, data collection began with an interview with the parent. Parental permission was then sought to interview the child. If consent was given, the child participated in an interview, which included a self-completion section for sensitive questions. For young people aged 17–19 years, agreement to participate was obtained directly from the individual. Further details on data collection can be found at: https://files.digital.nhs.uk/60/1CB03A/MHCYP%202017%20Survey%20Design%20and%20Methods.pdf |
| Data exclusions | In this study, we excluded participants aged 2–10 years (N= 5060) and those who reported not using social media (N = 717). Both these exclusions were pre-established (in our Stage 1 registered report) for two reasons: 1) children younger than 11 are in a different developmental time window (childhood/pre-adolescence), 2) we were interested in young people that used social media to answer our research questions. Each social media use item was analyzed independently, allowing us to retain data for individual items even when responses were incomplete for other items. For example, if a child answered the question about time spent on social media but not the question about online social comparison, their response to the time spent question was still included in the analysis. Consequently, the exact sample size varies for each question and is reported in Supplementary Tables 5–8. |
| Non-participation | No participant declined participation. |
| Randomization | We did not perform randomization nor we controlled for third variables. While it is common in research to use statistical control to remove confounding effects from a regression coefficient, appropriate control variables should be identified only after justifying a causal structure that includes the outcome, predictors, and all theorised confounders. When the selected control variables are inappropriate or remain unjustified, controlling can result in biased regression estimates. Further, recent literature warns against controlling for demographic factors such as sex without thought and instead prompts researchers to interrogate how this variable intersects with the predictors and outcomes under investigation. In the present work, treating sex or age as a confounding variable would mean ignoring the possibility that there are meaningful sex or age differences in the examined relationships. As our goal is to investigate the overall association between social media use and mental health diagnosis, we provided a descriptive account of the age and sex of adolescents included in each tested model rather than control for these demographics. |

# Reporting for specific materials, systems and methods

We require information from authors about some types of materials, experimental systems and methods used in many studies. Here, indicate whether each material, system or method listed is relevant to your study. If you are not sure if a list item applies to your research, read the appropriate section before selecting a response.

## Materials & experimental systems

| n/a | Involved in the study |
|---|---|
| ☒ | Antibodies |
| ☒ | Eukaryotic cell lines |
| ☒ | Palaeontology and archaeology |
| ☒ | Animals and other organisms |
| ☒ | Clinical data |
| ☒ | Dual use research of concern |
| ☒ | Plants |

## Methods

| n/a | Involved in the study |
|---|---|
| ☒ | ChIP-seq |
| ☒ | Flow cytometry |
| ☒ | MRI-based neuroimaging |

## Plants

| Seed stocks | NA |
|---|---|
| Novel plant genotypes | NA |
| Authentication | NA |

