## [Peer Review File · Nature Human Behaviour]

Social media use in adolescents with and without mental health conditions

Corresponding Author: Ms Luisa Fassi

Version 0:

Decision Letter:

15th March 2023

Dear Ms Fassi,

Thank you once again for your manuscript, entitled "Social media use in adolescent clinical and non-clinical populations," and for your patience during the peer review process.

Your manuscript has now been evaluated by 2 reviewers, whose comments are included at the end of this letter. Although the reviewers find your protocol to be of interest, they also raise some important concerns. We are very interested in the possibility of proceeding further with your submission in Nature Human Behaviour, but would like to consider your response to these concerns in the form of a revised manuscript before we make a decision on in principle acceptance and Stage 2 submission.

To guide the scope of the revisions, the editors discuss the referee reports in detail within the team, including with the chief editor, with a view to (1) identifying key priorities that should be addressed in revision and (2) overruling referee requests that are deemed beyond the scope of the current study. We hope that you will find the prioritised set of referee points to be useful when revising your study. Please do not hesitate to get in touch if you would like to discuss these issues further.

1. Please specifically address Reviewer 1's concerns about the meaningfulness of your classification of disorders for clinical practice. We ask that in your revised manuscript, you provide a new and more clinically-relevant classification of diagnoses.

2. We ask that you follow Reviewer 2's advice and consider changes to the design and the technical approach to generate meaningful robustness checks.

In sum, we invite you to revise your Stage 1 Registered Report taking into account reviewer and editor comments. Please highlight all changes in the manuscript text file.

* Include a "Response to reviewers" document detailing, point-by-point, how you addressed each referee comment. If no action was taken to address a point, you must provide a compelling argument. This response will be sent back to the reviewers along with the revised manuscript.

* Ensure that you use our template for Stage 1 Registered Reports to prepare your revised manuscript: https://www.nature.com/documents/NHB_Template_RR_Stage1.docx. Failure to ensure that your revised Stage 1 submission meets our requirements as specified in the template will result in your submission being returned to you, which will delay its consideration.

* In your cover letter, please include the following information:

--An anticipated timeline for completing the study if your Stage 1 submission is accepted in principle.

--A statement confirming that you agree to share your raw data, any digital study materials, computer code (if relevant), and laboratory log for all eventually published results.

--A statement confirming that, following Stage 1 in principle acceptance, you agree to register your approved protocol on the Open Science Framework (<https://osf.io/>) or other recognised repository, either publicly or under private embargo, until submission of the Stage 2 manuscript.

--A statement confirming that if you later withdraw your paper, you agree to the Journal publishing a short summary of the pre-registered study under a section Withdrawn Registrations.

Link Redacted

We hope to receive your revised manuscript within four to eight weeks. If you cannot send it within this time, please let us know. We will be happy to consider your revision so long as nothing similar has been accepted for publication at Nature Human Behaviour or published elsewhere.

Nature Human Behaviour is committed to improving transparency in authorship. As part of our efforts in this direction, we are now requesting that all authors identified as 'corresponding author' on published papers create and link their Open Researcher and Contributor Identifier (ORCID) with their account on the Manuscript Tracking System (MTS), prior to acceptance. ORCID helps the scientific community achieve unambiguous attribution of all scholarly contributions. You can create and link your ORCID from the home page of the MTS by clicking on 'Modify my Springer Nature account'. For more information please visit www.springernature.com/orcid.

Sincerely,

[Redacted]
[Redacted]
[Redacted]
Nature Human Behaviour

Reviewer expertise:

Reviewer #1: effects of social media use on the mental health of children and adolescents ; clinical psychology

Reviewer #2: effects of digital media ; open science

Reviewers' Comments:

Reviewer #1:

Remarks to the Author:

-Overall, this is a well-thought out, comprehensive study that once performed, will assuredly garner attention in the field. It is well-written, and the social media use of youth with psychiatric pathology is certainly an area where more attention needs to be paid from a research perspective.

However, I do have some significant concerns that may or not be able to be addressed in your proposed study design:

1) I think your diagnostic categorizations for internalizing vs externalizing disorders don't necessarily make sense from a clinical perspective. For example, complex post-traumatic stress disorder and mania (and therefore BPAD) are disorders with well-established externalizing features. I would think that lumping those diagnoses in with internalizing disorders would impact your findings.

2) Additionally, I don't know that it's a) beneficial from a clinical standpoint or b) accurate from a research standpoint to split diagnoses as internalizing vs externalizing, especially if you're not accounting for the presence youth with comorbid externalizing behaviors like substance use that won't get screened out as a "between-group comorbidity." The online behaviors of youth with a history of early childhood PTSD are generally quite different from youth with a generalized anxiety disorder diagnosis; I would recommend rethinking how you classify diagnoses in your questions 2 and 3 in a way that might be more clinically relevant e.g. PTSD, Anxiety disorders, Depressive disorders

Minor issues:

-There are some areas where the article and subsequent noun don't match from a grammar perspective (e.g. pg 16, "an internalising diagnoses.")

-I couldn't tell if you were asking about time spent on social media on weekends separately from on weekdays. Those are likely markedly different amounts of time and it may be hard for kids to provide a single answer if so. Moreover, weeknight social media use might actually be more interesting to examine given that it could give more insight into a youth's impulse control.

Reviewer #2:

Remarks to the Author:

The manuscript addresses a timely and highly relevant topic. The relationship between social media use and mental health among adolescents is an intensely debated topic in in academia as well as the public. Importantly, arguments made in this debate are often not backed by robust empirical evidence. This study can make a very important contribution in that regard. In the public

debate, but sometimes also in academic research, the lines between well-being and clinically relevant behaviors and outcomes are often blurred. Hence, one thing that I see as a particular strength of the study is the use of clinically validated measures (in particular, "standardized diagnostic assessments by professional clinical raters", p.24). Another very important issue that is addressed by this study is the lack of clinical populations in previous research. Overall, I believe that the study can make a very valuable contribution to the field as empirically sound and methodologically sophisticated and robust research is much needed in this contested area.

The study employs secondary use of high-quality data with a large enough sample size and, unlike many previous studies, considers quantitative as well as qualitative dimensions of social media use.

The hypotheses are precise, soundly formulated hypotheses, and convincingly derived from the existing literature. The distinction between different dimensions of social media use and between internalizing and externalizing conditions is particularly useful and interesting here. The manuscript provides detailed information about the sampling plan and data processing, both of which seem appropriate for testing the elicited hypotheses. The authors also present a suitable a priori power analysis for each group of hypotheses based on the expected sample sizes. The code for the power analysis as well as simulations of the hypothesis tests are provided and I was able to reproduce the reported outcomes of those on my computer. The software used for the analysis (and the version) are also reported, which is helpful for reproducing the analyses. The proposed analysis methods are simple yet appropriate for testing the hypotheses. I also found the appendix tables very helpful as summaries of the preregistration.

Overall, I only have a few remarks regarding how the authors might improve the eventual full paper:

I would say that the categorical measures of daily social media use may not be optimal. I am aware that the authors cannot change this as this is secondary data, but this should be discussed in the full paper. While the proposed processing of these data into a composite social media use score makes sense, it might make sense to alter some of the decisions made for this process to generate robustness checks for the results.

In general, as the authors are surely aware of, self-reports of media use can be biased (e.g., due to social desirability). Hence, I think this is imported to address and also discuss potential alternatives, such as tracking data or ambulatory/ecological momentary assessment methods.

While the general(ized) linear model underlies both of these options and I think the use of regression analysis is perfectly fine, the authors may want to at least briefly explain why they decided against using analysis of variance which is common in many study designs that compare groups on various outcome variables.

Regarding the sharing of materials: In addition to sharing scripts, I would suggest also using a dependency management solution like `renv` (<https://rstudio.github.io/renv/articles/renv.html>) or `rang` (<https://github.com/chainsawriot/rang>) plus `rocker` (<https://github.com/rocker-org/rocker>) to ensure that the package versions are the same for reproductions of the analyses.

Finally, what would be very interesting and valuable for a full paper, in my view, would be a detailed discussion of whether or in what way the results can be expected to generalize for other samples and esp. also other countries.

Version 1:

Decision Letter:

8th June 2023

Dear Ms Fassi,

Thank you once again for your manuscript, entitled "Social media use in adolescent clinical and non-clinical populations," and for your patience during the peer review process.

Your manuscript has now been evaluated by Reviewer 2 of the original review round, who is positive about the revision. Unfortunately, Reviewer 1 from the previous round was unable to re-review, so we recruited a new reviewer with expertise in social media use and clinical populations (Reviewer 3). Our reviewers' comments are included at the end of this letter. Although the reviewers find your protocol to be of interest, they also raise some important concerns. We are very interested in the possibility of proceeding further with your submission in *Nature Human Behaviour*, but would like to consider your response to these concerns in the form of a revised manuscript before we make a decision on in principle acceptance and Stage 2 submission.

Specifically, we ask that you address the remaining concerns of Reviewer 3 in full. Please highlight all changes in the manuscript text file.

* Include a "Response to reviewers" document detailing, point-by-point, how you addressed each referee comment. If no action was taken to address a point, you must provide a compelling argument. This response will be sent back to the reviewers along with the revised manuscript.

* Ensure that you use our template for Stage 1 Registered Reports to prepare your revised manuscript: https://www.nature.com/documents/NHB_Template_RR_Stage1.docx. Failure to ensure that your revised Stage 1 submission meets our requirements as specified in the template will result in your submission being returned to you, which will delay its consideration.

* In your cover letter, please include the following information:

--An anticipated timeline for completing the study if your Stage 1 submission is accepted in principle.

--A statement confirming that you agree to share your raw data, any digital study materials, computer code (if relevant), and

laboratory log for all eventually published results.

--A statement confirming that, following Stage 1 in principle acceptance, you agree to register your approved protocol on the Open Science Framework (<https://osf.io/>) or other recognised repository, either publicly or under private embargo, until submission of the Stage 2 manuscript.

--A statement confirming that if you later withdraw your paper, you agree to the Journal publishing a short summary of the pre-registered study under a section Withdrawn Registrations.

Link Redacted

We hope to receive your revised manuscript within four to six weeks. If you cannot send it within this time, please let us know. We will be happy to consider your revision so long as nothing similar has been accepted for publication at Nature Human Behaviour or published elsewhere.

Nature Human Behaviour is committed to improving transparency in authorship. As part of our efforts in this direction, we are now requesting that all authors identified as 'corresponding author' on published papers create and link their Open Researcher and Contributor Identifier (ORCID) with their account on the Manuscript Tracking System (MTS), prior to acceptance. ORCID helps the scientific community achieve unambiguous attribution of all scholarly contributions. You can create and link your ORCID from the home page of the MTS by clicking on 'Modify my Springer Nature account'. For more information please visit www.springernature.com/orcid.

Sincerely,

[Redacted]

[Redacted]

Nature Human Behaviour

Reviewer expertise:

Reviewer #2: effects of digital media ; open science

Reviewer #3: social media use ; clinical populations

Reviewers' Comments:

Reviewer #2:

Remarks to the Author:

As before, I enjoyed reading the manuscript and believe that it has the potential to make a very valuable contribution to the field. I also appreciate the detailed and thoughtful response letter. After reading the revised Stage 1 Registered Report as well as the response letter, I can say that the authors have adequately addressed all of my previous comments, and I look forward to reading the Stage 2 Registered Report.

Reviewer #3:

Remarks to the Author:

This manuscript addresses a timely topic of high public health importance through a carefully considered plan for evaluating how youth with mental health conditions use social media. Overall, the hypotheses are well stated, planned analyses are clear and seem aligned with the stated goals of the project. Sensitivity analyses seem appropriate to appropriately interrogate the data. Examining both quantitative and qualitative aspects of use, informed by a youth advisory board, are strengths of the planned approach. The delineation between internalizing & externalizing disorders, and disorders that were excluded were reasonable, and the transdiagnostic approach is likely to have wide-ranging utility. Some additional comments/suggested revisions are noted below, which may advance the quality and clarity of the manuscript.

1) Within the introduction, the background provides statistics on use that specifies older adolescents. Given that the age range for the study begins at age 11 offering statistics for younger adolescents seems necessary.

2) Greater context is needed in understanding social comparison hypotheses. Social comparison can have positive functions and be developmentally typical for youth. What specifically is the construct measured here, e.g., upward or downward social comparison, any form of social comparison that has carries negative affective consequences, etc.?

- 3) Given the limited evidence within clinical samples, it seems difficult to determine whether adolescents with mental health conditions will have greater social media use vs. statistically different social media use. For example, one study not cited here found lower usage was associated with suicidal ideation (see Hamilton et al, 2021). Other studies, evaluating social media data, have shown variability in use patterns among youth with different clinical characteristics with some groups showing trends toward blunted use (see Loveys et al, 2017 from the Fourth Workshop on Computational Linguistics and Clinical Psychology).
- 4) Greater context is needed around discussion of self-disclosure. While authentic disclosure is generally thought to be helpful toward mental health, youth with psychiatric disorders also navigate presenting their authentic selves with considerations of disclosing about their emotional state and mental health vulnerabilities, which could be met by negative evaluations or bullying. Authentic self-disclosure within this population is not clear cut, and it seems warranted to recognize this nuance.
- 5) It would help to understand more about the youth advisory board who aided in the development of qualitative measures of social media use. Were these youth who had psychiatric diagnoses, who could discuss the unique features of social media use for youth with mental health conditions?
- 6) The addition of supplementary analyses for individual diagnoses or diagnostic categories seems to be a sizable addition that warrants greater description within the main paper. Particularly given high rates of comorbidity, the analyses related to mental health severity that assess whether the presence of multiple diagnoses relate to social media use factors seems quite important.

Version 2:

Decision Letter:

4th August 2023

Dear Ms Fassi,

Thank you once again for your manuscript, entitled "Social media use in adolescent clinical and non-clinical populations," and for your patience during the peer review process.

Your revised manuscript has now been evaluated by original Reviewer 4, whose comments are included at the end of this letter. Although the reviewer finds your protocol to be of interest and signs off, we have some remaining editorial concerns about power, effect sizes and analytical choices, given that power cannot be known before the sample is accessed. We remain very interested in the possibility of proceeding further with your submission in Nature Human Behaviour, but would like to consider your response to these concerns in the form of a revised manuscript before we make a decision on in principle acceptance and Stage 2 submission.

Specifically, we would encourage you to seek out assistance from an expert with Bayesian statistics, which could be used to either supplement or replace the current sampling approach. A Bayesian approach seems much more suitable when the N is fixed and actual power cannot be calculated.

Attached to this decision letter, you can also find a marked-up copy of your manuscript. These edits are in order to make sure that your manuscript complies with our requirements for novelty and priority claims, as well as for qualitative characterizations of the work.

In sum, we invite you to revise your Stage 1 Registered Report taking into account reviewer and editor comments. Please highlight all changes in the manuscript text file.

* Ensure that you use our template for Stage 1 Registered Reports to prepare your revised manuscript: https://www.nature.com/documents/NHB_Template_RR_Stage1.docx. Failure to ensure that your revised Stage 1 submission meets our requirements as specified in the template will result in your submission being returned to you, which will delay its consideration.

* In your cover letter, please include the following information:

--An anticipated timeline for completing the study if your Stage 1 submission is accepted in principle.

--A statement confirming that you agree to share your raw data, any digital study materials, computer code (if relevant), and laboratory log for all eventually published results.

--A statement confirming that, following Stage 1 in principle acceptance, you agree to register your approved protocol on the Open Science Framework (<https://osf.io/>) or other recognised repository, either publicly or under private embargo, until submission of the Stage 2 manuscript.

--A statement confirming that if you later withdraw your paper, you agree to the Journal publishing a short summary of the pre-registered study under a section Withdrawn Registrations.

Link Redacted

We hope to receive your revised manuscript within four to eight weeks. If you cannot send it within this time, please let us know. We will be happy to consider your revision so long as nothing similar has been accepted for publication at Nature Human Behaviour or published elsewhere.

Nature Human Behaviour is committed to improving transparency in authorship. As part of our efforts in this direction, we are now requesting that all authors identified as 'corresponding author' on published papers create and link their Open Researcher and Contributor Identifier (ORCID) with their account on the Manuscript Tracking System (MTS), prior to acceptance. ORCID helps the scientific community achieve unambiguous attribution of all scholarly contributions. You can create and link your ORCID from the home page of the MTS by clicking on 'Modify my Springer Nature account'. For more information please visit www.springernature.com/orcid.

Sincerely,

[Redacted]

[Redacted] PhD

Nature Human Behaviour

Reviewers' Comments:

Reviewer #4:

Remarks to the Author:

I appreciated reviewing the paper once again, which is well written and addresses a very important area of research. The response to comments was thoughtful, thorough, and informative. Best wishes on your future research!

Version 3:

Decision Letter:

7th December 2023

Dear Ms Fassi,

Thank you once again for submitting your revised Stage 1 Registered Report, entitled "Social media use in adolescent clinical and non-clinical populations." Everything is in order and I am delighted to say that we can offer acceptance in principle. You may progress to Stage 2 and complete the study as approved.

Attached to this email, you find a checklist with some final formatting requests. Important points have been highlighted. Please carefully address these requests and return the final files to us within 2 weeks.

As you know, a condition of in-principle-acceptance is that the authors agree to deposit their Stage 1 accepted protocol in a repository, either publicly or under embargo until Stage 2 acceptance and publication. We are very keen to showcase our in-principle accepted protocols, so that our readers, reviewers, and potential authors can gain insight into the requirements of the format as well as an idea of the types of projects that are suitable for publication in Nature Human Behaviour. We have set up a space on figshare (https://springernature.figshare.com/registered-reports_NHB) to host all of our in-principle accepted protocols, which can either be made public or kept under embargo until Stage 2 acceptance (depending on author preference). This gives you the opportunity to have your work publicly associated with Nature Human Behaviour, and of course we will be very pleased to showcase your report if you agree to share it publicly.

Depositing the work on our figshare space does not preclude deposition of your Stage 1 protocol on other depositories – your protocol can also be posted on OSF, Dataverse, Dryad or any other public repository of your choice. You also do not need to do anything – if you agree with posting your protocol on our figshare space, we will upload your protocol on your behalf and either set it public or place it under embargo, depending on your choice. Your protocol will be licensed under a CC BY license (Creative Commons Attribution 4.0 International License). The CC BY license allows for maximum dissemination and re-use of open access materials and is preferred by many research funding bodies. Under this license users are free to share (copy, distribute and transmit) and remix (adapt) the contribution including for commercial purposes, providing they attribute the contribution in the manner specified by the author or licensor (read full legal code: <http://creativecommons.org/licenses/by/4.0/legalcode>) Please note that any use of <https://springernature.figshare.com> will be subject to the Figshare terms of use. Figshare has the right to enforce these terms and conditions where applicable. Use of third party services and sites will be subject to the relevant terms of use and will apply if we act on your behalf in this regard. Do let me know if you would like to take up this option or if you have any questions regarding the protocol deposition requirement.

Following completion of your study, we invite you to resubmit your paper for peer review as a Stage 2 Registered Report. Please note that your manuscript can still be rejected for publication at Stage 2 if the Editors consider any of the following to hold:

- The results were unable to test the authors' proposed hypotheses by failing to meet the approved outcome-neutral criteria
- The authors altered the Introduction, rationale, or hypotheses, as approved in the Stage 1 submission
- The authors failed to adhere closely to the registered experimental procedures
- Any post hoc (unregistered) analyses were either unjustified, insufficiently caveated, or overly dominant in shaping the authors' conclusions
- The authors' conclusions were not justified given the data obtained

We encourage you to read the complete guidelines for authors concerning Stage 2 submissions at <https://www.nature.com/nathumbehav/registeredreports>. Please especially note the requirements for protocol deposition, data sharing, and that withdrawing your manuscript will result in publication of a Retracted Registration.

In recognition of the time and expertise our reviewers provide to Nature Human Behaviour's editorial process, we would like to formally acknowledge their contribution to the external peer review of your manuscript entitled "Social media use in adolescent clinical and non-clinical populations". For those reviewers who give their assent, we will be publishing their names alongside the published article.

When you are ready, please use the following link to access your home page and submit your Stage 2 Registered Report:

Link Redacted

*This url links to your confidential homepage and associated information about manuscripts you may have submitted or be reviewing for us. If you wish to forward this e-mail to co-authors, please delete this link to your homepage first.

We expect your Stage 2 Registered Report to be submitted by the date specified in your latest cover letter. If unforeseen circumstances prevent submission by that date, please contact us as soon as possible to discuss any changes to the submission time-frame.

Thank you again for offering us this work and we look forward to receiving your Stage 2 Registered Report.

Yours sincerely,

[Redacted]

[Redacted] PhD

Nature Human Behaviour

Version 4:

Decision Letter:

4th December 2024

Dear Ms Fassi,

Thank you once again for submitting your Stage 2 Registered Report, entitled "Social media use in adolescents with and without mental health conditions," and for your patience during the re-review process.

Your manuscript has now been evaluated by Reviewers 2 and 3 from the previous rounds of review, whose comments are included at the end of this letter. In the light of our reviewers' advice, we are pleased to inform you that we will be able to accept your Stage 2 manuscript, pending revisions to address reviewer comments and editorial requests.

To guide the scope of the revisions, the editors discuss the referee reports in detail within the team, including with the chief editor, with a view to (1) identifying key priorities that should be addressed in revision and (2) overruling referee requests that are beyond the scope of Stage 2 Registered Reports.

Please find attached to this decision letter two documents: 1) author guidance checklist with general comments and requests, and 2) a marked-up version of your manuscript with specific comments, edits and requests.

One of the main reasons for delays in eventual acceptance is failure to fully comply with editorial policies and formatting requirements. To assist you with finalizing your manuscript for publication, I attach a checklist that lists all of our editorial policies and formatting requirements.

Please attend to *every item* in the checklist and upload a copy of the completed checklist with your submission. I have highlighted in the checklist items that require your attention. I also mention here a few points that are frequently missed and can cause delays:

- 1) Please insert the Protocol Registration information in your manuscript (including the figshare doi).
- 2) Ensure that all corresponding authors have linked their ORCID to their account on our online manuscript handling system. This is very frequently missed and invariably causes delays in formal acceptance.
- 3) Ensure that you provide all of the materials requested in the attached checklist and below with your final submission.

Nature Human Behaviour offers a transparent peer review option for new original research manuscripts submitted from 1st December 2019. We encourage increased transparency in peer review by publishing the reviewer comments, author rebuttal letters and editorial decision letters if the authors agree. Such peer review material is made available as a supplementary peer review file. **Please state in the cover letter 'I wish to participate in transparent peer review' if you want to opt in, or 'I do not wish to participate in transparent peer review' if you don't.** Failure to state your preference will result in delays in accepting your manuscript for publication.

We hope to hear from you within 4-8 weeks; please let us know if the revision process is likely to take longer.

To submit your revised manuscript, you will need to provide the following:

- Cover letter
- Point-by-point response to the reviewers (if applicable)
- Manuscript text (not including the figures) in .docx or .tex format
- Individual figure files (one figure per file)
- Extended Data & Supplementary Information, as instructed
- Reporting summary
- Editorial policy checklist
- Third-party rights table (if applicable)
- Suggestions for cover illustrations (if desired)

Consortia authorship:

For papers containing one or more consortia, all members of the consortium who contributed to the paper must be listed in the paper (i.e., print/online PDF). If necessary, individual authors can be listed in both the main author list and as a member of a consortium listed at the end of the paper. When submitting your revised manuscript via the online submission system, the consortium name should be entered as an author, together with the contact details of a nominated consortium representative. See <https://www.nature.com/authors/policies/authorship.html> for our authorship policy and <https://www.nature.com/documents/nr-consortia-formatting.pdf> for further consortia formatting guidelines, which should be adhered to prior to acceptance.

Forms:

Nature Human Behaviour has now transitioned to a unified Rights Collection system which will allow our Author Services team to quickly and easily collect the rights and permissions required to publish your work. Once your paper is accepted, you will receive an email in approximately 10 business days providing you with a link to complete the grant of rights. If you choose to publish Open Access, our Author Services team will also be in touch at that time regarding any additional information that may be required to arrange payment for your article.

For information regarding our different publishing models please see our <https://www.springernature.com/gp/open-research/transformative-journals> Transformative Journals page. If you have any questions about costs, Open Access requirements, or our legal forms, please contact ASJournals@springernature.com.

Link Redacted

With best regards,

██████████
██████████ PhD
██████████
Nature Human Behaviour

Reviewer #2 (Remarks to the Author):

As with the stage 1 report, I enjoyed reading the manuscript and am confident that this paper can make a very valuable contribution to the field. The presented study engages in an important comparison of social media use in adolescents with and without mental

health conditions based on high-quality data.

The authors kept the hypotheses and followed through with the analyses as described in the Stage 1 report. The change addition that is described and explained in the Stage 2 manuscript is the additional use of non-parametric tests for all hypotheses as “not all assumptions of linear models were met” (p. 16, line 456).

Overall, I only have a few minor comments that should be relatively easy to address for the authors. Many of them simply relate to some of the wording in the paper. I will list those in chronological order in the following.

To somewhat tone down the associated statement in the abstract I would propose to maybe replace the word “oversight” with “shortcoming” (same for the sentence on p. 5, line 97).

p. 6, line 134: I think “high levels of social comparisons” should be “high level of upward social comparisons” as that is what is described in the following parentheses here.

I know this is really minor and maybe a bit nitpicky (apologies for that), but, given that the authors used secondary data, I think wording like “We measured...” (e.g., at the beginning of the “Measures” section on p. 9) should be replaced with something like “The study measured...” or “XY was measured with/using...”.

I don’t think that the tense of all verbs relating to the measures needs to be changed to past. For example, when describing general attributes of the instruments used for the “mental health diagnoses”, I think it is fine (or even better) to stick with present tense (example: “The DAWBA used...”, p. 10, line 279). I would say the same for statements related to the reporting of results (e.g., p. 13, lines 364-365: “we reported regression coefficients” -> “we report...”).

p. 11, line 283: What exactly is meant by “if a participant scored highly on the SDQ...”. Does this refer to the “difficulties” part if this instrument?

On p. 12, lines 328-329, the authors write that “the simulated data and analysis code is available on OSF”. However, I did not find a dataset in the repository. What is included, is the “code for simulating and analyzing the data”. So, either the wording needs to be adapted or the (simulated) data should be uploaded to the associated repository.

Note: I have provided my remarks regarding the OSF repo and the code it contains in the separate field "Remarks on code availability".

As a final minor note, the manuscript contains a number of typos and punctuation errors that warrant another thorough proofreading (this could be done/facilitated via tools like Microsoft Editor, Grammarly, DeepL Write or Writefull).

Reviewer #2 (Remarks on code availability):

The authors have provided the full code for the power analysis and simulation as well as the descriptive and hypotheses-testing analysis via the Open Science Framework (OSF). The code is well-structured and properly commented. I was able to reproduce the power analysis and simulation (Stage 1 code). I could not reproduce the Stage 2 analyses as the underlying data cannot be shared.

Some things I noticed when checking the code:

1. Stage 1 code: The knitted HTML file is only provided for the `mhcyp_analysis.Rmd`, not for `mhcyp_power.Rmd`. Is there a reason for this or is this simply a minor oversight?
2. I would suggest that the authors provide ReadMe (esp. for the Stage 2 – analysis code) with further information, such as the order in which scripts should be run or information on using `renv` (as a side note: I very much appreciate that the authors use `renv` to increase computational reproducibility).

Reviewer #3 (Remarks to the Author):

This is a compelling manuscript that carefully reviews the relationship between qualitative and quantitative aspects of social media use with youth mental health diagnostic domains. The study is well designed, and the methods are transparently communicated. The introduction explains the current literature related to social media and youth mental health superbly, recognizing the many nuances of this area of study and topics for which the research is limited, and communicates the need for this study clearly. The results are quite interesting and are likely to be very useful to informing future research as well as prevention/intervention efforts. Below, I offer minor comments meant to increase clarity and transparency of the manuscript.

Introduction

- The introduction is quite compelling. One very minor comment would be to clearer earlier on the construct related to monitoring. There is much discussion of parental monitoring within the literature. Given that, a clearer notation related to self-monitoring may be helpful.

Results

- It would be helpful to briefly comment on the limitations of the sensitivity analyses within the main paper, e.g., the limited sample sizes and exploratory nature of the analyses.
- While a reasonable and unchangeable aspect of the study, I would suggest mentioning the data is now 7 years old as a potential limitation. Trends related to social media use change at a fast pace. It is not totally clear if some of the constructs studied would remain the same as new and different kinds of platforms become more prevalent (e.g., public vs. private platforms, those on which communication is anonymous, or platforms on which moderation is common).

Supplements

- Supplementary Figure 5 seems to pertain to sex rather than gender.
- SES could be a key factor that could contribute to differences in qualitative aspects of use, e.g., social comparison. A suggestion would be to provide an additional supplementary figure, like Figure 5, based on levels of SES.
- Supplementary Figures 3 and 4, while nice looking, are difficult to decipher. Please define the axes.

Version 5:

Decision Letter:

Dear Ms Fassi,

We are pleased to inform you that your Registered Report "Social media use in adolescents with and without mental health conditions", has now been accepted for publication in Nature Human Behaviour.

With best regards,

 PhD
 Nature Human Behaviour

P.S. Click on the following link if you would like to recommend Nature Human Behaviour to your librarian <http://www.nature.com/subscriptions/recommend.html#forms>

** Visit the Springer Nature Editorial and Publishing website at http://editorial-jobs.springernature.com?utm_source=ejp_NHumB_email&utm_medium=ejp_NHumB_email&utm_campaign=ejp_NHumB for more information about our career opportunities. If you have any questions please click www.springernature.com/editorial-and-publishing-jobs

href="mailto:editorial.publishing.jobs@springernature.com">here.**

Dear Dr [REDACTED] [REDACTED]

Thank you for sending us the reviews for our manuscript ‘Social media use in adolescent clinical and non-clinical populations’ (NATHUMBEHAV-22123378). We are grateful for the feedback provided that has allowed us to improve our Stage 1 Registered Report. We address the editor and reviewers’ comments below and highlight any changes made to our manuscript.

Editor comments

E1) Please specifically address Reviewer 1’s concerns about the meaningfulness of your classification of disorders for clinical practice. We ask that in your revised manuscript, you provide a new and more clinically-relevant classification of diagnoses.

We thank the editor for raising this point; we have now updated our manuscript to provide a new and more clinically relevant classification of diagnoses by implementing three changes.

First, we have amended our internalising category to exclude mania and bipolar affective disorder. We adopted this more conservative approach to be in line with relevant research literature examining the structure of internalising and externalising pathology (see the response to R1.1 and R1.2). In practical terms, given the very low prevalence of these disorders, together equalling less than 0.1% in the MHCYP dataset, we do not expect this change to meaningfully affect our hypotheses or power analysis.

Second, we have added an exploratory analysis (see the response to R1.2) aimed at assessing the effects of interest across individual diagnoses or categories of diagnoses. We will a) provide descriptive statistics for each diagnosis separately (e.g., generalised anxiety disorder, obsessive-compulsive disorder) and b) conduct follow-up analyses with specific diagnoses or categories of diagnoses (e.g., anxiety disorders, depressive disorders) where we have sufficient statistical power. We will note the limits of our confidence in these results, as they will be exploratory and data-dependent.

Third, we have added an additional exploratory analysis of mental health severity, as indexed by the presence of multiple diagnoses, on the effects of interest (see the response to R1.2). This analysis will allow us to assess the importance of comorbid mental health diagnoses in relation to social media use.

Overall, we believe that these three changes have increased the clinical value of our analysis and will offer practical information of use to clinicians and researchers interested in this field and area in particular.

E2) We ask that you follow Reviewer 2’s advice and consider changes to the design and the technical approach to generate meaningful robustness checks.

We thank the editor for highlighting this point, which we address in two ways.

First, we have added a sensitivity analysis to assess whether considering social media use on weekdays and weekends/holidays separately shows a different pattern of results compared to the main analyses that use a weighted average.

Second, we have implemented a dependency management solution using *renv* to ensure the full reproducibility of our analysis pipelines. We have now added dependency management using *renv* to our project on the Open Science Framework by 1) updating our analysis scripts (*20230330_RR_S1_mbcyp_analysis.Rmd*) and 2) adding a file (*renv.lock*) that includes all dependencies and sub-dependencies with the correct version.

Reviewer 1 comments

Overall, this is a well-thought out, comprehensive study that once performed, will assuredly garner attention in the field. It is well-written, and the social media use of youth with psychiatric pathology is certainly an area where more attention needs to be paid from a research perspective.

We thank the reviewer for their overall support of this work.

R1.1) However, I do have some significant concerns that may or not be able to be addressed in your proposed study design: I think your diagnostic categorisations for internalising vs externalising disorders don't necessarily make sense from a clinical perspective. For example, complex post-traumatic stress disorder and mania (and therefore BPAD) are disorders with well-established externalising features. I would think that lumping those diagnoses in with internalising disorders would impact your findings.

We thank the reviewer for raising their concerns about our diagnostic categorisation. In agreement with the reviewer, we have decided to exclude mania and bipolar affective disorder (BPAD) from the analyses run to address Research Question 2 (comparing internalising conditions to no diagnosis and comparing externalising conditions to no diagnosis) and Research Question 3 (comparing internalising to externalising conditions), as presented in section A below. In addition, while complex PTSD (C-PTSD) was not assessed in the present dataset, we argue for keeping the standard PTSD diagnosis among our internalising diagnosis categories, as presented in section B below.

A) Mania and bipolar affective disorder

We initially included mania and BPAD in the internalising category based on research suggesting that bipolar disorder shows notable relations with internalising spectrum pathology (e.g., Eaton et al., 2012; Kessler et al., 2011; Vaidyanathan, Patrick, & Iacono, 2011; Watson, 2005). However, we agree with the reviewer's point that mania, especially, and consequently BPAD, does not neatly align with "internalising" symptomatology. This perspective is also supported by the work noted above, which suggests that BPAD's diagnostic variance accounted for by internalising pathology is lower than most other internalising disorders. Specifically, around 50% of BPAD's diagnostic variance is captured by internalising pathology, compared to around 90% of major depressive disorder's diagnostic variance (Eaton et al., 2012). We have therefore removed both mania and BPAD from the internalising category. These diagnoses will be excluded from analyses in Research Questions 2 and 3. Given the very low prevalence of these disorders (together equalling less than

.1% in the MHCYP dataset; Vizard et al., 2018), we do not expect this to change our hypotheses or power analysis meaningfully.

We have updated our Methods and Table 2 accordingly, as reported below (Revised Manuscript, Page 37, Table 2). Parts of the text highlighted in grey indicate additions to the original text.

Table 2. Summary of our categorisation of mental health diagnoses into internalising and externalising conditions.

Grouped diagnostic category	Diagnoses received in the DAWBA
Internalising	Separation anxiety disorder, generalised anxiety disorder, obsessive-compulsive disorder, specific phobia, social phobia, agoraphobia, panic disorder, post-traumatic stress disorder, other anxiety disorder, body dysmorphic disorder, major depressive episode, other depressive episode, bipolar disorder, mania and eating disorders.
Externalising	Hyperkinetic disorder, other hyperactivity disorder, oppositional defiant disorder, conduct disorder confined to family, unsocialised conduct disorder, socialised conduct disorder, other conduct disorders.
Excluded^a from analyses in Question 2 and 3	Autism spectrum disorder, tic disorder, and psychotic disorders, mania, and bipolar affective disorder.

Note^a. We exclude adolescents diagnosed with these conditions as they do not clearly map onto the symptomatology of either internalising or externalising diagnoses^{33,57}. **Further, because the diagnostic variance of mania and bipolar disorder accounted for by internalising pathology is lower than most other internalising disorders, such as anxiety and depression (Eaton et al., 2012), we decided to exclude these diagnoses from the internalising category.**

B) Post-traumatic stress disorder

The diagnosis of complex PTSD (C-PTSD) was not assessed in the MHCYP survey. The DAWBA, which was used to collect diagnostic information, is based on ICD-10 and DSM-5 diagnoses, and C-PTSD was not introduced as a diagnosis into the ICD-11 until 2018. Related diagnoses in the ICD-10, such as enduring personality change after catastrophic experiences (code F62.0), are not assessed by the DAWBA. Compared to C-PTSD, there is stronger evidence suggesting a preponderance of internalising over externalising features for PTSD (e.g., Kotov et al., 2021; Miller & Resick, 2007), hence our decision to include that diagnosis in the internalising category.

References:

- Eaton, N. R., Krueger, R. F., Keyes, K. M., Wall, M., Hasin, D. S., Markon, K. E., Skodol, A. E., & Grant, B. F. (2013). The structure and predictive validity of the internalising disorders. *Journal of Abnormal Psychology*, 122(1), 86–92. <https://doi.org/10.1037/a0029598>
- Kessler, R. C., Ormel, J., Petukhova, M., McLaughlin, K. A., Green, J. G., Russo, L. J., Stein, D. J., Zaslavsky, A. M., Aguilar-Gaxiola, S., Alonso, J., Andrade, L., Benjet, C., de Girolamo, G., de Graaf, R., Demyttenaere, K., Fayyad, J., Haro, J. M., Hu, C. yi, Karam, A., ... Üstün, T. B. (2011). Development of lifetime comorbidity in the World Health Organization World Mental Health Surveys. *Archives of General Psychiatry*, 68(1), 90–100. <https://doi.org/10.1001/archgenpsychiatry.2010.180>
- Kotov, R., Krueger, R. F., Watson, D., Cicero, D. C., Conway, C. C., DeYoung, C. G., Eaton, N. R., Forbes, M. K., Hallquist, M. N., Latzman, R. D., Mullins-Sweatt, S. N., Ruggero, C. J., Simms, L. J., Waldman, I. D., Waszczuk, M. A., & Wright, A. G. C. (2021). The Hierarchical Taxonomy of Psychopathology (HiTOP): A quantitative nosology based on consensus of evidence. *Annual Review of Clinical Psychology*, 17(1), 83–108. <https://doi.org/10.1146/annurev-clinpsy-081219-093304>
- Miller, M. W., & Resick, P. A. (2007). Internalising and externalising subtypes in female sexual assault survivors: Implications for the understanding of complex PTSD. *Behavior Therapy*, 38(1), 58–71. <https://doi.org/10.1016/j.beth.2006.04.003>
- Vaidyanathan, U., Patrick, C. J., & Iacono, W. G. (2011). Patterns of comorbidity among mental disorders: A person-centered approach. *Comprehensive Psychiatry*, 52(5), 527–535. <https://doi.org/10.1016/j.comppsy.2010.10.006>
- Vizard, T., Pearce, N., Davis, J., Sadler, K., Ford, T., Goodman, A., Goodman, R., McManus, S. (2018). Mental health of children and young people in England, 2017: Emotional disorders. Retrieved from <https://files.digital.nhs.uk/14/0E2282/MHCYP%202017%20Emotional%20Disorders.pdf>
- Watson, D. (2005). Rethinking the mood and anxiety disorders: A quantitative hierarchical model for DSM-V. *Journal of Abnormal Psychology*, 114(4), 522–536. <https://doi.org/10.1037/0021-843X.114.4.522>

R1.2) Additionally, I don't know that it's a) beneficial from a clinical standpoint or b) accurate from a research standpoint to split diagnoses as internalising vs externalising, especially if you're not accounting for the presence youth with comorbid externalising behaviors like substance use that won't get screened out as a "between-group comorbidity." The online behaviors of youth with a history of early childhood PTSD are generally quite different from youth with a generalised anxiety disorder diagnosis; I would recommend rethinking how you classify diagnoses in your questions 2 and 3 in a way that might be more clinically relevant e.g., PTSD, Anxiety disorders, Depressive disorders.

We agree with the sentiment expressed by the reviewer about establishing and ensuring the clinical utility of this work. To address the reviewer's concerns, we first expand on our reasons for splitting

diagnoses into internalising and externalising (section A below), and we then update our methods to include a new and clinically relevant classification of diagnoses (section B below).

A) Classification of internalising vs externalising diagnoses

Our decision to divide diagnoses into internalising and externalising categories is informed by three arguments.

First, given the high rates of comorbidity within internalising and externalising conditions (Kessler et al., 2011) and the comparatively lower reliability of individual diagnoses (Baca-Garcia et al., 2007; Regier et al., 2013), we expect that splitting along this dimension will help capture some of the variance associated with transdiagnostic mental health symptoms.

Second, the majority of our analyses examine the quality of adolescents' engagement with social media (e.g., "I am happy with the number of friends I have on social media", "I compare myself to others on social media") rather than the content of that engagement (e.g., who they follow, what they search or post). Theoretically, the quality of engagement, as opposed to the content, should be more consistent across diagnoses which share underlying mechanisms. For example, cognitive theory suggests that both GAD and PTSD are characterised, at least in part, by the presence of negative beliefs about the self, others, and the world (Clark, Beck & Alford, 1999; Clark & Beck, 2011). Hence, while individuals with GAD and PTSD may use social media in different ways, it is likely that the quality of their interaction with social media would show similarities, given the overlap in beliefs (e.g., about the self as damaged) which are, by a cognitive account, maintaining their symptoms.

Third, classifying diagnoses into internalising and externalising categories allows us to be more confident that we will have the statistical power necessary to test our hypotheses. As we have not accessed the data and cannot be certain of the distribution of diagnoses across the sample, we chose to maximise our power by splitting the sample along only one dimension (i.e., internalising and externalising diagnoses). This is key as splitting the sample based on specific diagnoses (e.g., depression, social anxiety) would result in many of our analyses being underpowered.

B) New classification of diagnoses

We agree with the reviewer's perspective that removing individuals with between-group comorbidity (e.g., those with comorbid ADHD and depression) could impact the clinical utility of this work. In light of this, we have decided to add a sensitivity analysis for Research Question 2, including adolescents with between-group comorbidities when comparing internalising and externalising to no diagnosis. While this analysis will not allow us to draw conclusions about internalising and externalising diagnoses only, it will inform the extent to which adolescents with these and any other comorbid diagnosis differ in social media use from those without a diagnosis. as reported below (Page 11, Lines 303-304 and Page 12, Lines 329-330):

"To increase the clinical utility of this work, we will run a sensitivity analysis for Question 2, including people with between-group comorbidity."

Further, we will conduct and report exploratory analyses to examine differences in social media use between adolescents falling within specific diagnostic categories and adolescents without a

diagnosis. For cases in which we will have the requisite power, these analyses will be included as exploratory (Page 10, Lines 75-83), as reported below:

“To increase the clinical utility of this work, we will run additional exploratory analyses examining responses to all social media questions (a-g) separately for individual diagnoses (e.g., major depressive disorder) or diagnostic categories (e.g., depressive disorders or anxiety disorders). First, we will present descriptive statistics for each specific diagnosis, as long as there are at least ten individuals with that diagnosis. Next, for individual diagnoses or diagnostic categories with sufficient sample size, we will conduct follow-up analyses to test whether individuals falling within each group respond differently to each social media question (a-g) than individuals with no diagnosis. As comorbid mental health conditions are extremely common (Kessler et al., 2005), we will include individuals with multiple conditions in these analyses. Further, we will conduct sensitivity analyses to assess whether the results are meaningfully different when individuals with comorbidities are removed.”

To further address concerns raised by the reviewer related to the clinical value of this work, we plan to conduct additional exploratory analyses accounting for mental health severity, as indexed by the presence of multiple diagnoses. These analyses will be exploratory and are described in the Supplementary Methods (Page 10, Line 84-88):

“To examine whether social media use varies with the severity of mental health concerns, we will conduct exploratory analyses to assess whether the presence of multiple diagnoses is related to the outcomes of interest (a-g). First, we will plot responses to each social media question by the number of diagnoses. Next, we will test for a linear effect of the number of diagnoses on social media question response using linear regression models.”

Of note, we decided to these analyses on specific diagnoses and mental health severity as exploratory analyses because a) our ability to conduct them will be conditioned on sample size and power, and b) we do not have specific hypotheses related to these questions.

Finally, we also agree with the reviewer’s suggestion that comorbid substance use disorders (which would be considered externalising pathology) may be present. Since substance use disorders are not assessed as part of our dataset, it is possible that individuals with comorbid depression and substance use, for example, will be identified in the internalising (only) category, despite having an undiagnosed substance use disorder. Given that substance use is comorbid with externalising as well as internalising conditions (e.g., O’Neil et al., 2011), we do not think this is a major threat to the internal validity of this work. However, we fully intend to address this as a limitation of the generalisability of our work in our discussion.

References:

- Kessler, R. C., Ormel, ... Üstün, T. B. (2011). Development of lifetime comorbidity in the World Health Organization World Mental Health Surveys. *Archives of General Psychiatry*, 68(1), 90–100. <https://doi.org/10.1001/archgenpsychiatry.2010.180>
- Baca-Garcia, E., Perez-Rodriguez, M. M., Basurte-Villamor, I., Moral, A. L. F. D., Jimenez-Arriero, M. A., Rivera, J. L. G. D., Saiz-Ruiz, J., & Oquendo, M. A. (2007). Diagnostic stability of

psychiatric disorders in clinical practice. *The British Journal of Psychiatry*, 190(3), 210–216. <https://doi.org/10.1192/bjp.bp.106.024026>

Clark, D. A., & Beck, A. T. (2011). *Cognitive therapy of anxiety disorders: Science and practice*. Guilford Press.

Clark, D. A., Beck, A. T., & Alford, B. A. (1999). *Scientific foundations of cognitive theory and therapy of depression*. John Wiley & Sons Inc.

O'Neil, K. A., Conner, B. T., & Kendall, P. C. (2011). Internalising disorders and substance use disorders in youth: Comorbidity, risk, temporal order, and implications for intervention. *Clinical Psychology Review*, 31(1), 104–112. <https://doi.org/10.1016/j.cpr.2010.08.002>

Regier, D. A., Narrow, W. E., Clarke, D. E., Kraemer, H. C., Kuramoto, S. J., Kuhl, E. A., & Kupfer, D. J. (2013). DSM-5 Field Trials in the United States and Canada, Part II: Test-retest reliability of selected categorical diagnoses. *American Journal of Psychiatry*, 170(1), 59–70. <https://doi.org/10.1176/appi.ajp.2012.12070999>

R1.3) Minor issues:

a) There are some areas where the article and subsequent noun don't match from a grammar perspective (e.g. pg 16, "an internalising diagnoses.")

We thank the reviewer for noticing this grammar error, which we have corrected (Revised Manuscript, Page 16, Lines 429, 434, 437). Further, we also corrected additional spelling errors in the Revised manuscript (see Page 9, Line 225 and 233; Page 11, Line 302; Page 12, Line 336; Page 13, Line 357).

b) I couldn't tell if you were asking about time spent on social media on weekends separately from on weekdays. Those are likely markedly different amounts of time and it may be hard for kids to provide a single answer if so.

We agree with the reviewer about the potential importance of asking separate questions about time spent on social media on weekdays compared to weekends. Adolescents were indeed asked two separate questions regarding time spent on social media. We have edited the text in the Revised Manuscript to clarify this, parts of the text highlighted in grey indicate additions to the original text (Revised Manuscript, Page 8, Line 212-222):

“We will measure time spent on social media using two questionnaire items: “When you use social media sites or apps, how much time do you spend using them on a typical school day?” (SMTimeSpentS), and “When you use social media sites or apps how much time in total do you spend using them on a typical weekend or holiday day?” (SMTimeSpentW). Participants answered both questions on a 9-point Likert scale: 1 = Less than 30 minutes; 2 = More than 30 minutes but less than an hour; 3 = One to two hours; 4 = Two to three hours; 5 = Three to four hours; 6 = Four to five hours; 7 = Five to six hours; 8 = Six to seven hours; 9 = More than seven hours.”

Our main analyses will examine time spent on social media by computing a single variable reflecting the weighted average from the two questionnaire items capturing hours spent on weekdays and weekends/holidays. Following the reviewer's comment on the importance of

differentiating time spent on social media on weekdays from time spent on weekends, we have now added a sensitivity analysis. Specifically, we will assess whether considering social media use on weekdays vs weekends/holidays separately changes the results of the main analyses that use a weighted average, as reported below (Revised Manuscript, Page 9, Lines 228-232):

“We will use a sensitivity analysis to test whether examining social media use on weekdays vs weekends/holidays separately, rather than as a weighted average, changes the main results.”

c) Moreover, weeknight social media use might actually be more interesting to examine given that it could give more insight into a youth's impulse control.

We very much agree with the reviewer on the importance of assessing weeknight social media use. However, the MHCYP questionnaire did not include questions on night-time use, so it will not be possible for us to assess its association with mental health.

Reviewer 2 comments

The manuscript addresses a timely and highly relevant topic. The relationship between social media use and mental health among adolescents is an intensely debated topic in in academia as well as the public. Importantly, arguments made in this debate are often not backed by robust empirical evidence. This study can make a very important contribution in that regard. In the public debate, but sometimes also in academic research, the lines between well-being and clinically relevant behaviors and outcomes are often blurred. Hence, one thing that I see as a particular strength of the study is the use of clinically validated measures (in particular, "standardised diagnostic assessments by professional clinical raters", p.24). Another very important issue that is addressed by this study is the lack of clinical populations in previous research. Overall, I believe that the study can make a very valuable contribution to the field as empirically sound and methodologically sophisticated and robust research is much needed in this contested area. The study employs secondary use of high-quality data with a large enough sample size and, unlike many previous studies, considers quantitative as well as qualitative dimensions of social media use. The hypotheses are precise, soundly formulated hypotheses, and convincingly derived from the existing literature. The distinction between different dimensions of social media use and between internalising and externalising conditions is particularly useful and interesting here. The manuscript provides detailed information about the sampling plan and data processing, both of which seem appropriate for testing the elicited hypotheses. The authors also present a suitable a priori power analysis for each group of hypotheses based on the expected sample sizes. The code for the power analysis as well as simulations of the hypothesis tests are provided and I was able to reproduce the reported outcomes of those on my computer. The software used for the analysis (and the version) are also reported, which is helpful for reproducing the analyses. The proposed analysis methods are simple yet appropriate for testing the hypotheses. I also found the appendix tables very helpful as summaries of the preregistration. Overall, I only have a few remarks regarding how the authors might improve the eventual full paper:

We thank the reviewer for these kind comments and overall appreciation for our work.

R2.1) I would say that the categorical measures of daily social media use may not be optimal. I am aware that the authors cannot change this as this is secondary data, but this should be discussed in the full paper.

We thank the reviewer for this comment, and confirm that the limitations of using categorical measures of social media use will be comprehensively addressed in the discussion section of the Stage 2 Registered Report.

R2.2) While the proposed processing of these data into a composite social media use score makes sense, it might make sense to alter some of the decisions made for this process to generate robustness checks for the results.

We agree with the reviewer on the importance of checking the robustness and accuracy of using a composite score (weighted average) for hours spent on social media on weekdays and weekends/holidays. Following both reviewers' comments (see also R1.3), we have now added a sensitivity analysis to our Revised Manuscript. Specifically, we will assess whether analysing social

media use on weekdays and weekends/holidays separately changes the results of the main analyses, which will initially use a weighted average. We now detail this in the manuscript (Revised Manuscript, Page 9, Lines 228-232):

“We will use a sensitivity analysis to test whether examining social media use on weekdays vs weekends/holidays separately, rather than as a weighted average, changes the main results.”

R2.3) In general, as the authors are surely aware of, self-reports of media use can be biased (e.g., due to social desirability). Hence, I think this is imported to address and also discuss potential alternatives, such as tracking data or ambulatory/ecological momentary assessment methods.

We thank the reviewer for this comment and confirm that the limitations of self-report measures of social media use will be comprehensively addressed in the discussion section of the Stage 2 Registered Report.

R2.4) While the general(ized) linear model underlies both of these options and I think the use of regression analysis is perfectly fine, the authors may want to at least briefly explain why they decided against using analysis of variance which is common in many study designs that compare groups on various outcome variables.

We thank the reviewer for this comment. There are three key reasons why we chose to use a regression instead of Analysis of Variance (ANOVA). First, ANOVA is a special case of generalised linear models. The main difference between the two is that predictors in ANOVA, also identified as factors, are usually categorical, while regressions can handle a wider range of predictors, e.g., those that are continuous (Nelson & Zaichkowsky, 1979). Given that the social media variables in our study will be coded as continuous, regression is best suited for our analyses. Second, regression allows us to handle unbalanced designs (e.g., the size of the groups compared is unequal), which is a limitation of ANOVA (Judd et al., 2017; Nelson & Zaichkowsky, 1979). Third, regression allows for more analytical flexibility, as it can include multiple covariates. This will allow us to run exploratory analyses as part of the Stage 2 Registered Report without substantially changing our analytical approach. We have now clarified our reasons for using regression models in the manuscript (Revised Manuscript, Page 13, Lines 363-365):

“Of note, our analytical approach is based on regressions rather than ANOVAs as the former allow for more diverse predictors, unbalanced groups and inclusion of covariates in potential exploratory analyses (Judd et al., 2017).”

References

- Judd, C. M., McClelland, G. H., & Ryan, C. S. (2017). *Data analysis: A model comparison approach to regression, ANOVA, and beyond*. Routledge.
- Nelson, L. R., & Zaichkowsky, L. D. (1979). A case for using multiple regression instead of ANOVA in educational research. *The Journal of Experimental Education*, 47(4), 324-330.

R2.5) Regarding the sharing of materials: In addition to sharing scripts, I would suggest also using a dependency management solution like `renv` (<https://rstudio.github.io/renv/articles/renv.html>) or `rang` (<https://github.com/chainsawriot/rang>) plus `rocker` (<https://github.com/rocker-org/rocker>) to ensure that the package versions are the same for reproductions of the analyses.

We thank the reviewer for this helpful suggestion. We have now implemented dependency management using `renv` into our analysis scripts (`20230330_RR_S1_mbcyp_analysis.Rmd`), which have been updated on the Open Science Framework. We have also added the `renv.lock` file to our Open Science Framework repository, containing all project dependencies and sub-dependencies with the correct version (`renv.lock`). We note that `renv` is currently applied to the packages used for data simulation (i.e., `simstudy` and `dphyr`). We will extend this to the packages used for data analysis and visualisation in the Stage 2 Registered Report.

R2.6) Finally, what would be very interesting and valuable for a full paper, in my view, would be a detailed discussion of whether or in what way the results can be expected to generalise for other samples and esp. also other countries.

We thank the reviewer for this comment and confirm that we will comprehensively address the generalisability of this work to different samples and countries in the discussion section of the Stage 2 Registered Report. Further, we aim to include descriptive statistics reporting not only the age and gender but also the ethnicity and socioeconomic background of all participants in the Stage 2 Registered Report.

We greatly appreciate your reviews and look forward to your response.

Yours sincerely,

Luisa Fassi

PhD Researcher

MRC Cognition and Brain Sciences Unit,
Department of Psychiatry,
University of Cambridge

Dr Amanda Ferguson

Postdoctoral researcher

MRC Cognition and Brain Sciences,
University of Cambridge

Dr Amy Orben

Programme Leader Track Scientist

MRC Cognition and Brain Sciences Unit,
University of Cambridge

19th June 2023

Dear Dr [REDACTED] [REDACTED]

Thank you for sending us the reviews for our manuscript “Social media use in adolescent clinical and non-clinical populations” (NATHUMBEHAV-22123378A). We are grateful for the feedback provided that has allowed us to improve our Stage 1 Registered Report. We address the reviewers’ comments below and highlight any changes made. A revised version of the manuscript and supplementary materials are attached as separate documents.

Reviewer 2

R2.1) As before, I enjoyed reading the manuscript and believe that it has the potential to make a very valuable contribution to the field. I also appreciate the detailed and thoughtful response letter. After reading the revised Stage 1 Registered Report as well as the response letter, I can say that the authors have adequately addressed all of my previous comments, and I look forward to reading the Stage 2 Registered Report.

We thank the reviewer for their ongoing support of this work.

Reviewer 3

R3.1) This manuscript addresses a timely topic of high public health importance through a carefully considered plan for evaluating how youth with mental health conditions use social media. Overall, the hypotheses are well stated, planned analyses are clear and seem aligned with the stated goals of the project. Sensitivity analyses seem appropriate to appropriately interrogate the data. Examining both quantitative and qualitative aspects of use, informed by a youth advisory board, are strengths of the planned approach. The delineation between internalizing & externalizing disorders, and disorders that were excluded were reasonable, and the transdiagnostic approach is likely to have wide-ranging utility. Some additional comments/suggested revisions are noted below, which may advance the quality and clarity of the manuscript.

We thank the reviewer for these positive comments and overall appreciation for our work.

R3.2) Within the introduction, the background provides statistics on use that specifies older adolescents. Given that the age range for the study begins at age 11 offering statistics for younger adolescents seems necessary.

We now include statistics for both younger adolescents’ mental health (Revised Manuscript, Page 3, Lines 24-26) and social media use (Revised Manuscript, Page 3, Line 30):

“Recent UK data suggests that one in six 7-16-year-olds and one in four 17-19-year-olds have a probable mental health condition, a clear rise from the one in nine and one in ten recorded in 2017, respectively (NHS Digital, 2022)”

“Many have raised concerns that this trend has been caused, at least in part, by increased adolescent social media use, which has revolutionised how adolescents live, learn and interact (93% of 12-17-year-olds now report having a social media profile; Ofcom, 2022).”

References

NHS Digital 2022 Report. <https://digital.nhs.uk/news/2022/rate-of-mental-disorders-among-17-to-19-year-olds-increased-in-2022-new-report-shows>

Ofcom 2022 Report.

https://www.ofcom.org.uk/data/assets/pdf_file/0027/255852/childrens-media-use-and-attitudes-report-2023.pdf

R3.3) Greater context is needed in understanding social comparison hypotheses. Social comparison can have positive functions and be developmentally typical for youth. What specifically is the construct measured here, e.g., upward or downward social comparison, any form of social comparison that has carries negative affective consequences, etc.?

The item addressing social comparison in our dataset (i.e., “I compare myself to others on social media sites and apps”) does not specify upward or downward social comparison. In generating hypotheses for this item, we drew from literature demonstrating that upward, compared to downward, social comparison is more common on social media platforms (Verduyn et al., 2020). Adolescents’ bias for posting positive or flattering (vs unflattering) information on social media generally creates more opportunities for upward social comparison (e.g., Chou & Edge, 2012; Gonzales & Hancock, 2011). This aligns with additional research suggesting that the tendency to compare oneself to others online (e.g., measured with questions such as “I use electronic interaction to compare my life with other people’s lives.” from the Technology-Based Social Comparison and Feedback-Seeking Scale; Nesi & Prinstein, 2015) is related to depressive symptoms in young people (Nesi & Prinstein, 2015). We have now integrated this background information in the Revised Manuscript as follows (Page 6, Lines 129-133):

“However, high levels of upward social comparisons (i.e., comparisons with those believed to be of higher status than the self) have been associated with poor mental health³⁹⁻⁴³. Previous work suggests that most social comparisons made on social media sites are upward rather than downward (Verduyn et al., 2020), possibly because individuals tend to portray themselves in an ideal manner online (Chou & Edge, 2012; Gonzales & Hancock, 2011).”

Further, we acknowledge this as a limitation of our study and we will therefore address it in the discussion section of the Stage 2 Registered Report.

References

Chou, H.-T. G., & Edge, N. (2012). “They are happier and having better lives than I am”: The impact of using Facebook on perceptions of others’ lives. *Cyberpsychology, Behavior and Social Networking*, 15, 117–121.

Gonzales, A. L., & Hancock, J. T. (2011). Mirror, mirror on my Facebook wall: Effects of exposure to Facebook on self-esteem. *Cyberpsychology, Behavior and Social Networking*, 14, 79–83.

Nesi, J., & Prinstein, M. J. (2015). Using social media for social comparison and feedback-seeking: Gender and popularity moderate associations with depressive symptoms. *Journal of abnormal child psychology*, 43, 1427-1438.

Verduyn, P., Gugushvili, N., Massar, K., Täht, K., & Kross, E. (2020). Social comparison on social networking sites. *Current Opinion in Psychology*, 36, 32-37.

R3.4) Given the limited evidence within clinical samples, it seems difficult to determine whether adolescents with mental health conditions will have greater social media use vs. statistically different social media use. For example, one study not cited here found lower usage was associated with suicidal ideation (see Hamilton et al, 2021). Other studies, evaluating social media data, have shown variability in use patterns among youth with different clinical characteristics with some groups showing trends toward blunted use (see Loveys et al, 2017 from the Fourth Workshop on Computational Linguistics and Clinical Psychology).

We agree that, given the mixed state of the literature, establishing directional hypotheses for the differences in social media use between clinical and non-clinical groups is challenging. In this Registered Report, we address this question by drawing insights from three streams of literature, namely: youth mental health, developmental psychology and social media research.

To address the reviewer's comment, we summarise our rationale for the hypothesised differences between clinical and non-clinical samples separately for a) time spent on social media and b) other measures of social media engagement.

A) Time spent on social media

First, we thank the reviewer for highlighting the study by Hamilton et al. (2021), which we have now cited (see the Revised Manuscript, Table 3, Page 42):

“Further, in an independent clinical sample, using social media less (overall and messaging) was associated with higher levels of suicidal ideation over the next 30 days.”

Despite the mixed results, particularly in the very small number of studies on clinical groups, the majority of studies that examine community samples present converging evidence that time spent on social media is associated with modestly higher internalising and externalising symptoms (Cunningham et al., 2021; Valkenburg et al., 2022; Riehm et al., 2019). As this has been confirmed by well-power meta-analyses and reviews, we hypothesise that adolescent clinical groups – that experience higher levels of such mental health symptoms – spend more time overall on social media compared to non-clinical groups. We have now restructured the first part of Table 3 to provide a more comprehensive summary of the evidence on the relationship between time spent on social media and mental health (see the Revised Manuscript, Table 3, Page 42).

“Research on the relationship between social media use and mental health outcomes in different samples has yielded limited and conflicting results. On one hand, studies have found that young individuals diagnosed with depression tend to spend more time on social media compared to healthy controls (Akkin Gürbüz et al., 2017). Further, studies that focus on excessive rather than average time spent on social media show comorbidity between anxiety, depression, attention deficit hyperactivity disorder and excessive time spent on social media (Hussain & Griffiths, 2018).

However, in a sample of hospitalized adolescents with psychiatric conditions, the frequency of social media use and perception of overuse was not associated with clinical severity (Nesi et al., 2019). Additionally, in an independent clinical sample, using social media less, both overall and for messaging, was linked to higher levels of suicidal ideation over the next 30 days (Hamilton et al., 2021).

There is more work on community samples available, examining the average time spent on social media (mostly self-reported) and its relation to depression, anxiety and other indicators of mental health. Recent reviews and meta-analyses have reached a general agreement that associations are weak and positive (higher social media use is linked with higher levels of anxiety and depression, see Cunningham et al., 2021; Valkenburg et al., 2022; Riehm et al., 2019). Overall, despite mixed evidence, research therefore seems to suggest a small relationship between more time spent on social media and lower mental health, both when considering internalising and externalising symptoms. We hypothesise this to hold for clinical samples that are, therefore, expected to spend more time on social media.”

References

- Akkın Gürbüz, H. G., Demir, T., Gökalp Özcan, B., Kadak, M. T., & Poyraz, B. Ç. (2017). Use of social network sites among depressed adolescents. *Behaviour & Information Technology*, 36(5), 517–523.
- Hamilton, J. L., Biernesser, C., Moreno, M. A., Porta, G., Hamilton, E., Johnson, K., Poling, K. D., Sakolsky, D., Brent, D. A., & Goldstein, T. G. (2021). Social media use and prospective suicidal thoughts and behaviors among adolescents at high risk for suicide. *Suicide and Life-Threatening Behavior*, 51(6), 1203–1212.
- Nesi, J., Wolff, J. C. & Hunt, J. (2019) Patterns of Social Media Use Among Adolescents Who Are Psychiatrically Hospitalized. *J Am Acad Child Adolescent Psychiatry* 58, 635-639.e1.
- Cunningham, S., Hudson, C. C., & Harkness, K. (2021). Social Media and Depression Symptoms: A Meta-Analysis. *Research on Child and Adolescent Psychopathology*, 49(2), 241–253.
- Valkenburg, P. M., Meier, A., & Beyens, I. (2022). Social media use and its impact on adolescent mental health: An umbrella review of the evidence. *Current Opinion in Psychology*, 44, 58–68.
- Riehm, K. E., Feder, K. A., Tormohlen, K. N., Crum, R. M., Young, A. S., Green, K. M., ... & Mojtabai, R. (2019). Associations between time spent using social media and internalizing and externalizing problems among US youth. *JAMA psychiatry*, 76(12), 1266-1273.

B) Social media engagement

As noted by the reviewer, there is limited research on social media use beyond time spent (e.g., online self-disclosure, authentic self-presentation etc), making this literature unable to support well-reasoned hypotheses. Our hypotheses regarding differences between clinical and non-clinical groups in social media engagement, therefore, draw mostly from the mental health and developmental literature. They are based on the premise that the behavioural patterns exhibited

offline by adolescents with mental health conditions parallel their online engagements. For instance, adolescents with internalizing symptoms often hold negative self-views, which lead to engaging in offline impression management (Flett & Hewitt, 2014; Sassaroli et al., 2008) - a tendency likely to manifest online in the curation of their social media profiles.

The reviewer highlighted a study by Loveys et al. (2017), which examines micropatterns of social media use related to the expression of emotions and affect in clinical vs non-clinical groups. We believe that the specific nature of the data (concerning emotional expression) does not provide a suitable foundation for informing our hypotheses. However, we will cite the study in the discussion section of the Stage 2 Registered Report, underscoring the importance and potential of examining objective social media data to examine differences between clinical vs non-clinical groups.

References

Flett, G. L., & Hewitt, P. L. (2014). Perfectionism and Perfectionistic Self-Presentation in Social Anxiety. In *Social Anxiety* (pp. 159–187). Elsevier.

Loveys, K., Crutchley, P., Wyatt, E., & Coppersmith, G. (2017, August). Small but mighty: affective micropatterns for quantifying mental health from social media language. In *Proceedings of the fourth workshop on computational linguistics and clinical Psychology—From linguistic signal to clinical reality* (pp. 85-95).

Sassaroli, S., Romero Lauro, L. J., Maria Ruggiero, G., Mauri, M. C., Vinai, P., & Frost, R. (2008). Perfectionism in depression, obsessive-compulsive disorder and eating disorders. *Behaviour Research and Therapy*, 46(6), 757–765.

R3.5) Greater context is needed around the discussion of self-disclosure. While authentic disclosure is generally thought to be helpful toward mental health, youth with psychiatric disorders also navigate presenting their authentic selves with considerations of disclosing about their emotional state and mental health vulnerabilities, which could be met by negative evaluations or bullying. Authentic self-disclosure within this population is not clear cut, and it seems warranted to recognize this nuance.

We agree that this nuanced topic requires explanatory care and have therefore added the following to the Revised Manuscript (Table 3, Page 48):

“Communicating personal and emotional information increases the risk of embarrassment and rejection (Omarzu, 2000) which is compounded by adolescents’ increased sensitivity to peer feedback and anxiety regarding negative social evaluations (van den Bos et al., 2014). We expect this to be especially difficult for adolescents with internalising disorders, as they must balance the rewards associated with self-disclosure (Vijayakumar et al., 2020) with considerations of how that disclosure might be received by their peers.”

References

Omarzu, J. (2000). A disclosure decision model: Determining how and when individuals will self-disclose. *Personality and Social Psychology Review*, 4, 174–185.

Van den Bos, E., de Rooij, M., Miers, A. C., Bokhorst, C. L., & Westenberg, P. M. (2014). Adolescents’ increasing stress response to social evaluation: Pubertal effects on cortisol and alpha-amylase during public speaking. *Child Development*, 85, 220-236.

Vijayakumar, N., Flournoy, J. C., Mills, K. L., Cheng, T. W., Mobasser, A., Flannery, J. E., ... & Pfeifer, J. H. (2020). Getting to know me better: An fMRI study of intimate and superficial self-disclosure to friends during adolescence. *Journal of Personality and Social Psychology*, 118(5), 885-899.

R3.6) It would help to understand more about the youth advisory board who aided in the development of qualitative measures of social media use. Were these youth who had psychiatric diagnoses, who could discuss the unique features of social media use for youth with mental health conditions?

The youth advisory board was conducted by the National Center of Social Research and included 10 young people aged 16-24 years old. Participants were identified through existing mental health research networks and came from across England, with a wide range of experiences and interests. They all shared a common interest in mental health, linked to either first-lived experiences or involvement in the field. We have now added this information to the Supplementary Methods (Page 2, Lines 12-16):

“The questions capturing social media engagement were developed through consultation with a youth advisory board conducted by the National Center of Social Research in 2015. The board included 10 young people aged 16-24 years old, that were identified through existing mental health research networks and came from across England, with a wide range of experiences and interests. They all shared a common interest in mental health, linked to either first-lived experiences or involvement in the field.”

R3.7) The addition of supplementary analyses for individual diagnoses or diagnostic categories seems to be a sizable addition that warrants greater description within the main paper. Particularly given high rates of comorbidity, the analyses related to mental health severity that assess whether the presence of multiple diagnoses relate to social media use factors seems quite important.

We have now added an additional section in our methods providing more details about the analyses concerning mental health severity (Revised Manuscript, Page 15, Lines 408-412):

“To examine whether social media use varies with mental health severity, we will conduct sensitivity analyses to assess whether the presence of multiple diagnoses is related to the outcomes of interest (a-g). First, we will plot responses to each social media question by the number of diagnoses. Next, we will test for a linear effect of the number of diagnoses on social media question response using linear regression models.”

Regarding the analyses for individual diagnoses, we decided to keep them in the Supplementary materials as exploratory analyses because of two reasons. First, our ability to conduct these analyses will be conditional on sample size and power. Second, we do not have specific hypotheses related to these questions and, therefore, we cannot test them in a confirmatory research framework as required by the Registered Report.

We greatly appreciate your reviews and look forward to your response.

Yours sincerely,

On behalf of all co-authors,

Luisa Fassi
PhD Researcher
MRC Cognition and Brain Sciences Unit,
Department of Psychiatry,
University of Cambridge

Dr Amanda Ferguson
Postdoctoral researcher
MRC Cognition and Brain Sciences,
University of Cambridge

Dr Amy Orben
Programme Leader Track Scientist
MRC Cognition and Brain Sciences Unit,
University of Cambridge

23rd October 2023

Dear Dr [REDACTED] [REDACTED]

Thank you for sending us the reviews for our manuscript “Social media use in adolescent clinical and non-clinical populations” (NATHUMBEHAV-22123378B). We are grateful for the feedback provided, which has allowed us to further improve the manuscript following three successful rounds of revisions. It is our hope that these improvements will enable our manuscript to be accepted as a Stage 1 Registered Report at Nature Human Behaviour without further delay. In line with this, we would welcome a (virtual) meeting to discuss any responses or changes we note below if you or the editorial team have further questions.

In this response, we address the Reviewer and Editor's comments and highlight the changes made. A revised version of the manuscript and supplementary materials are attached as separate documents.

Reviewer 4 comments

R4) I appreciated reviewing the paper once again, which is well-written and addresses a very important area of research. The response to comments was thoughtful, thorough, and informative. Best wishes on your future research!

We thank the reviewer for the overall appreciation of our work and are pleased that they judged our manuscript adequate for Stage I acceptance.

Editor's general comments

E1.1) Your revised manuscript has now been evaluated by original Reviewer 4, whose comments are included at the end of this letter. Although the reviewer finds your protocol to be of interest and signs off, we have some remaining editorial concerns about power, effect sizes and analytical choices, given that power cannot be known before the sample is accessed. We remain very interested in the possibility of proceeding further with your submission in Nature Human Behaviour but would like to consider your response to these concerns in the form of a revised manuscript before we make a decision on in-principle acceptance and Stage 2 submission. Specifically, we would encourage you to seek out assistance from an expert with Bayesian statistics, which could be used to either supplement or replace the current sampling approach. A Bayesian approach seems much more suitable when the N is fixed and actual power cannot be calculated. Attached to this decision letter, you can also find a marked-up copy of your manuscript. These edits are in order to make sure that your manuscript complies with our requirements for novelty and priority claims, as well as for qualitative characterizations of the work.

We thank the Editor for the comments on the Registered Report, which has allowed us to substantially improve our Stage 1 submission. Regarding the suggestion of adding Bayesian analyses, we have consulted with two statisticians, Dr Matti Vuorre (U. Tilburg) and Prof Daniel Lakens (TU. Eindhoven), both experts in Bayesian Statistics and Null Hypothesis Significance Testing (NHST), who have also reviewed this response letter. Our response, which is detailed in E2.4-E2.6, was directly informed by these consultations.

Editor's in-text comments

E2.1) Please follow this statement with an explanation of the key limitation of this work (i.e., it does not provide causal evidence) and explain what evidence instead it can provide.

We have now updated the manuscript as follows (green shading indicates the added sentences, Revised Manuscript, p. 5, lines 98-103):

“This Registered Report will provide data and evidence-based insights into how social media and mental health are related across clinical and non-clinical adolescent populations (see Table 1 for an overview). We note that, due to the cross-sectional nature of the data and planned analyses, the study results will not provide causal evidence. Hence, all reported coefficients indicate associations, with the possibility of bidirectional relationships and third variables affecting social media use, mental health or the relationship between the two.”

E2.2) Please provide references to those studies if in addition/other than 16 and 37.

We have now updated the reference list to include more studies that measure social media use by examining user appraisal (green shading indicates the added references, Revised Manuscript, p. 5, Line 116):

“Researchers have therefore called for quantitative time-based measures of social media use to be complemented by more qualitative engagement-based measures capturing adolescents' social media activities and their appraisal of them (e.g., Dumas et al., 2023; Frison & Eggermont, 2016; Nesi et al. 2017; Li et al., 2022; Odgers & Jensen, 2020; Verbeij et al., 2021.)”

The references cover the following studies:

1. Dumas et al (2023) measured pressure to gain social media attention with a 1-item question, namely: “how much pressure did you feel to get likes and views on your posted social media content over the past 3 weeks?”.
2. Frison & Eggermont (2016) measured perceptions of online social support using the Multidimensional Scale of Perceived Social Support.
3. Nesi et al. (2017) measured technology-based social comparison and feedback-seeking using the Motivations for Electronic Interaction Scale.
4. Li et al. (2022) measured emotional connection to online social networking using the Facebook Intensity Scale, replacing “Facebook” with “online social networking”.

E2.3) It is unclear whether here you mean ‘power sensitivity analyses’ or sensitivity analyses as indicated above. We ask that you commit to running power sensitivity analyses (as distinct from other sensitivity analyses registered elsewhere in the manuscript) and you specify how these power sensitivity analyses will be performed (please see, for example, Lakens, D. *Collabra Psychol.* 8, 33267 (2022)).

In addition to our revised power analyses (discussed in E2.4 below), we will report the results of power sensitivity analyses once the final sample size is known. Specifically, we will use the final sample size to estimate the smallest effect size that can be observed with 95% power (and $\alpha = 0.05$, i.e., power sensitivity analyses). We discuss this in the Revised Manuscript (p. 14, lines 383-385):

“In addition to our a-priori power calculations, we will conduct power sensitivity analyses once our final sample size is known. That is, we will determine the smallest effect size that is observable with 95% power, given an alpha of 0.05 and our final sample size.”

E2.4) Throughout this section, the effect sizes for which power is calculated will need to be not arbitrary, but chosen on the basis of existing literature on which each hypothesis is based and aiming for the smallest theoretically or practically/clinically meaningful effect size (not an arbitrary effect size). For each of the power calculations, you should state how likely it is that .95 power will be achieved given existing demographic information of prevalence of each population. It seems that for some hypotheses the tests will almost certainly be underpowered – if that is indeed the case, these hypotheses will need to be addressed as exploratory, not confirmatory.

We thank the editor for making this point, which has allowed us to substantially rethink and improve our approach to the smallest effect sizes of interest (SESOI), power and equivalence testing. We have revised our power calculations by first setting the SESOI and then solving for power, rather than just reporting the smallest effect size we have .95 power to detect. Below, we further describe the choice of both primary and secondary SESOIs (updated in the Supplementary Methods and Revised Manuscript), followed by the changes applied to the power calculations and equivalence testing (updated in the Revised Manuscript).

Choice of SESOI (Revised Supplementary Methods; p. 10-11, lines 83-164):

“Choice of the smallest effect size of interest

We chose to identify our smallest effect sizes of interest (SESOI) at different levels of explanation (i.e., theoretically, practically and clinically meaningful). Given the aim of our study to test the differences in social media use across clinical and non-clinical populations, we believe that no single SESOI provides a comprehensive answer as to what is a meaningful effect. Our research questions hold significance for a diverse range of stakeholders, including researchers, policymakers, the general public and clinicians, each requiring insights from a different level of explanation. For instance, with a theoretically meaningful SESOI as the benchmark, our findings can be contextualised in relation to the

broader field of mental health research. On the contrary, with practically and clinically meaningful SESOIs, our results can aid decision-making for policy and clinical practice.

Therefore, while a theoretically meaningful SESOI based on everyday behaviours and their link to mental health will serve as our primary SESOI for the power calculations and equivalence tests (to allow for a clear confirmatory approach), secondary SESOIs will play an important role in supporting the interpretation of our findings, which we plan to specifically address in a separate section of the Stage 2 results. Overall, this will ensure the applicability of our study to both academic and practical domains.

Primary SESOI

A theoretically meaningful SESOI serves as a way to assess the significance of a specific effect size in the context of a research question, using a theoretical framework as the reference point. After an extensive scoping exercise to identify a suitable theoretical foundation, we determined that the most relevant benchmarks for our research questions are everyday behaviours linked to mental health, such as sleep and physical activity. These behaviours, much like social media use, are a regular part of daily routines but, unlike the latter, they are well-established markers of psychopathology based on both theory (Crone et al., 2005; Knowlden et al., 2012; Rebar & Taylor, 2017) and matched empirical evidence (Mollayeva et al., 2016; Ruscio, 2008). Consequently, if the effect size of social media use is comparable to that observed for behaviours like sleep and physical activity, we can conclude that social media use also represents an everyday behaviour that exhibits meaningful group-level differences between clinical and non-clinical populations.

An alternative possibility that we considered to identify the theoretical SESOI is the use of social media literature. However, despite the small effect sizes that have made headlines and attracted public attention in this space (Burn-Murdoch, J., 2023; Haidt, J., 2023), there is still a lack of theoretical foundation as to what effects can be considered meaningful (Odgers & Jensen, 2020; Parry et al., 2021). This combined with a lack of studies that a) directly compare social media use in clinical vs non-clinical populations (see Fassi et al., 2023 for a review) and b) use comparable measures of engagement to the ones examined in our study, makes it currently impossible to adequately identify the theoretical SESOI of interest solely based on social media research. We therefore decided to draw from research on everyday behaviours other than social media use, as expanded on below.

In the case of sleep, research suggests a meta-analytic effect size of WMD = .48 (i.e., weighted mean difference), corresponding to $d = 0.413$ (Ruscio, 2008), for the difference between clinical and non-clinical populations in sleep duration (Mollayeva et al., 2016). In terms of physical exercise, a large ($N = 78,886$) nationally representative study (Okoro et al., 2005) suggests that, when compared to healthy individuals, individuals with mental health conditions are less likely to engage in recommended levels of physical activity, with a prevalence ratio = 0.84 after controlling for other health factors, corresponding to a medium effect size. Based on this evidence, we define $d = 0.4$ as the smallest theoretically meaningful effect size that would indicate a difference in social media use between clinical

and non-clinical populations at least as large as that found in well-defined behavioural markers of psychopathology. We use this as our primary SESOI, informing both power calculations and equivalence tests.

Secondary SESOIs

Given the relevance of our research question for different stakeholders (e.g., researchers, the public/policy sector, and clinicians), we also chose to identify SESOIs based on practical and clinical meaningfulness. This information will aid the interpretation of our findings that we plan to address in a separate section of the Stage 2 results named “Results into context: practical and clinical SESOI”.

Practically meaningful. A practically meaningful SESOI represents the level of change in a phenomenon that individuals are likely to subjectively notice. Setting a benchmark for practical significance helps researchers define thresholds for meaningful differences or effects that are relevant in real-world scenarios. Reviews of the literature suggest that, across various physical health conditions including pain, cardiovascular and respiratory disease, when patients are asked to subjectively identify a minimal change in symptomatology, their estimates consistently fall close to half a standard deviation, corresponding to $d = 0.5$ (Norman et al., 2003). This value approximates the range of human discrimination first identified by Miller (1956), who characterised the limit of people's abilities across a wide range of discrimination tasks as being equivalent to $d = 0.36-0.63$. Based on these grounds, $d = 0.5$ has been employed as a practically meaningful anchor of change in research on experimental design for treatment efficacy (Lipsey & Hurley, 2009), media effects (Ferguson, 2009; that used $d = .41$) and screen time (Przybylski et al., 2020). Therefore, we set $d = 0.5$ as the benchmark for the smallest practically meaningful effect size (i.e., a difference in social media use that an individual might subjectively notice).

Clinically meaningful. An effect is clinically meaningful to the extent that it can distinguish between clinical and non-clinical populations. For example, psychotherapy research suggests that clinically meaningful interventions are those that reduce psychological distress such that an individual with a mental health diagnosis - and therefore high levels of distress - will report non-clinical levels of distress following treatment (Jacobson, N. S., & Truax, P., 1992). To quantify the magnitude of change that is clinically meaningful, previous research has examined Health-Related Quality of Life (HRQoL) as a transdiagnostic index of distress across clinical and non-clinical populations (Guyatt et al., 1993). The HRQoL is a multi-dimensional measure that captures both the physical and mental health status of individuals as well as the overall impact that health status has on their quality of life. Studies consistently show that HRQoL differs between patients with and without mental health conditions. For instance, one meta-analysis found that patients with generalised anxiety disorder differed from controls by $d = 1.35$ in HRQoL (Olatunji et al., 2007). Similarly, studies of children and adolescents diagnosed with ADHD have found differences in HRQoL of $d = .72$ and $d = 1.05$ compared to controls (Hampel & Desman, 2006; Varni & Burwinkle, 2006). Given that reasonable SESOIs based on clinical significance are all larger than our established theoretical SESOI ($d = 0.4$), with a range of

$d = 0.72-1.35$, our study will be powered to detect clinically meaningful differences in social media use across clinical and non-clinical populations.”

Based on the new approach to the SESOI, we have updated the power calculations and equivalence tests in the Revised Manuscript, as detailed below (for the updates to Table 1, see response E2.7 and Appendix E.27).

Power calculations (Revised Manuscript, p. 14, Lines 358-384):

“Power calculations

We calculated power by setting the smallest effect size of interest (SESOI, $d = .4$, refer to Supplementary Methods for a detailed explanation), the alpha level to 0.05 and the estimated sample size. Regarding sample size, we estimated a total sample size of $N = 3,854$ based on existing MHCYP documentation (NHS Digital, 2017). For the equivalence tests, power was calculated using the *TOSTER* package in R (Caldwell, 2022; Lakens, 2018). For the regression models, power was determined using the *pwr* package (Champely, S., 2020) in R. The code for these calculations is available on the OSF (see https://osf.io/s2dwu/?view_only=5acced2ddb884f9481d439a7746f4dd1).

H0 (1). We calculated the exact power for equivalence testing to detect a SESOI of $d = 0.4$ given at least $N \geq 577$ individuals with a diagnosis and $N \geq 3,277$ individuals without a diagnosis. Results indicate 100% power to reject the presence of effects that are larger than $d = 0.4$.

H1 (1). We calculated power to detect a statistical effect of a diagnosis (yes/no) on social media responses using linear regression. Results indicate 100% power to detect the SESOI ($d=0.4$) with at least $N \geq 577$ individuals with a diagnosis and $N \geq 3,277$ individuals without a diagnosis.

H0 (2). We calculated the exact power for equivalence testing to detect a SESOI of $d = 0.4$ given at least $N \geq 370$ individuals with an internalising only diagnosis and $N \geq 3,277$ with no diagnosis. Results indicate 100% power to reject the presence of effects that are larger than $d = 0.4$.

H1 (2). We calculated power to detect a statistical effect of internalising diagnosis type (i.e., internalising vs no diagnosis) on social media responses using linear regression. Results indicate 100% power to detect the SESOI ($d=0.4$) with at least $N \geq 370$ individuals with an internalising-only diagnosis and $N \geq 3,277$ with no diagnosis. We calculated power to detect an effect of externalising diagnosis type (i.e., externalising vs no diagnosis) on social media engagement responses using linear regression. Results indicate 100% power to detect the SESOI ($d=0.4$) with at least $N \geq 199$ individuals with an externalising-only diagnosis and $N \geq 3,277$ with no diagnosis.

H0 (3). We calculated the exact power for equivalence testing to detect a SESOI of $d = 0.4$ given at least $N \geq 370$ individuals with internalising only and $N \geq 199$ with externalising only diagnoses. Results indicate 96% power to reject the presence of effects that are larger than $d = 0.4$.

H1 (3). We calculated the power to detect a statistical effect of internalising only vs externalising only diagnosis type on social media responses. Results indicate 99.8% power to the SESOI ($d=0.4$) with at least $N \geq 370$ individuals with internalising only and $N \geq 199$ with externalising only diagnoses.”

Equivalence testing (Revised Manuscript, p. 18, Lines 497-526):

“Equivalence testing

Given that null hypothesis significance testing does not allow a comprehensive interpretation of statistically non-significant results (Lakens, 2017), we will run equivalence tests to quantify support for the null hypotheses (H0, 1-3). This involves assessing whether the 90% confidence intervals for the effect size lie inside pre-specified equivalence bounds that indicate the SESOI. If the confidence intervals lie inside the equivalence bounds, the effect size is interpreted as negligible. On the contrary, if the confidence intervals spread outside the equivalence bounds, the effect size is considered meaningful in size. In this case, the 90% confidence intervals are used rather than 95% because the effect size is tested against two equivalence bounds separately (i.e., the upper and lower bound), reflecting $(1 - 2\alpha) \times 100\%$.

We established the equivalence bounds by identifying a theoretically meaningful SESOI ($d = 0.4$ refer to Supplementary Methods for a detailed explanation). After an extensive scoping exercise to identify a suitable theoretical foundation, we determined that the most relevant benchmark for our research questions are everyday behaviours linked to mental health, such as sleep and physical activity. These behaviours, much like social media use, are a regular part of daily routines but, unlike the latter, they are well-established markers of psychopathology, based on both theory (Crone et al., 2005; Knowlden et al., 2012; Rebar & Taylor, 2017) and matched empirical evidence (Mollayeva et al., 2016; Ruscio, 2008). Consequently, if the actual effect size of social media use is comparable to that observed for behaviours like sleep and physical activity, we can confidently conclude that social media use also represents an everyday behaviour that exhibits meaningful group-level differences between clinical and non-clinical populations. The interpretation and analysis plan are presented separately for each hypothesis in Table 1.

Overall, we will infer support for the null hypotheses (i.e., no meaningful difference in social media use between groups) if the 90% confidence interval for this association lies within the equivalence bounds. Of note, while a theoretical SESOI based on everyday behaviours and their link to mental health will serve as our primary effect size of interest to allow for a clear confirmatory approach, we also identified secondary SESOIs based on the effect sizes that are practically and clinically meaningful. These secondary SESOIs will play an important role in supporting the interpretation of our findings (as detailed in

Supplementary Methods, p. 6-10), ensuring the applicability of our study to both academic and practical domains.”

E2.5) We strongly encourage you to also preregister Bayesian statistical analyses for each of your hypotheses. This will enable you to draw meaningful conclusions for confirmatory analyses when power is not met but effect sizes are large, as well as to interpret statistically not significant findings in a meaningful way.

Regarding the potential benefits of Bayesian analyses, we have consulted with two statisticians, Dr. Matti Vuorre and Prof. Daniel Lakens to improve this aspect of our study. Specifically, we have verified that our current methodology, which employs equivalence testing, is well suited to achieve the two goals highlighted by the Editor, namely 1) dealing with large effect sizes and low power, and 2) interpreting statistically non-significant findings in a meaningful way.

Regarding the first point, given the updated power analyses and SESOI, we now articulate a clear argument legitimating why the analyses we are using are powered to detect effect sizes that are theoretically, practically and clinically meaningful, as specified in E2.4. We have addressed many of the challenges highlighted by the editor through this change. Further, even if we are in a situation of relatively low power and large effect sizes, equivalence testing will allow us to address concerns about interpreting non-significant findings. Specifically, when effect sizes are large, this method is more likely to detect statistical significance, as a larger effect size is more likely to fall within the equivalence margin. Further, equivalence testing is less prone to false positives (Type I errors) compared to NHST in the absence of equivalence testing, reducing the risk of erroneously concluding significance when power is low (Lakens et al. 2018). Therefore, equivalence testing applied to all hypotheses in the registered report will allow us to draw meaningful conclusions from our confirmatory analyses.

Regarding the second point (i.e., interpretation of non-significant findings), we note that equivalence testing is a suitable approach as it explicitly addresses the question of meaningful significance (Lakens et al., 2020). Namely, it reverses the type of question that is asked in NHST: instead of examining whether we can reject an effect of zero, it tests whether we can reject effects that are as or more extreme than the equivalence bounds based on the SESOI. The equivalence bounds, therefore, correspond to the null hypothesis in a one-sided test, while the region around zero reflects the alternative hypothesis. Equivalence testing assesses whether there is enough evidence to conclude that the effect is equivalent to a meaningful benchmark, this approach allows for interpretation of non-significant results.

In order to maximize the interpretability of our confirmatory findings and avoid overcomplicating our Registered Report with redundant analyses, we (as well as our statistical consultants Dr Vuorre and Professor Lakens) therefore suggest not including Bayesian statistical analyses.

E2.6) We strongly encourage to either replace or supplement equivalence testing with BF's. The reason for this is that actual power is not known because your actual sample sizes for each comparison is not known. This will render it impossible to draw meaningful conclusions from null results.

We thank the Editor for this suggestion. We have consulted with two statisticians, Dr Matti Vourre and Prof Daniel Lakens, both experts in Bayesian statistics and equivalence testing, who agreed in light of our changes in E2.4 the use of Bayesian factors will not be necessary. We highlight two points in response.

First, despite taking two different approaches to defining probability and incorporating prior information, the use of Bayesian factors and equivalence tests typically lead to converging answers. Lakens et al. (2020) directly compared these methods by re-analysing four independent studies and showed agreement in all cases, suggesting that the addition of Bayes factors in our study will not assist in interpretation beyond what is already offered by equivalence testing in our planned analyses.

Second, we note that our estimates of the sample size are based on a conservative reading of existing documentation of the dataset (MHCYP, 2017) and should therefore not substantially deviate from the final sample size. The current power analyses show at least .99 power to detect the theoretical, practical and clinical SESOI for all our hypotheses. Hence, even in the scenario of smaller sample sizes than expected, there is a negligible risk that we will have lower than .95 power. For instance, in our first research question that tests differences in social media use in adolescents with versus without a mental health diagnosis, even in the unrealistic scenario that we lose 2000 people from the no-diagnosis group and 500 from the diagnosis group, we would still have 96% power to detect our effect size of interest.

E2.7) Please make sure to revise the tables according to the guidance above. Please also include information on power sensitivity analyses in the table, given that actual power is not known for any of these comparisons.

In addition to updating the power calculations and equivalence testing as outlined in response E2.4, we also updated Table 1 and the Analysis Plan section to reflect the changes made. We attach Table 1 in Appendix E2.7 below (Revised Manuscript, p. 33-43). Further, we updated the Analysis Plan in the Revised Manuscript (p. 18, Lines 497-500):

“Overall, we will infer support for the alternative hypotheses if 1) the coefficient for the association of social media use and our grouping variable for the mental health diagnosis is significant, and 2) the association follows the hypothesised direction. The interpretation and analysis plan are presented separately for each hypothesis in Table 1. Further, we detail the planned exploratory analyses in Supplementary Exploratory Analyses.”

We greatly appreciate your reviews and we hope you find our work suitably revised. We welcome the opportunity to discuss any response or suggested changes in a virtual meeting if necessary.

Yours sincerely,

On behalf of all co-authors,

Luisa Fassi
PhD Researcher
MRC Cognition and Brain Sciences Unit,
Department of Psychiatry,
University of Cambridge

Dr Amanda Ferguson
Postdoctoral Researcher
MRC Cognition and Brain Sciences,
University of Cambridge

Dr Amy Orben
Programme Leader Track Scientist
MRC Cognition and Brain Sciences Unit,
University of Cambridge

Dr Andrew K. Przybylski
Professor of human behaviour and technology
Oxford Internet Institute
University of Oxford

References

- Burn-Murdoch, J. (2023). *Smartphones and social media are destroying children's mental health*. *Financial Times*. <https://www.ft.com/content/0e2f6f8e-bb03-4fa7-8864-f48f576167d2>
- Caldwell, A. R. (2022). *Exploring Equivalence Testing with the Updated TOSTER R Package* [Preprint]. PsyArXiv. <https://doi.org/10.31234/osf.io/ty8de>
- Champely, S. (2020). *Pwr package*. <https://github.com/heliosdrm/pwr>
- Crone, D., Smith, A., & Gough, B. (2005). 'I feel totally at one, totally alive and totally happy': A psycho-social explanation of the physical activity and mental health relationship. *Health Education Research*, 20(5), 600–611. <https://doi.org/10.1093/her/cyh007>
- Dumas, T. M., Tremblay, P. F., Ellis, W., Millett, G., & Maxwell-Smith, M. A. (2023). Does pressure to gain social media attention have consequences for adolescents' friendship closeness and mental health? A longitudinal examination of within-person cross-lagged relations. *Computers in Human Behavior*, 140, 107591. <https://doi.org/10.1016/j.chb.2022.107591>
- Fassi, L., Thomas, K., Parry, D. A., Leyland-Craggs, A., Ford, T. J., & Orben, A. (2023). *Social media use and internalising symptoms in clinical and community adolescent samples: A systematic review and meta-analysis* [Preprint]. *Psychiatry and Clinical Psychology*. <https://doi.org/10.1101/2023.09.12.23295355>
- Ferguson, C. J. (2009). An effect size primer: A guide for clinicians and researchers. *Professional Psychology: Research and Practice*, 40(5), 532–538. <https://doi.org/10.1037/a0015808>
- Frison, E., & Eggermont, S. (2016). Exploring the Relationships Between Different Types of Facebook Use, Perceived Online Social Support, and Adolescents' Depressed Mood. *Social Science Computer Review*, 34(2), 153–171. <https://doi.org/10.1177/0894439314567449>
- Guyatt, G.H., Feeny, P., & Patrick, D.L. (1993). *Measuring Health-Related Quality of Life*. <https://doi.org/10.7326/0003-4819-118-8-199304150-00009>
- Haidt, J. (2023). *Social media is a major cause of the mental illness epidemic in teen girls. Here's the evidence*. *After Babel*. <https://jonathanhaidt.substack.com/p/social-media-mental-illness-epidemic>
- Hampel, P., & Desman, C. (2006). Coping and quality of life among children and adolescents with attention deficit/hyperactivity disorder. *Praxis Der Kinderpsychologie Und Kinderpsychiatrie*, 55(6), 425–443.
- Jacobson, N. S., & Truax, P. (1992). *Clinical significance: A statistical approach to defining meaningful change in psychotherapy research*. <https://psycnet.apa.org/doi/10.1037/10109-042>
- Knowlden, A. P., Sharma, M., & Bernard, A. L. (2012). A Theory of Planned Behavior Research Model for Predicting the Sleep Intentions and Behaviors of Undergraduate College Students. *The Journal of Primary Prevention*, 33(1), 19–31. <https://doi.org/10.1007/s10935-012-0263-2>
- Lakens, D. (2022a). *Improving Your Statistical Inferences*. <https://lakens.github.io/statistical-inferences/>
- Lakens, D. (2022b). Sample Size Justification. *Collabra: Psychology*, 8(1), 33267. <https://doi.org/10.1525/collabra.33267>
- Lakens, D., McLatchie, N., Isager, P. M., Scheel, A. M., & Dienes, Z. (2020). Improving Inferences About Null Effects With Bayes Factors and Equivalence Tests. *The Journals of Gerontology: Series B*, 75(1), 45–57. <https://doi.org/10.1093/geronb/gby065>
- Lakens, D., Scheel, A. M., & Isager, P. M. (2018). Equivalence Testing for Psychological Research: A Tutorial. *Advances in Methods and Practices in Psychological Science*, 1(2), 259–269. <https://doi.org/10.1177/2515245918770963>
- Li, J.-B., Lau, J. T. F., Feng, L.-F., Zhang, X., Li, J.-H., Mai, J.-C., Chen, Y.-X., & Mo, P. K. H. (2022). Associations of intensity and emotional connection related to online social networking use on the risk of incident depression among Chinese adolescents: A prospective cohort study. *Journal of Affective Disorders*, 308, 116–122. <https://doi.org/10.1016/j.jad.2022.04.029>

- Lipsey, M., & Hurley, S. (2009). Design Sensitivity: Statistical Power for Applied Experimental Research. In L. Bickman & D. Rog, *The SAGE Handbook of Applied Social Research Methods* (pp. 44–76). SAGE Publications, Inc.
<https://doi.org/10.4135/9781483348858.n2>
- NHS Digital. *MHCYP 2017*. <https://digital.nhs.uk/data-and-information/publications/statistical/mental-health-of-children-and-young-people-in-england/2017/2017>
- Miller, G. A. (1956). The magical number seven plus or minus two: Some limits on our capacity for processing information. *Psychological Review*, *63*(2), 81–97.
- Mollayeva, T., Thurairajah, P., Burton, K., Mollayeva, S., Shapiro, C. M., & Colantonio, A. (2016). The Pittsburgh sleep quality index as a screening tool for sleep dysfunction in clinical and non-clinical samples: A systematic review and meta-analysis. *Sleep Medicine Reviews*, *25*, 52–73. <https://doi.org/10.1016/j.smrv.2015.01.009>
- Nesi, J., Miller, A. B., & Prinstein, M. J. (2017). Adolescents' depressive symptoms and subsequent technology-based interpersonal behaviors: A multi-wave study. *Journal of Applied Developmental Psychology*, *51*, 12–19. <https://doi.org/10.1016/j.appdev.2017.02.002>
- Norman, G. R., Sloan, J. A., & Wyrwich, K. W. (2003). Interpretation of Changes in Health-related Quality of Life: The Remarkable Universality of Half a Standard Deviation. *Medical Care*, *41*(5), 582–592. <https://doi.org/10.1097/01.MLR.0000062554.74615.4C>
- Odgers, C. L., & Jensen, M. R. (2020). Annual Research Review: Adolescent mental health in the digital age: facts, fears, and future directions. *Journal of Child Psychology and Psychiatry*, *61*(3), 336–348. <https://doi.org/10.1111/jcpp.13190>
- Okoro, C. A., Stoodt, G., Rohrer, J. E., Strine, T. W., Li, C., & Balluz, L. S. (2014). Physical Activity Patterns among U.S. Adults with and without Serious Psychological Distress. *Public Health Reports*, *129*(1), 30–38. <https://doi.org/10.1177/003335491412900106>
- Olatunji, B. O., Cisler, J. M., & Tolin, D. F. (2007). Quality of life in the anxiety disorders: A meta-analytic review. *Clinical Psychology Review*, *27*(5), 572–581.
<https://doi.org/10.1016/j.cpr.2007.01.015>
- Parry, D. A., Fisher, J. T., Mieczkowski, H., Sewall, C. J. R., & Davidson, B. I. (2022). Social media and well-being: A methodological perspective. *Current Opinion in Psychology*, *45*, 101285. <https://doi.org/10.1016/j.copsyc.2021.11.005>
- Przybylski, A. K., Orben, A., & Weinstein, N. (2020). How Much Is Too Much? Examining the Relationship Between Digital Screen Engagement and Psychosocial Functioning in a Confirmatory Cohort Study. *Journal of the American Academy of Child & Adolescent Psychiatry*, *59*(9), 1080–1088. <https://doi.org/10.1016/j.jaac.2019.06.017>
- Rebar, A. L., & Taylor, A. (2017). Physical activity and mental health; it is more than just a prescription. *Mental Health and Physical Activity*, *13*, 77–82.
<https://doi.org/10.1016/j.mhpa.2017.10.004>
- Ruscio, J. (2008). A probability-based measure of effect size: Robustness to base rates and other factors. *Psychological Methods*, *13*(1), 19–30. <https://doi.org/10.1037/1082-989X.13.1.19>
- Varni, J. W., & Burwinkle, T. M. (2006). The PedsQL™ as a patient-reported outcome in children and adolescents with Attention-Deficit/Hyperactivity Disorder: A population-based study. *Health and Quality of Life Outcomes*, *4*(1), 26. <https://doi.org/10.1186/1477-7525-4-26>
- Verbeij, T., Pouwels, J. L., Beyens, I., & Valkenburg, P. M. (2021). The accuracy and validity of self-reported social media use measures among adolescents. *Computers in Human Behavior Reports*, *3*, 100090. <https://doi.org/10.1016/j.chbr.2021.100090>

Appendix E2.7.

Table 1. Design Table summarising the study's research questions, hypotheses, power calculations, analyses, and conditions for interpretation.

Question	Hypothesis	Power analysis	Analysis plan	Interpretation given to different outcomes
1) Investigate whether adolescents with any mental health condition use social media differently than those without a condition.	Hypothesis 1.1) Adolescents with any mental health diagnosis will score higher than adolescents without a diagnosis in a) time spent on social media, b) online social comparison, c) lack of control over time spent online, d) monitoring of online feedback and e) feeling impacted by online feedback.	We calculated power given a theoretical SESOI of $d = 0.4$, and a sample size of at least $N \geq 577$ individuals with a diagnosis and $N \geq 3,277$ individuals without a diagnosis. Results indicated 100% power to reject effects larger than the SESOI using the <code>'power_t_TOST'</code> function for equivalence testing and 100% power to detect the SESOI in the hypothesised direction using the <code>'pwr.t2n.test'</code> function in R.	Five linear regression models will test the association between social media use (time- and engagement-based dimensions; a-e) with mental health diagnosis (two levels: no diagnosis, diagnosis), with the no diagnosis group set as the reference level. We will employ equivalence tests to assess whether the 90% confidence interval (CI) for the association between social media use and mental health diagnosis lies between the equivalence bounds, -0.4 and 0.4. We will employ regression models to assess whether the association between social media use and mental health diagnosis follows the hypothesised direction.	 • If the CI for this association is within the equivalence bounds, it will suggest that adolescents with any mental health diagnosis do not have a meaningful difference in their social media use compared to adolescents without a diagnosis. On the contrary, if the CIs do not fall within the equivalence bounds, it will suggest that the association is of meaningful magnitude. • A positive and statistically significant association between social media use and mental health diagnosis will suggest that adolescents with any mental health diagnosis score higher on the examined dimension of social media use (a-e) compared to adolescents without a diagnosis. • In summary, Hypothesis 1.1 will be supported if a) the CIs for this association do not fall within the equivalence bounds, b) the association between social media use and mental health diagnosis is statistically significant, and c) the regression coefficient reflecting this association is positive.

1) Investigate whether adolescents with any mental health condition use social media differently than those without a condition.	Hypothesis 1.2) Adolescents with any mental health diagnosis will score lower than adolescents without a diagnosis in f) happiness about the number of online friendships, g) honest online self-disclosure, and h) authentic self-presentation online.	We calculated power given a theoretical SESOI of $d = 0.4$, and a sample size of at least $N \geq 577$ individuals with a diagnosis and $N \geq 3,277$ individuals without a diagnosis. Results indicated 100% power to reject effects larger than the SESOI using the <code>'power.t.TOST'</code> function for equivalence testing and 100% power to detect the SESOI in the hypothesised direction using the <code>'pwr.t2n.test'</code> function in R.	Three linear regression models will test the association between each dimension of social media engagement (f-h) and mental health diagnosis (two levels: no diagnosis, diagnosis), with the no diagnosis group set as the reference level. We will employ equivalence tests to assess whether the 90% confidence interval (CI) for the association between social media use and mental health diagnosis lies between the equivalence bounds, -0.4 and 0.4. We will employ regression models to assess whether the association between social media use and mental health diagnosis follows the hypothesised direction.	 • If the CI for this association is within the equivalence bounds, it will suggest that adolescents with any mental health diagnosis do not have a meaningful difference in their social media use compared to adolescents without a diagnosis. On the contrary, if the CIs do not fall within the equivalence bounds, it will suggest that the association is of meaningful magnitude. • A negative and statistically significant association between social media engagement and mental health diagnosis will suggest that adolescents with any mental health diagnosis score lower on the examined dimension of social media engagement (f-h) compared to adolescents without a diagnosis. • In summary, Hypothesis 1.2 will be supported if a) the CIs for this association do not fall within the equivalence bounds, b) the association between social media use and mental health diagnosis is statistically significant, and c) the regression coefficient reflecting this association is negative.
--	--	---	--	---

2) Investigate whether adolescents with an internalising or externalising condition use social media differently than those without a condition.	Hypothesis 2.0) Adolescents with internalising mental health diagnoses will not differ from adolescents without a diagnosis in c) lack of control over time spent online.	We calculated power given a theoretical SESOI of $d = 0.4$, and a sample size of at least $N \geq 370$ individuals with internalising only diagnoses and $N \geq 3,277$ with no diagnosis. Results indicated 100% power to reject effects larger than the SESOI using the <code>'power.t.TOST'</code> function for equivalence testing and 100% power to detect the SESOI in the hypothesised direction using the <code>'pwr.t2n.test'</code> function in R.	One linear regression model will test the association between social media engagement (c) and diagnostic category (three levels: no diagnosis, internalising, externalising), with the no diagnosis group set as the reference level. For this hypothesis, we will examine the regression coefficient for externalising vs no diagnosis. We will employ an equivalence test to assess whether the 90% confidence interval (CI) for the association between social media use and internalising diagnosis lies between the equivalence bounds, -0.4 and 0.4.	 • If the CI for this association is within the equivalence bounds, it will suggest that adolescents with an internalising diagnosis do not have a meaningful difference in their social media use compared to adolescents without a diagnosis. On the contrary, if the CIs do not fall within the equivalence bounds, it will suggest that the association is of meaningful magnitude. • A non-significant association between social media use and diagnostic category (comparing internalising and no diagnosis) will suggest that adolescents with internalising diagnoses do not differ from adolescents without a diagnosis on lack of control over time spent online (c). • In summary, Hypothesis 2.0 will be supported if a) the CIs for this association fall within the equivalence bounds, and b) the association between social media use and diagnostic category (comparing internalising and no diagnosis) is not statistically significant.
--	--	--	--	--

2) Investigate whether adolescents with an internalising or externalising condition use social media differently than those without a condition.	Hypothesis 2.0) Adolescents with externalising mental health diagnoses will not differ from adolescents without a diagnosis in b) online social comparison, d) monitoring of online feedback and e) feeling impacted by online feedback, g) honest online self-disclosure, and h) authentic self-presentation online.	We calculated power given a theoretical SESOI of $d = 0.4$, and a sample size of at least $N \geq 199$ individuals with externalising only diagnoses and $N \geq 3,277$ with no diagnosis. Results indicated 100% power to reject effects larger than the SESOI using the <code>'power.t.TOST'</code> function for equivalence testing and 100% power to detect the SESOI in the hypothesised direction using the <code>'pwr.t2n.test'</code> function in R.	Five linear regression models will test the association between social media use (b, d, e, g, h) and diagnostic category (three levels: no diagnosis, internalising, externalising), with the no diagnosis group set as the reference level. For this hypothesis, we will examine the regression coefficient for externalising vs no diagnosis. We will employ an equivalence test to assess whether the 90% confidence interval (CI) for the association between social media use and mental health diagnosis lies between the equivalence bounds, -0.4 and 0.4.	 • If the CI for this association is within the equivalence bounds, it will suggest that adolescents with an externalising diagnosis do not have a meaningful difference in their social media use compared to adolescents without a diagnosis. On the contrary, if the CIs do not fall within the equivalence bounds, it will suggest that the association is of meaningful magnitude. • A non-significant association between social media use and diagnostic category (comparing externalising and no diagnosis) will suggest that adolescents with externalising diagnoses do not differ from adolescents without a diagnosis on the considered dimension of social media engagement (b, d, e, g, h). • In summary, Hypothesis 2.0 will be supported if a) the CIs for this association fall within the equivalence bounds, and b) the association between social media use and diagnostic category (comparing externalising and no diagnosis) is not statistically significant.
--	--	--	---	---

2) Investigate whether adolescents with an internalising or externalising condition use social media differently than those without a condition.	Hypothesis 2.1) Adolescents with internalising mental health diagnoses will score higher than adolescents without a diagnosis in a) time spent on social media, b) online social comparison, d) monitoring of online feedback and e) feeling impacted by online feedback.	We calculated power given a theoretical SESOI of $d = 0.4$, and a sample size of at least $N \geq 370$ individuals with internalising only diagnoses and $N \geq 3,277$ with no diagnosis. Results indicated 100% power to reject effects larger than the SESOI using the <code>'power.t.TOST'</code> function for equivalence testing and 100% power to detect the SESOI in the hypothesised direction using the <code>'pwr.t2n.test'</code> function in R.	Four linear regression models will test the association between social media use (time- and engagement-based dimensions; a, b, d, e) and diagnostic category (three levels: no diagnosis, internalising, externalising), with the no diagnosis group set as the reference level. For this hypothesis, we will examine the regression coefficient for internalising vs no diagnosis. We will employ an equivalence test to assess whether the 90% confidence interval (CI) for the association between social media use and mental health diagnosis lies between the equivalence bounds, -0.4 and 0.4. We will employ regression models to assess whether the association between social media use and diagnostic category (comparing internalising and no diagnosis) follows the hypothesised direction.	 • If the CI for this association is within the equivalence bounds, it will suggest that adolescents with an internalising diagnosis do not have a meaningful difference in their social media use compared to adolescents without a diagnosis. On the contrary, if the CIs do not fall within the equivalence bounds, it will suggest that the association is of meaningful magnitude. • A positive and statistically significant association between social media engagement and diagnostic category (comparing internalising and no diagnosis) will suggest that adolescents with an internalising diagnosis score higher on the examined dimension of social media engagement (a, b, d, e) compared to adolescents without a diagnosis. • In summary, Hypothesis 2.1 will be supported if a) the CIs for this association do not fall within the equivalence bounds, b) the association between social media use and diagnostic category (comparing internalising and no diagnosis) is statistically significant, and c) the regression coefficient reflecting this association is positive.
--	--	--	---	---

2) Investigate whether adolescents with an internalising or externalising condition use social media differently than those without a condition.	Hypothesis 2.2) Adolescents with internalising mental health diagnoses will score lower than adolescents without a diagnosis in f) happiness about online friendships, g) honest online self-disclosure and h) authentic self-presentation online.	We calculated power given a theoretical SESOI of $d = 0.4$, and a sample size of at least $N \geq 370$ individuals with internalising only diagnosis and $N \geq 3,277$ with no diagnosis. Results indicated 100% power to reject effects larger than the SESOI using the <code>'power.t.TOST'</code> function for equivalence testing and 100% power to detect the SESOI in the hypothesised direction using the <code>'pwr.t2n.test'</code> function in R.	Three linear regression models will test the association between each dimension of social media engagement (f, g, h) and diagnostic category (three levels: no diagnosis, internalising, externalising), with the no diagnosis group set as the reference level. For this hypothesis, we will examine the regression coefficient for internalising vs no diagnosis. We will employ an equivalence test to assess whether the 90% confidence interval (CI) for the association between social media use and mental health diagnosis lies between the equivalence bounds, -0.4 and 0.4. We will employ regression models to assess whether the association between social media use and diagnostic category (comparing internalising and no diagnosis) follows the hypothesised direction.	 • If the CI for this association is within the equivalence bounds, it will suggest that adolescents with an internalising diagnosis do not have a meaningful difference in their social media use compared to adolescents without a diagnosis. On the contrary, if the CIs do not fall within the equivalence bounds, it will suggest that the association is of meaningful magnitude. • A negative and statistically significant association between social media engagement and diagnostic category (comparing internalising and no diagnosis) will suggest that adolescents with an internalising diagnosis score lower on the examined dimension of social media engagement (f, g, h) compared to adolescents without a diagnosis. • In summary, Hypothesis 2.2 will be supported if a) the CIs for this association do not fall within the equivalence bounds, b) the association between social media use and diagnostic category (comparing internalising and no diagnosis) is statistically significant, and c) the regression coefficient reflecting this association is negative.
--	---	--	---	---

2) Investigate whether adolescents with an internalising or externalising condition use social media differently than those without a condition.	Hypothesis 2.3) Adolescents with externalising mental health diagnoses will score higher than adolescents without a diagnosis in a) time spent on social media and c) lack of control over time spent online.	We calculated power given a theoretical SESOI of $d = 0.4$, and a sample size of at least $N \geq 199$ individuals with externalising only diagnoses and $N \geq 3,277$ with no diagnosis. Results indicated 100% power to reject effects larger than the SESOI using the <code>'power.t.TOST'</code> function for equivalence testing and 100% power to detect the SESOI in the hypothesised direction using the <code>'pwr.t2n.test'</code> function in R.	Two linear regression models will test the association between each dimension of social media use and diagnostic category (three levels: no diagnosis, internalising, externalising), with the no diagnosis group set as the reference level. For this hypothesis, we will examine the regression coefficient for externalising vs no diagnosis. We will employ an equivalence test to assess whether the 90% confidence interval (CI) for the association between social media use and mental health diagnosis lies between the equivalence bounds, -0.4 and 0.4. We will employ regression models to assess whether the association between social media use and diagnostic category (comparing externalising and no diagnosis) follows the hypothesised direction.	 • If the CI for this association is within the equivalence bounds, it will suggest that adolescents with externalising only diagnoses do not have a meaningful difference in their social media use compared to adolescents without a diagnosis. On the contrary, if the CIs do not fall within the equivalence bounds, it will suggest that the association is of meaningful magnitude. • A positive and statistically significant association between social media engagement and diagnostic category (comparing externalising and no diagnosis) will suggest that adolescents with externalising only diagnoses score higher on the examined dimension of social media engagement (a, c) compared to adolescents without a diagnosis. • In summary, Hypothesis 2.3 will be supported if a) the CIs for this association do not fall within the equivalence bounds, b) the association between social media use and diagnostic category (comparing externalising and no diagnosis) is statistically significant, and c) the regression coefficient reflecting this association is positive.
--	--	--	--	---

2) Investigate whether adolescents with an internalising or externalising condition use social media differently than those without a condition.	Hypothesis 2.4) Adolescents with externalising diagnoses will score lower than adolescents without a diagnosis in f) happiness about the number of online friendships.	We calculated power given a theoretical SESOI of $d = 0.4$, and a sample size of at least $N \geq 199$ individuals with externalising only diagnoses and $N \geq 3,277$ with no diagnosis. Results indicated 100% power to reject effects larger than the SESOI using the <code>'power.t.TOST'</code> function for equivalence testing and 100% power to detect the SESOI in the hypothesised direction using the <code>'pwr.t2n.test'</code> function in R.	One linear regression model will test the association between happiness about the number of online friends and diagnostic category (three levels: no diagnosis, internalising, externalising), with the no diagnosis group set as the reference level. For this hypothesis, we will examine the regression coefficient for externalising vs no diagnosis. We will employ an equivalence test to assess whether the 90% confidence interval (CI) for the association between social media use and mental health diagnosis lies between the equivalence bounds, -0.4 and 0.4. We will employ regression models to assess whether the association between social media use and diagnostic category (comparing externalising and no diagnosis) follows the hypothesised direction.	 • If the CI for this association is within the equivalence bounds, it will suggest that adolescents with externalising only diagnoses do not have a meaningful difference in their social media use compared to adolescents without a diagnosis. On the contrary, if the CIs do not fall within the equivalence bounds, it will suggest that the association is of meaningful magnitude. • A negative and statistically significant association between social media engagement and diagnostic category (comparing externalising and no diagnosis) will suggest that adolescents with externalising only diagnoses score lower on the examined dimension of social media engagement (f) compared to adolescents without a diagnosis. • In summary, Hypothesis 2.4 will be supported if a) the CIs for this association do not fall within the equivalence bounds, b) the association between social media use and diagnostic category (comparing externalising and no diagnosis) is statistically significant, b) the regression coefficient reflecting this association is negative.
--	---	--	---	---

3) Investigate whether adolescents with an internalising condition use social media differently than those with an externalising condition.	Hypothesis 3.0) Adolescents with internalising diagnoses will not differ from adolescents with an externalising diagnoses in a) time spent on social media and f) happiness about the number of online friendships.	We calculated power given a theoretical SESOI of $d = 0.4$, and a sample size of at least $N \geq 370$ individuals with internalising only and $N \geq 199$ with externalising only diagnoses. Results indicated 96% power to reject effects larger than the SESOI using the <code>'power.t.TOST'</code> function for equivalence testing and 99.8% power to detect the SESOI in the hypothesised direction using the <code>'pwr.t2n.test'</code> function in R.	Two linear regression models will test the association between each dimension of social media engagement and diagnostic category (three levels: no diagnosis, internalising, externalising), with the internalising diagnosis group set as the reference level. For this hypothesis, we will examine the regression coefficient for internalising vs externalising. We will employ an equivalence test to assess whether the 90% confidence interval (CI) for the association between social media use and mental health diagnosis lies between the equivalence bounds, -0.4 and 0.4.	 • If the CI for this association is within the equivalence bounds, it will suggest that adolescents with externalising diagnoses do not have a meaningful difference in their social media use compared to adolescents with internalising diagnoses. On the contrary, if the CIs do not fall within the equivalence bounds, it will suggest that the association is of meaningful magnitude. • A non-significant association between social media use and diagnostic category (comparing internalising and externalising) will suggest that adolescents with internalising diagnoses do not differ from adolescents with externalising diagnoses on the considered dimension of social media engagement (a, f). • In summary, Hypothesis 3.0 will be supported if a) the CIs for this association fall within the equivalence bounds, and b) the association between social media use and diagnostic category (comparing internalising and externalising) is not statistically significant.
---	--	--	---	---

3) Investigate whether adolescents with an internalising condition use social media differently than those with an externalising condition.	Hypothesis 3.1) Adolescents with internalising diagnoses will score higher than adolescents with externalising diagnoses in b) online social comparison, d) monitoring of online feedback, and e) feeling impacted by online feedback.	We calculated power given a theoretical SESOI of $d = 0.4$, and a sample size of at least $N \geq 370$ individuals with internalising only and $N \geq 199$ with externalising only diagnoses. Results indicated 96% power to reject effects larger than the SESOI using the 'power.t.TOST' function for equivalence testing and 99.8% power to detect the SESOI in the hypothesised direction using the 'pwr.t2n.test' function in R.	Three linear regression models will test the association between each dimension of social media engagement and diagnostic category (three levels: no diagnosis, internalising, externalising), with the externalising diagnosis group set as the reference level. For this hypothesis, we will examine the regression coefficient for internalising vs externalising. We will employ an equivalence test to assess whether the 90% confidence interval (CI) for the association between social media use and mental health diagnosis lies between the equivalence bounds, -0.4 and 0.4. We will employ regression models to assess whether the association between social media use and diagnostic category (comparing internalising and externalising) follows the hypothesised direction.	 • If the CI for this association is within the equivalence bounds, it will suggest that adolescents with internalising only diagnoses do not have a meaningful difference in their social media use compared to adolescents with externalising only diagnoses. On the contrary, if the CIs do not fall within the equivalence bounds, it will suggest that the association is of meaningful magnitude. • A positive and statistically significant association between social media engagement and diagnostic category (comparing internalising and externalising) will suggest that adolescents with internalising only diagnoses score higher on the examined dimension of social media engagement (b, d, e) compared to adolescents with externalising only diagnoses. • In summary, Hypothesis 3.1 will be supported if a) the CIs for this association do not fall within the equivalence bounds, b) the association between social media use and diagnostic category (comparing internalising and externalising) is statistically significant, and c) the regression coefficient reflecting this association is positive.
---	---	--	---	--

3) Investigate whether adolescents with an internalising condition use social media differently than those with an externalising condition.	Hypothesis 3.2) Adolescents with internalising diagnoses will score lower than adolescents with externalising diagnoses in c) lack of control over time spent online, g) online honest self-disclosure and h) authentic self-presentation online.	We calculated power given a theoretical SESOI of $d = 0.4$, and a sample size of at least $N \geq 370$ individuals with internalising only and $N \geq 199$ with externalising only diagnoses. Results indicated 96% power to reject effects larger than the SESOI using the <code>'power.t.TOST'</code> function for equivalence testing and 99.8% power to detect the SESOI in the hypothesised direction using the <code>'pwr.t2n.test'</code> function in R.	Three linear regression models will test the association between each dimension of social media engagement and diagnostic category (three levels: no diagnosis, internalising, externalising), with the externalising diagnosis group set as the reference level. For this hypothesis, we will examine the regression coefficient for internalising vs externalising. We will employ an equivalence test to assess whether the 90% confidence interval (CI) for the association between social media use and mental health diagnosis lies between the equivalence bounds, -0.4 and 0.4. We will employ regression models to assess whether the association between social media use and diagnostic category (comparing internalising and externalising) follows the hypothesised direction.	 • If the CI for this association is within the equivalence bounds, it will suggest that adolescents with internalising only diagnoses do not have a meaningful difference in their social media use compared to adolescents with externalising only diagnoses. On the contrary, if the CIs do not fall within the equivalence bounds, it will suggest that the association is of meaningful magnitude. • A negative and statistically significant association between social media engagement and diagnostic category (comparing internalising and externalising) will suggest that adolescents with internalising only diagnoses lower higher on the examined dimension of social media engagement (c, g, h) compared to adolescents with externalising only diagnoses. • In summary, Hypothesis 3.2 will be supported if a) the CIs for this association do not fall within the equivalence bounds, b) the association between social media use and diagnostic category (comparing internalising and externalising) is statistically significant, b) the regression coefficient reflecting this association is negative.
---	--	--	--	--

Thank you for sending us the reviews for our manuscript “Social media use in adolescents with and without mental health conditions” (NATHUMBEHAV-22123378D). We are grateful for the feedback provided on the Stage 2 registered report. In this response, we include a point-by-point response to the editors and reviewers’ comments and highlight the changes made. A revised version of the manuscript and supplementary materials are attached as separate documents.

Editor

We thank the editor for their detailed feedback on the manuscript, which we address below.

E1.1) Please revise the abstract to not exceed 150w.

Below we report the previous and shortened abstract adhering to the 150 word count:

Previous: There is much concern that adolescent mental health declines and concurrent increases in social media use are linked. To address this, previous research has mostly employed questionnaires rather than diagnostic instruments of mental health, limiting the possibility of comparing patterns of social media use in adolescents with and without mental health conditions. In this Registered Report, we address this shortcoming by analysing the UK Mental Health of Children and Young People survey ($N = 3340$, 11-19 years old) that collected standardised multi-informant diagnostic mental health assessments, as well as time- and engagement-based social media measures. We used linear regressions and equivalence testing to test our hypotheses of differences in social media use between adolescents 1) with and without mental health conditions, and 2) with internalising/externalising conditions, compared both with each other and with those adolescents without a mental health condition. Adolescents with any mental health condition, compared to those without a condition, spent more time using social media and were less happy about their number of online friends. Adolescents with internalising conditions further reported higher social comparison, less control over time spent online, were more impacted by online feedback, and engaged in less honest self-disclosure than those without a condition. Adolescents with externalising conditions reported spending more time online, and were more impacted by online feedback than those without a condition. Lastly, adolescents with internalising conditions reported more online social comparison and less happiness with their number of online friends than those with externalising conditions. Overall, these findings highlight the importance of social media research on adolescents with and without mental health conditions to identify targets for interventions, policy and clinical practice.

Updated: “Concerns about the relationship between social media use and adolescent mental health are growing, yet few studies focus on adolescents with clinical-level mental health symptoms. This limits our understanding of how social media use varies across mental health profiles. In this Registered Report, we analyse nationally representative UK data (N = 3,340, aged 11–19) including diagnostic assessments by clinical raters alongside quantitative and qualitative social media measures. As hypothesised, adolescents with mental health conditions spent more time on social media and were less happy about online friends than adolescents without conditions. Additional differences emerged by condition type: adolescents with internalising conditions reported more time spent, social comparison and impact of feedback on mood, alongside lower happiness about online friends and honest self-disclosure. In contrast, those with externalising conditions reported higher time spent. These findings emphasise the need to consider diverse adolescent mental health profiles in policy and clinical practice.”

E1.2) The only changes that are allowed to your in-principle-accepted Stage 1 protocol are the tense of the sentences and any scientific errors. Please reverse all other changes.

We have reverted all changes to the Stage 1 report except those justified below. First, as explained in E1.4, we replaced the term "clinical vs. non-clinical population" with "adolescents with and without mental health conditions." This is because the term “clinical population” is typically used in epidemiological research to describe individuals recruited from clinical settings. However, as noted by the editor in their in-text comment, the participants in this study were neither recruited from a clinical setting nor diagnosed through a clinic. Instead, their mental health status was assessed by clinical raters as part of the MHCYP national survey. Therefore, we change the terminology to clarify this distinction throughout the manuscript:

Original: “clinical and non-clinical adolescent populations”

Updated: “adolescents with and without mental health conditions” (throughout the Main Manuscript and Supplementary Information)

Given this clarification, we would also like to retain two changes to the Stage 1 report, which we highlight below in green. However, if this is not possible, we are happy to revert to the initial version.

Updated: “The rationale for doing so is that questionnaires capturing continuous clinical ~~symptomatology in mostly healthy samples~~ symptoms are informative when reasoning about social media use in relation to the whole spectrum of mental health ~~in adolescents with~~, across types of severity and clinical conditions presentation.” (Main manuscript, Page 3, Line 44)

Updated: “Second, it ignores the potentially important differences between those who endorse symptoms on a questionnaire and those ~~who have received a formal mental health diagnosis~~ reach diagnostic criteria in standard clinical classifications. For example, an adolescent can score very highly on a questionnaire measuring depressive symptoms, but

not meet the criteria for a diagnosis if the queried symptoms have only been present for a short time, or if they are better explained by a different condition or situation.” (Main manuscript, Page 4, Line 50)

Updated: “In other words, although researchers would like to demonstrate the presence or absence of specific links between mental health conditions and social media use, the ~~samples~~ measures of psychopathology they ~~study~~ employ might not be appropriate for these goals.” (Main manuscript, Page 4, Line 59)

E1.3) ‘Significant’ should only be used in the context of statistical significance.

We thank the editor for this comment, we removed “significant”:

Updated: “Adolescents around the world have experienced a ~~significant~~ decline in their mental health over the past decade.” (Main manuscript, Page 3, Line 14)

E1.4) It seems that the terminology was changed throughout - was that because participants qualified for a diagnosis (i.e. met diagnostic criteria), but did not have an official diagnosis? If so, the change is fine (but on first mention of the changed terminology, a parenthetical statement needs to be added to explain to readers the change in terminology from the Stage 1 version). If there is another reason, please revert back to the original terminology.

Thank you for your feedback. As discussed in E1.1, we confirm that the change in terminology was applied for the reason you suggested. Namely, we wanted to clarify that participants "qualified for a diagnosis" but did not have a diagnosis from their GP. Instead, they were diagnosed as part of the MCHYP survey. A parenthetical statement has been added on the first mention of the term to explain this change in terminology from the Stage 1 version.

Updated: “We note that in the Stage 1 report, the terms ‘clinical and non-clinical population’ were used, but in Stage 2, this was changed to ‘adolescents with and without mental health conditions’ to clarify that participants in this study were not recruited or diagnosed by a clinic but instead underwent the mental health assessment as part of the MHCYP survey).” (Main manuscript, Page 4, Line 78)

E1.5) Why was the sensitivity analysis not conducted? All preregistered analyses must be performed and reported in the manuscript.

We have now included the sensitivity analyses for time spent on social media, separately for weekdays and weekends.

Updated: “Lastly, we tested our hypotheses for time spent separately for school days and weekends, rather than using a composite score (Supplementary Table 22).” (Main manuscript, Page 13, Line 338)

Updated: Added Supplementary Table 22. (Supplementary, Page 54)

E1.6) Please correct - while the revisions suggest a 5-point Likert scale, there are 6 labels.

We thank the editor for noticing this inconsistency. The Likert-scale had 5 points because we did not include “Don’t know” responses. We have now clarified this:

Updated: “Participants responded to these measures using a 5-point Likert scale (1 = Disagree a lot; 2 = Disagree a little; 3 = Neither agree nor disagree; 4 = Agree a little; 5 = Agree a lot; ~~6 = Don’t know~~). We omitted the “Don’t know” responses and coded 1-5 responses as continuous” (Main manuscript, Page 18, Line 510)

E1.7) This suggests that the power analysis was adjusted post-acceptance in principle of the Stage 1 protocol. Please explain.

Thank you for your comment. We would like to clarify that no adjustments were made to the power analysis post-acceptance. In the Stage 1 report, we indicated that we would apply a correction for multiple comparisons but did not specify the exact alpha level that we would set for this correction. In the Stage 2 report, we provided this clarification by specifying the corrected alpha level of 0.0125 to control the Type I error rate across our four tested hypotheses. Below, we report the original and updated version:

Original: “To account for multiple tests, we will provide false discovery rate corrected p -values for all models.”

Updated: “To control for the Type I error rate across multiple tests, we set a corrected alpha level of 0.0125, accounting for the false discovery rate across our four tested hypotheses for any given social media item.” (Main manuscript, Page 20, Line 564)

E1.8) Why did the alpha level change?

Thank you for your comment. As noted in response E1.7, in the Stage 1 report we indicated that we would correct for multiple comparisons but did not specify the exact corrected alpha level. In Stage 2, we clarified this by setting alpha to 0.0125 (instead of 0.05). This allowed us to control the Type I error rate across the four sets of hypotheses. Consequently, we updated the power sensitivity analysis to reflect this more stringent alpha level. This change ensures consistency with our commitment to correct for multiple comparisons.

Original: “In addition to our a-priori power calculations, we will conduct power sensitivity analyses once our final sample size is known. That is, we will determine the smallest effect size that is observable with 95% power, given an alpha of 0.05 and our final sample size.”

Updated: “In addition to our a-priori power calculations, we conducted power sensitivity analyses with the final sample size. That is, we determined the smallest effect size that is observable with 95% power, given the corrected alpha of 0.0125 and our final sample size (Supplementary Table 22).” (Main manuscript, Page 26, Line 775)

Reviewer 2

R2.1) As with the stage 1 report, I enjoyed reading the manuscript and am confident that this paper can make a very valuable contribution to the field. The presented study engages in an important comparison of social media use in adolescents with and without mental health conditions based on high-quality data. The authors kept the hypotheses and followed through

with the analyses as described in the Stage 1 report. The change addition that is described and explained in the Stage 2 manuscript is the additional use of non-parametric tests for all hypotheses as “not all assumptions of linear models were met” (p. 16, line 456).

We thank the reviewer for the overall appreciation of our work.

R2.2) Overall, I only have a few minor comments that should be relatively easy to address for the authors. Many of them simply relate to some of the wording in the paper. I will list those in chronological order in the following. To somewhat tone down the associated statement in the abstract I would propose to maybe replace the word “oversight” with “shortcoming” (same for the sentence on p. 5, line 97).

We agree with the reviewer’s comment, we changed the word to “shortcoming” in the introduction. Given the need to shorten the abstract to adhere to the 150-word count limit, we have now removed the word (see E1.1). If it is not possible to change the introduction (Stage 1 report), we will keep the original version:

Updated: “Studies that assess mental health with select questionnaires cannot comprehensively account for and investigate such clinical diversity. This is a substantial ~~oversight~~ shortcoming, given the need for research to understand how social media use relates to the growing number of adolescents experiencing mental health symptoms at clinical levels.” (Main manuscript, Page 5, Line 94)

R2.2) p. 6, line 134: I think “high levels of social comparisons” should be “high level of upward social comparisons” as that is what is described in the following parentheses here.

We thank the reviewer for this comment; we added the word “upward”. If it is not possible to change the introduction (Stage 1 report), we will keep the original version:

Updated: “However, high levels of upward social comparisons (i.e., comparisons with those believed to be of higher status than the self) have been associated with poorer mental health⁴⁵⁻⁴⁹.” (Main manuscript, Page 6, Line 131)

R2.4) I know this is really minor and maybe a bit nitpicky (apologies for that), but, given that the authors used secondary data, I think wording like “We measured...” (e.g., at the beginning of the “Measures” section on p. 9) should be replaced with something like “The study measured...” or “XY was measured with/using...”.

We thank the reviewer for their attention to details and agree that using “the study measured” is more accurate. Also in this case, if it is not possible to change the introduction (Stage 1 report), we will keep the original version:

Updated: “The study measured time spent on social media using two questionnaire items” (Main manuscript, Page 17, Line 473).

R2.5) I don't think that the tense of all verbs relating to the measures needs to be changed to past. For example, when describing general attributes of the instruments used for the "mental health diagnoses", I think it is fine (or even better) to stick with present tense (example: "The DAWBA used...", p. 10, line 279). I would say the same for statements related to the reporting of results (e.g., p. 13, lines 364-365: "we reported regression coefficients" -> "we report...").

We thank the reviewer for this comment, we have changed the sentence tense from past to present in Page 10-11 when referring to the DAWBA. To avoid inconsistencies in the use of present and past tense in the methods, we have kept the past tense for the "questions and hypotheses" section.

Updated: "After the initial screening items, the subsequent structured items in the DAWBA modules relate directly to diagnostic criteria in DSM-5 and ICD-10. They are close-ended questions about specific mental health symptoms (e.g., "In the last 4 weeks, have there been times when you have been very sad, miserable, unhappy or tearful?"). If a participant responds positively to these structured items, they are subsequently asked open-ended questions about these problems (e.g., "Please describe your mood - sadness or irritability - and your level of interest in things"). During the assessment, interviewers transcribe open-ended responses verbatim and are also able to add personal comments beneath each response." (Main manuscript, Line 531, Page 19).

R2.6) p. 11, line 283: What exactly is meant by "if a participant scored highly on the SDQ...". Does this refer to the "difficulties" part of this instrument?

We thank the reviewer for this comment. To clarify, this means that a participant scored above a certain score (i.e., clinical thresholds), which is reported in the DAWBA website that we reference in the study.

R2.7) On p. 12, lines 328-329, the authors write that "the simulated data and analysis code is available on OSF". However, I did not find a dataset in the repository. What is included, is the "code for simulating and analyzing the data". So, either the wording needs to be adapted or the (simulated) data should be uploaded to the associated repository.

We agree with the author that the term "simulated data" is not clear. Given that we only partially simulate the dataset for the power analyses, we changed the phrasing:

Updated: Further, the ~~simulated data and~~ analysis code is available on OSF (https://osf.io/s2dwu/?view_only=5acced2ddb884f9481d439a7746f4dd1). (Manuscript, Page 20, Line 574).

R2.8) Note: I have provided my remarks regarding the OSF repo and the code it contains in the separate field "Remarks on code availability".

We appreciate that the reviewer looked into the code and provided the comments below.

R2.9) As a final minor note, the manuscript contains a number of typos and punctuation errors that warrant another thorough proofreading (this could be done/facilitated via tools like Microsoft Editor, Grammarly, DeepL Write or Writefull).

We thank the reviewer for this comment; we have proofread the manuscript and corrected some typos, which are shown in the track-changes version.

Reviewer #2 (Remarks on code availability):

R2.10) The authors have provided the full code for the power analysis and simulation as well as the descriptive and hypotheses-testing analysis via the Open Science Framework (OSF). The code is well-structured and properly commented. I was able to reproduce the power analysis and simulation (Stage 1 code). I could not reproduce the Stage 2 analyses as the underlying data cannot be shared. Some things I noticed when checking the code: 1. Stage 1 code: The knitted HTML file is only provided for the `mhcyp_analysis.Rmd`, not for `mhcyp_power.Rmd`. Is there a reason for this or is this simply a minor oversight?

We thank the reviewer for this comment; this was indeed an oversight, we have now also included the HTML file for the `mhcyp.power`. We also specified information about each script in the ReadMe file added to OSF. The scripts are now organised separately for the Stage 1 and Stage 2 report.

R2.12) I would suggest that the authors provide ReadMe (esp. for the Stage 2 – analysis code) with further information, such as the order in which scripts should be run or information on using `renv` (as a side note: I very much appreciate that the authors use `renv` to increase computational reproducibility).

We thank the reviewer for this comment; we agree and have now added a ReadMe file to the OSF with detailed instructions on all the scripts.

Reviewer 3

R3.1) This is a compelling manuscript that carefully reviews the relationship between qualitative and quantitative aspects of social media use with youth mental health diagnostic domains. The study is well designed, and the methods are transparently communicated. The introduction explains the current literature related to social media and youth mental health superbly, recognizing the many nuances of this area of study and topics for which the research is limited, and communicates the need for this study clearly. The results are quite interesting and are likely to be very useful to informing future research as well as prevention/intervention efforts. Below, I offer minor comments meant to increase clarity and transparency of the manuscript.

We thank the reviewer for this positive comment and their feedback on this work.

R3.2) The introduction is quite compelling. One very minor comment would be to clearer earlier

on the construct related to monitoring. There is much discussion of parental monitoring within the literature. Given that, a clearer notation related to self-monitoring may be helpful.

We thank the reviewer for this comment. Given that the Stage 1 (i.e., introduction) cannot be changed, we will not add information on monitoring. However, we note that we expand on the literature related to monitoring in Table 3 on Page 59, which should give an overview for the reader on this topic. Particularly, the table reports:

Original: “Adolescents and adults with depression, anxiety and eating disorders, among other internalising conditions, have a higher tendency to seek feedback from others¹¹¹, particularly when negative, and engage in more reassurance seeking⁵⁶. Further, compulsivity in social media checking behaviours has been associated with anxiety, depression and problematic smartphone use¹¹², therefore, becoming a cause of clinical and developmental concern⁷⁷. Consistent with these results, online feedback-seeking has been associated with depressive symptoms in adolescent community samples⁵⁵. This association holds even after accounting for the effects of the overall frequency of technology use, offline excessive reassurance-seeking and prior depressive symptoms.” (Main Manuscript, Table 3, Page 59)

R3.3) It would be helpful to briefly comment on the limitations of the sensitivity analyses within the main paper, e.g., the limited sample sizes and exploratory nature of the analyses.

We thank the reviewer for this comment; we have added a sentence in the result section to note the limitations of the sensitivity analyses:

Updated: “We note that these sensitivity analyses were exploratory and conducted on relatively small sample sizes, which limits the robustness of the findings.” (Main manuscript, Line 349, Page 16)

R3.4) While a reasonable and unchangeable aspect of the study, I would suggest mentioning the data is now 7 years old as a potential limitation. Trends related to social media use change at a fast pace. It is not totally clear if some of the constructs studied would remain the same as new and different kinds of platforms become more prevalent (e.g., public vs. private platforms, those on which communication is anonymous, or platforms on which moderation is common).

We agree and have therefore added a sentence to acknowledge the potential limitation of using seven-year-old data and the evolving nature of social media trends.

Updated: “Moreover, we acknowledge that the data was collected in 2017. While the core constructs of social media use remain relevant, the rapid evolution of platforms and user behaviors presents a potential limitation when applying our findings to current trends.” (Page 15, Lines 420).

R3.5) Supplementary Figure 5 seems to pertain to sex rather than gender.

We thank the reviewer for this comment; we agree and have now changed the figure legend and title to report “sex” instead of “gender”.

Updated: “Descriptive statistics split by social media items, ~~gender~~ sex and diagnostic groups are reported in Supplementary Tables 5-8, Figure 3-5).” (Main manuscript, Line 209, Page 8)

Updated: Supplementary Figure 5, Page 36.

R3.6) SES could be a key factor that could contribute to differences in qualitative aspects of use, e.g., social comparison. A suggestion would be to provide an additional supplementary figure, like Figure 5, based on levels of SES.

We agree that SES could be an important factor influencing qualitative aspects of social media use, such as social comparison. However, given the scope of this paper and the current length, we have decided not to include this additional analysis at this stage. We believe that this is an interesting avenue for future research and would be better addressed in a separate, more focused investigation.

R3.7) Supplementary Figures 3 and 4, while nice looking, are difficult to decipher. Please define the axes.

We thank the reviewer for this comment and have clarified the axes in Supplementary Figures 3 and 4, as reported below.

Updated: “Dimensions of social media use by mental health diagnosis group. The x-axis represents the grouping variable (Any Mental Health Diagnosis: Yes/No, or Type of Mental Health Diagnosis: None, Externalising, Internalising). The y-axis displays social media variables, measured on a 5-point Likert scale, except Social Media Time Spent, which is measured on a 10-point Likert scale. The plot includes violin shapes to show the distribution of data, with boxplots indicating the median and interquartile range, and black dots representing the mean with bootstrapped 95% confidence intervals.” (Supplementary, Page 34, Figures 3 and 4)

Other changes

Below, we report two other changes that we implemented:

C1.1) Time spent

We also note that we corrected an error in the way the self-reported social media time spent was initially coded, which did not fully account for the survey routing. As detailed in Supplementary Table 1, the SMTimeSpent question (weekday and weekend; SMTimeSpentS/SMTimeSpentw) were only asked of adolescents who responded to another survey question that they use social media (SMUse) and the frequency of use (SMFreqofUse) is “daily or most days”. Adolescents who responded that they used social media sites between “a few times a week” to “less often than once a month” did not complete the SMTimeSpentS/W questions. Supplementary Table 1 provides an

overview of the routing for the relevant social media questions. Supplementary Table 2 details our approach to handling missing data for the SMTimeSpent variables, which was not applied in the Stage 1 report and has now been corrected.

C1.2) Figure 1 legend:

We updated the legend of Figure 1 to be more detailed:

Updated: “Note. Data are presented as mean differences based on Hedges's g effect size (g) and its corresponding 90% confidence interval. The shaded area indicates the smallest effect size of interest (SESOI; $g = 0.4$). If 90% CI lies completely within the SESOI, we concluded equivalence and, therefore, no meaningful differences. Bolded effect sizes reflect comparisons for which we supported our hypothesis while faded effect sizes reflect comparisons for which we did not support our hypotheses. The starting sample size includes social media users ($N = 519$ for adolescents with any mental health condition, $N = 282$ for internalising conditions, $N = 104$ for externalising conditions and $N = 2821$ for no mental health condition). However, the exact sample size for each comparison is reported in Supplementary Tables 5 and 6 and differs for each dimension of social media use, given that we planned to analyse those separately.” (Main manuscript, Page 40).

We greatly appreciate your support and feedback.

Yours sincerely,

Luisa Fassi on behalf of all co-authors
PhD Researcher
MRC Cognition and Brain Sciences Unit,
Department of Psychiatry,
University of Cambridge